# Imitating Deep Learning Dynamics via Locally Elastic Stochastic Differential Equations

**Jiayao Zhang**      **Hua Wang**      **Weijie J. Su**

University of Pennsylvania

{zjiayao,wanghua,suw}@wharton.upenn.edu

## Abstract

Understanding the training dynamics of deep learning models is perhaps a necessary step toward demystifying the effectiveness of these models. In particular, how do data from different classes gradually become separable in their feature spaces when training neural networks using stochastic gradient descent? In this study, we model the evolution of features during deep learning training using a set of stochastic differential equations (SDEs) that each corresponds to a training sample. As a crucial ingredient in our modeling strategy, each SDE contains a drift term that reflects the impact of backpropagation at an input on the features of all samples. Our main finding uncovers a sharp phase transition phenomenon regarding the *intra-class* impact: if the SDEs are *locally elastic* [19] in the sense that the impact is more significant on samples from the same class as the input, the features of the training data become linearly separable, meaning vanishing training loss; otherwise, the features are not separable, regardless of how long the training time is. Moreover, in the presence of local elasticity, an analysis of our SDEs shows that the emergence of a simple geometric structure called the neural collapse of the features. Taken together, our results shed light on the decisive role of local elasticity in the training dynamics of neural networks. We corroborate our theoretical analysis with experiments on a synthesized dataset of geometric shapes and CIFAR-10.

## 1  Introduction

Deep learning models have achieved significant empirical success over the past decade across a wide spectrum of domains spanning computer vision, natural language processing, and reinforcement learning [31, 43, 48]. Despite these remarkable achievements at the empirical level, there is still much to learn about deep neural networks, as evidenced by the fact that almost all important advances concerning architecture design and optimization for deep learning are based on heuristics, without much input from a theoretical perspective [20, 11, 21, 27].

An important step toward opening these black-box models and unveiling their formidable details is to quantitatively understand the impact of backpropagation in deep learning training. While there has been a continued effort to demystify how simple optimization methods give rise to impressive generalization performance, for example, [49, 26, 4], this is by no means an easy problem, perhaps because of the daunting nonconvex nature of neural networks. Accordingly, for near-term purposes, a more practical approach is to take a phenomenological viewpoint by relating simple empirical patterns to the effectiveness of deep learning models.

In this spirit, we are interested in how data from different classes gradually become separable in their feature space by repetitively calling backpropagation. From a phenomenological viewpoint, this question can be addressed by first analyzing the impact of a single update using a stochastic gradient on the performance of the neural networks. More precisely, imagine that the gradient is evaluated in an image of a cat, how does the hidden representation of another image—say, an image of another

35th Conference on Neural Information Processing Systems (NeurIPS 2021).

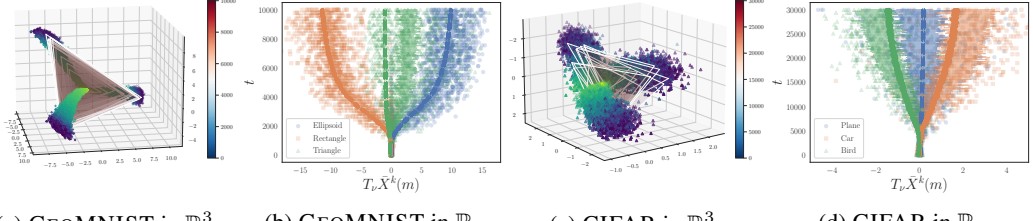

| (a) GEOMNIST in $\mathbb{R}^3$. | (b) GEOMNIST in $\mathbb{R}$. | (c) CIFAR in $\mathbb{R}^3$. | (d) CIFAR in $\mathbb{R}$. |

Figure 1: **Separation of features (logits).** GEOMNIST dataset ((a)—(b))) and CIFAR dataset ((c)—(d)) trained using $K = 3$ classes on a variant of the ALEXNET model. Separation in $\mathbb{R}$ is done by projecting the logits to $\boldsymbol{\nu} \in \mathbb{R}^3$, which is set to be the difference between a pair of class means $\bar{\boldsymbol{X}}^k(T) - \bar{\boldsymbol{X}}^l(T)$ for a large $T$, where $k, l$ are chosen heuristically. The dashed lines in (b) and (d) are simulated paths from Equation (9) using estimated $\widehat{\alpha}(t)$ and $\widehat{\beta}(t)$. More details are given in Section 4 and Appendix D.2.

cat or an image of a plane—evolves because of the backpropagation? Recent studies answer this question by introducing a phenomenon called *local elasticity*, which, roughly speaking, means that the impact is generally larger on a similar sample (an image of another cat) than on a dissimilar sample (an image of a plane) [19].

Motivated by the phenomenon of local elasticity, we propose a model that captures the interaction between different training samples during deep learning training using a set of stochastic differential equations (SDEs) that reflect local elasticity in neural networks. Characterizing the *intra-class* and *inter-class* effects is an essential component of our modeling strategy, each of which contains a drift term that imitates the impact of backpropagation on specific training data of all samples.

Our main finding uncovers a sharp phase transition phenomenon regarding the intra-class and inter-class impact; if the SDEs are *locally elastic* in the sense that the impact is more significant on samples from the same class as the input (the intra-class effect is strictly greater than the inter-class effect), the features of the training data are guaranteed to be linearly separable, meaning vanishing training loss; otherwise, the features are not separable, no matter how long the training time is. This result provides convincing theoretical evidence for the presence of *local elasticity* in deep learning [19]. Our model is also quite accurate in simulating the feature dynamics of deep learning. As shown in Figure 1 and detailed in Section 4, the dynamics of the simulated logits are reasonably close to the real dynamics of deep learning on both synthetic and real datasets, indicating a well-suited model for theoretical and practical purposes. Moreover, in the presence of local elasticity, our SDEs also predict the emergence of a simple geometric structure called neural collapse of features [38].

Taken together, our results shed light on the decisive role of local elasticity in the training dynamics of neural networks. We corroborate our theoretical analysis with experiments on a synthetic dataset of geometric shapes, as well as on CIFAR-10. The experimental evidence consistently supported our model, which provides new insights into the dynamics of deep learning training.

## 1.1 Related Work

**Dynamics in Deep Neural Nets.** Many properties of linear deep neural nets are relatively well understood, such as the loss landscapes [25], trajectory-based convergence [2, 12], and implicit acceleration [3]. Exact solutions of the training dynamics can be obtained in certain initialization schemes [40, 41, 30]. In the presence of non-linearity, various assumptions are generally made. [39, 17] studied the dynamics of shallow neural nets with non-linearity and the neural tangent kernel (NTK) literature [22, 4, 14] linearizes the network function of an infinitely wide neural net at initialization, which is similar to that of the deep Gaussian process literature [9, 18, 32] (a treatise comparing and contrasting them can be found in [47]). Although as approximations, NTKs are generally used when studying optimization trajectories of neural nets such as [34], which also appears implicitly in many works when studying the optimization trajectories of neural net training [45, 1, 13, 24, 7].

**SGD as SDEs in Neural Nets.** The study of dynamics or trajectories of weights in deep neural nets via SDEs relies on the more precise characterization of stochasticity [33, 36]. Built on top of

this formalism, [44] studied the trade-off between batch size and learning rate, [23], analyzed factors influencing the quality of local minima, [5] studied the behavior of the SGD near local minima, and [42] studied the effect of learning rates. Although SGD-SDE approximation requires an infinitesimal learning rate, [35] verified that the SDE approximation can be meaningful in practical settings and obtained necessary conditions for the validation of such approximation.

**Local Elasticity and Phenomenological Models.** Local elasticity is proposed in [19] as a phenomenological approach to reasoning the behaviors of neural networks. This phenomenon has inspired several works on generalization bounds [10] and an improvement on the NTK [6].

## 2 Binary Separation via LE-SDE

### 2.1 Setup, Notations and Assumptions

Throughout the paper, we work with the following setup and assumptions. For ease of reading, vectors and matrices are written in boldface and we denote by $[n]$ the set $\{1, \ldots, n\}$. When there is no ambiguity, we will write both $X(t)$ and $X_t$ for a continuous-time (possibly stochastic) process.

**Classification Problem.** Consider a $K$-class classification problem with $K \geq 2$, with each class having $n$ training examples. We denote by $\boldsymbol{z}_i^k \in \mathbb{R}^d$ the $i$-th sample of the $k$-th class, and $y_i^k \in [K]$ its label, where $i \in [n], k \in [K]$. A neural net is a function $f : \mathbb{R}^d \to \mathbb{R}^K$ that maps the samples to logits ( pre-activation of the softmax).

**Feature Vectors.** We denote by $\boldsymbol{X}_i^k(m) \in \mathbb{R}^p$ a $p$-dimensional feature of the $i$-th sample in the $k$-th class learned by the neural net at iteration $m$. For example, it can be the logits or the output of the second-to-last layer. Assume that the initial values $\boldsymbol{X}_i^k(0)$ are i.i.d. samples from some distribution for each $i \in [n], k \in [K]$. We use $i, j \in [n]$ as indices for an individual sample, $k, l \in [K]$ for classes, and capital letters $J$ and $L$ to indicate random samples from $\mathrm{Unif}([n])$ and $\mathrm{Unif}([K])$, respectively.

**Training Dynamics.** We model the training dynamics in neural nets under SGD with an emphasis on *local elasticity*. At the $m$-th iteration, the $J_m$-th sample is sampled from the $L_m$-th class, where $J_m \sim \mathrm{Unif}([n])$ and $L_m \sim \mathrm{Unif}([K])$. Training on $\boldsymbol{z}_{J_m}^{L_m}$ affects the features of another data sample $\boldsymbol{z}_i^k$ in the form of

$$\boldsymbol{X}_i^k(m) - \boldsymbol{X}_i^k(m-1) = h \cdot E_{k,L_m}(m)\boldsymbol{X}_{J_m}^{L_m}(m-1) + \sqrt{h}\boldsymbol{\zeta}_i^k(m-1) \tag{1}$$

where $i \in [n], k \in [K]$, $h$ is the step size, and $\boldsymbol{\zeta}_i^k(m)$ is the noise term that is modeled as Gaussian noise. The scalar $E_{k,L_m}$ measures the strength of local elasticity that $\boldsymbol{z}_{J_m}^{L_m}$ exerts on $\boldsymbol{z}_i^k$ at iteration $m$. We assume $\boldsymbol{X}_i^k(0), \boldsymbol{\zeta}_i^k(m)$ are jointly independent.

**Local Elasticity.** Clearly, by writing $E_{k,l}(m)$, we assume that this effect depends only on the class $l$, $k$, and time $m$. We write the effect matrix as $\boldsymbol{E}(m) = (E_{k,l}(m))_{k,l=1}^K$. For ease of exposition, we assume $\boldsymbol{E}$ only consists of two values $\alpha(m)$ and $\beta(m)$, with $\alpha(m)$ representing the *intra-class* effect and $\beta(m)$ the *inter-class* effect. To this end, we assume the *effective training assumption*, that is, as training progresses, the features become more discriminative: features from the same class are more similar, whereas those from different classes are more distinct, as measured by some similarity measure in the feature space. We also assume that the LE effect is "*proportional*" to the feature $\boldsymbol{X}_{J_m}^{L_m}(m)$ itself. We generalize this point in Section 3 by introducing a transformation matrix $\boldsymbol{H}$ on the features.

### 2.2 Binary LE-SDE

Our construction of equation (1) emphasizes the effect of intra- and inter-class effects on the dynamics of *features*, and thus differs from the usual weight dynamics that is common in the literature. Before deriving the general form of our locally elastic SDE (LE-SDE), we shall familiarize the reader with our model by demonstrating this in the case of binary classification ($K = 2$) with a one-dimensional features ($p = 1$) — the output of the model to be fed into the softmax function, also called the *logit*.

Let the intra-class effect be $E_{11} = E_{22} = \alpha$, and the inter-class effect is $E_{12} = E_{21} = \beta$, both of which are time-independent. Expanding equation (1), for $1 \leq i, j \leq n$ and $m \geq 0$, when we train the

model on the $J_m$-th training example from the $L_m$-th class, we have

$$\begin{cases} X_i^1(m) &= X_i^1(m-1) + h \cdot \alpha X_{J_m}^{L_m}(m-1) + \sqrt{h} \cdot \zeta_i^{L_m}(m-1), \\ X_j^2(m) &= X_j^2(m-1) + h \cdot \beta X_{J_m}^{L_m}(m-1) + \sqrt{h} \cdot \zeta_j^{L_m}(m-1). \end{cases}$$

In the limit of $h \to 0$, we can show that $X_i^k(m)$ approximates some continuous-time stochastic processes $X_i^k(t)$ (under the identification of $t = mh$) governed by the set of stochastic differential equations as follows:

$$\mathrm{d}X_i^k(t) = \left( \frac{\alpha}{2} \bar{X}^k(t) + \frac{\beta}{2} \bar{X}^{3-k}(t) \right) \mathrm{d}t + \sigma \, \mathrm{d}W_i^k(t), \quad t \geq 0, k \in [K], i \in [n] \tag{2}$$

where $\bar{X}^k(t) := \left( X_1^k(t) + \cdots + X_n^k(t) \right) / n$, and $W_i^k$ are independent standard Wiener processes. The detailed derivation is given in Appendix A.

Now averaging over $i$ for each $k$ in equation (2), we obtain the following set of two ordinary differential equations (ODEs) governing the per-class means that $\bar{X}^k(t)$ for $k = 1, 2$:

$$\mathrm{d}\bar{X}^k(t) = \left( \frac{\alpha}{2} \bar{X}^k(t) + \frac{\beta}{2} \bar{X}^{3-k}(t) \right) \mathrm{d}t + \sigma \, \mathrm{d}\frac{W_i^k(t) + \cdots + W_i^k(t)}{n}.$$

Taking the limit of $n \to \infty$, we observe that $\sigma \mathrm{d}\frac{W^1(t) + \cdots + W^n(t)}{n} \Rightarrow 0$. Thus, the above display converges weakly to the following ODE:

$$\frac{\mathrm{d}\bar{X}^k(t)}{\mathrm{d}t} = \frac{\alpha}{2} \bar{X}^k(t) + \frac{\beta}{2} \bar{X}^{3-k}(t). \tag{3}$$

With the initial conditions $\mathbb{E}X_i^k(0) = c_k$, the solution to the above ODE is

$$\bar{X}^1(t) = \frac{c_1 - c_2}{2} \mathrm{e}^{\frac{\alpha-\beta}{2}t} + \frac{c_1 + c_2}{2} \mathrm{e}^{\frac{\alpha+\beta}{2}t}, \quad \bar{X}^2(t) = -\frac{c_1 - c_2}{2} \mathrm{e}^{\frac{\alpha-\beta}{2}t} + \frac{c_1 + c_2}{2} \mathrm{e}^{\frac{\alpha+\beta}{2}t}.$$

In the finite-sample setting, we may replace $\bar{X}^1, \bar{X}^2$ in the SDE (2) by their deterministic solutions and obtain

$$\begin{cases} X_i^1(t) &= \frac{c_1-c_2}{2} \mathrm{e}^{\frac{\alpha-\beta}{2}t} + \frac{c_1+c_2}{2} \mathrm{e}^{\frac{\alpha+\beta}{2}t} - c_1 + X_i^1(0) + \sigma W_i^1(t), \\ X_j^2(t) &= -\frac{c_1-c_2}{2} \mathrm{e}^{\frac{\alpha-\beta}{2}t} + \frac{c_1+c_2}{2} \mathrm{e}^{\frac{\alpha+\beta}{2}t} - c_2 + X_j^2(0) + \sigma W_j^2(t). \end{cases}$$

We are now ready to derive the condition under which these $2n$ feature vectors become *asymptotically separable*, that is, $\min_i X_i^1(t) > \max_j X_j^2(t)$ or $\max_i X_i^1(t) < \min_j X_j^2(t)$ as $t \to \infty$.

**Theorem 2.1** (Separation in Binary Classification). *Given the feature vectors $X_i^1(t)$, $X_j^2(t)$ for $i, j \in [n]$, as $t \to \infty$ and large $n$,*

1. *if $\alpha > \beta$, they are asymptotically separable with probability tending to one,*

2. *if $\alpha \leq \beta$, they are asymptotically separable with probability tending to zero.*

This result indicates a sharp phase transition when $\alpha$ is *just* above $\beta$, that is, in the regime of *local elasticity*. As long as the intra-class effect is slightly greater than the inter-class effect, separation is guaranteed. This simple model already captures local elasticity and reveals the important role it plays in the *perfect separation* of training samples. We can generalize this model to more realistic settings: when there are multiple classes, when features are high-dimensional, and when the LE matrix $\boldsymbol{E}$ is time-dependent. In the next section, we discuss each of these three generalizations in more depth.

## 3 General LE-SDE Model

Now, we consider the general case where $K \geq 2$ and the feature vectors are $p$-dimensional with $p \geq K$. Inquisitive readers may have already noticed that Theorem 2.1 only asserts the *emergence* of the separation of features, while being inconclusive to their relative orders at separation, that is, *which class converges to where?* This drawback is intrinsic to the toy model as neither intra-class nor inter-class effect identifies different classes. In this section, we introduce the general LE-SDE model that alleviates this difficulty with the help of an extra block matrix $\boldsymbol{H}$ with the $(i, j)$-th block $\boldsymbol{H}_{i,j}$ models how features in the $j$-th class affect those in the $i$-th class, which also partially defines how

classes are separated in higher dimensions. In the local elasticity formalism, $\boldsymbol{H}_{i,j}$ can be viewed as inducing a metric on the feature space under which local elasticity manifests.

As hinted before, in the case of multiple-class features in higher dimensions, we want to guarantee a stronger separation: to know which class converges to where, thus incorporating supervision from label information. For example, when the features are logits (outputs of the neural nets) and the model is trained under the softmax cross-entropy loss, previous work suggests they separate according to specific geometric structures [38]. To this end, we need to adjust the raw feature vectors $\boldsymbol{X}_i^k$ with a proper transformation that incorporates the label information into the dynamics. This motivates the following modification of the dynamics (1) by adding an extra transformation $\boldsymbol{H}_{k,L_m} \in \mathbb{R}^{p \times p}$ to the features. For $k \in [K]$, $i \in [n]$, and at iteration $m$, we have the following:

$$\boldsymbol{X}_i^k(m) = \boldsymbol{X}_i^k(m-1) + h \cdot E_{k,L_m}(m) \boldsymbol{H}_{k,L_m}(m) \boldsymbol{X}_{J_m}^{L_m}(m-1) + \sqrt{h} \boldsymbol{\zeta}_i^k(m-1). \qquad (4)$$

The $\boldsymbol{H}_{k,L_m}(m)$ term models the LE effect as *proportional* to a linear "transformation" of the features. The dynamics in Equation (1) are special cases when $\boldsymbol{H}_{k,l}(m) \equiv \boldsymbol{I}_p$ for all $k, l \in [K]$. By specifying a proper $\boldsymbol{H}$, we can overcome the limitation in our toy example of not knowing which class converges to where. We specify interesting choices of $\boldsymbol{H}$ in Section 3.2.

A further step of abstraction is to write $\widetilde{\boldsymbol{X}}^k(m)$ instead of $\boldsymbol{X}_i^k(m)$, to indicate one generic sample from the distribution $\mathcal{D}^k(m)$ of all the features of class $k$ at iteration $m$. As in Section 2.2, we can derive the continuous dynamics of equation (4) in the limit of $h \to 0$ in the same way as equation (2). Similar to writing $\widetilde{\boldsymbol{X}} = (\widetilde{\boldsymbol{X}}^k)_{k=1}^K \in \mathbb{R}^{Kp}$ for the concatenation of per-class features, $\bar{\boldsymbol{X}} = (\bar{\boldsymbol{X}}^k)_{k=1}^K \in \mathbb{R}^{Kp}$ is the concatenation of per-class mean features. Our model (4) approximates the following SDE with identification $t = mh$ as $h \to 0$. We term this model the LE-SDE:

$$\mathrm{d}\widetilde{\boldsymbol{X}}_t = \boldsymbol{M}_t \bar{\boldsymbol{X}}_t \, \mathrm{d}t + \boldsymbol{\Sigma}_t^{\frac{1}{2}} \, \mathrm{d}\boldsymbol{W}_t, \qquad (5)$$

where $\boldsymbol{W}_t$ is the standard Wiener process in $\mathbb{R}^{Kp}$, $\boldsymbol{\Sigma}_t$ is the covariance matrix, and $\boldsymbol{M}_t \in \mathbb{R}^{Kp \times Kp}$ is a $K \times K$ block matrix, with each block of size $p \times p$. The $(k,l)$th block of $\boldsymbol{M}_t$ is $E_{k,l}(t) \boldsymbol{H}_{k,l}(t)/K$ when $l \neq k$, and $E_{l,l}(t) \boldsymbol{H}_{l,l}(t)/K$ when $l = k$. The rationale for dividing $K$ is that we assume that the data are balanced; therefore, each of the $K$ possible classes has an equal chance of being sampled, as proved in Equation (2), where $K = 2$. In Appendix E, we discuss how we can generalize this to model SGD with mini-batches, imbalanced data, and label corruptions.

Taking expectation with respect to the randomness arising from sampling $\widetilde{\boldsymbol{X}}_t$ from its distribution, the per-class mean $\bar{\boldsymbol{X}}_t$ satisfies the following system, which we term the LE-ODE:

$$\bar{\boldsymbol{X}}_t' = \boldsymbol{M}_t \bar{\boldsymbol{X}}_t. \qquad (6)$$

Under the assumptions in Section 2.1, we define $\gamma(t) = \min \{\alpha(t) - \beta(t), \alpha(t) + (K-1)\beta(t)\}$, $A(t) = \int_0^t \alpha(\tau) \, \mathrm{d}\tau$, $B(t) = \int_0^t \beta(\tau) \, \mathrm{d}\tau$, and $\Gamma(t) = \min \{A(t) - B(t), A(t) + (K-1)B(t)\}$.

## 3.1 The Separation Theorem

Similar to the discussions in Theorem 2.1, the LE-SDE allows us to derive the separability result for a general $K$ and $p \geq K$. We say the feature vectors $\left\{(\boldsymbol{X}_i^k)_{i \in [n]}\right\}_{k \in [K]}$ are *separable* if for any two classes $k \neq l$, there exists a hyperplane in $\mathbb{R}^p$ that linearly separates the features of the two classes. To characterize the separation as in Theorem 2.1, we need conditions on $\alpha(t), \beta(t)$ as therein. Intuitively, when $\gamma(t) \leq 0$, the classes cannot be separated, even in a pairwise manner. Therefore, we focus on a more interesting case when $\gamma(t) > 0$. We now state the following characterization theorem of separability for general LE-SDE dynamics:

**Theorem 3.1** (Separation of LE-SDE). *Under our working assumptions in Section 2.1, and in the case of local elasticity (i.e., $\gamma(t) > 0$), assume $\boldsymbol{H} = (\boldsymbol{H}_{ij})_{ij}$ is positive semi-definite (PSD) with positive diagonal entries. As $t \to \infty$, we have[1]:*

*1. if $\gamma(t) = \omega(1/t)$, the features are separable with probability tending to 1;*

---

[1] Here, $\gamma(t) = \omega(1/t)$ stands for $\gamma(t) \gg 1/t$ as $t \to \infty$. For example, $1/t^{0.5} = \omega(1/t)$ and $(t \ln t)^{-1} = o(1/t)$ as $t \to \infty$.

2. *if $\gamma(t) = o\,(1/t)$, and the number of per-class-feature $n$ tending to $\infty$ at an arbitrarily slow rate, the features are asymptotically pairwise separable with probability $0$.*

This theorem sheds light on the crucial impact of the local elasticity effect for separation in a general case. The proof to Theorem 3.1 as well as discussions on the empirically best ways of choosing the universal direction $\nu$ (i.e., a direction that does not depend on the class index) are detailed in Appendix C.2.

## 3.2   Two Specific Models

We next discuss two specific choices of the $H$ matrix that allows us to analyze $\widetilde{X}$ precisely.

### 3.2.1   Isotropic Feature Learning Model

As a straightforward extension to Section 2.2, we can simply choose $H_{lk} = I_p$ to be the identity matrix. This choice of $H$ is PSD, and thus, we can apply Theorem 3.1 to obtain the conditions for asymptotic separation. In this case, the solution $\bar{X}(t)$ to the LE-ODE can be computed analytically as given in the following proposition.

**Proposition 3.2** (I-model). *Let $H_{k,l} = I_p$, then the solution to the LE-ODE* (6) *is given by*

$$\bar{X}(t) = c\,e^{\frac{1}{K}A(t) - \frac{1}{K}B(t)} + \left(1_K \otimes c_0\right) e^{\frac{1}{K}A(t) + \frac{K-1}{K}B(t)}, \tag{7}$$

*where $c = (c_k)_{k=1}^K \in \mathbb{R}^{Kp}$ and $c_0 \in \mathbb{R}^p$ are constants with $\sum_{k=1}^K c_k = 0 \in \mathbb{R}^p$ and $\bar{X}(0) = c + c_0$.*

The derivation of Equation (7) is deferred to Appendix C.2.1. From equation (7), we can easily reconstruct Theorem 3.1 in this special case. The difference between a feature vector from class $k$ and that from class $l$ at time $t$ is given by $(c_k - c_l)e^{\frac{1}{K}A(t) - \frac{1}{K}B(t)} + \Sigma^{1/2}(W_t^k - W_t^l)$, provided that the first deterministic term dominates the random second term, thus ensuring separation, which are precisely the conditions specified in Theorem 3.1. We term this model as the *isotropic feature model*, or I-model for short; as the $H$ matrix has identity matrices as its blocks and consequently the dynamics do not prescribe any preferred directions for each class.

### 3.2.2   Logits-as-Features Model

An important type of features in neural nets is the *logits*, the outputs of the neural net before the softmax layer. A logit vector (or logits) is $K$-dimensional, and in this model we identify $\widetilde{X}^k(t)$ as the logits at time $t$ of a generic sample from the $k$-th class. In a well-trained neural net, the logits of a learned data instance from the $k$-th class should have its $k$-th logit being the largest, and heuristically, the other coordinates should be approximately equal and negative. As we shall detail in Appendix B, the exact dynamics of neural net training pushes the logits $\widetilde{X}^k$ by its *margin*, $d_k := e_k - \text{softmax}(\widetilde{X}^k)$, which roughly aligns with the direction of $e_k - 1_p/K$. This suggests us how to choose the metric under which local elasticity acts: we can choose $H_{l,k}$ such that it always aligns $\widetilde{X}^k$ in the direction of $d_k$, that is,

$$H_{ij} = \bar{H}^j := \frac{d_j d_j^\top}{\|d_j\|_2^2} \in \mathbb{R}^{p \times p}, \quad d_j := e_j - \frac{1}{K}1_p \in \mathbb{R}^p, \quad j \in [K]. \tag{8}$$

Roughly speaking, the map $x \mapsto \bar{H}^j x$ projects $x$ in the direction of $d_j$ and ideally aligns $x$ with $d_j$ after iterative applications; hence, $\bar{H}^j$ can be viewed as an approximation of the nonlinear transformation in the exact dynamics in the sense that the direction of their stationary point coincides. Furthermore, $\bar{H}^j$ thus defined has operator norm 1; thus, it does not affect the magnitudes, but only directions. Note that $H$ does not satisfy the condition in Theorem 3.1 as it is not symmetric; yet the separation theorem can be easily extended in light of the following proposition.

**Proposition 3.3** (L-model). *Let $H$ be the same as in equation* (8)*, then the solution to the LE-ODE* (6) *is given by*

$$\bar{X}(t) = c_0 + C_1 d\,e^{\frac{1}{K}A(t) - \frac{1}{K}B(t)} + \left(\sum_{l=1}^{K-1} C_{2l} f_l\right) e^{\frac{1}{K}A(t) + \frac{1}{K(K-1)}B(t)}, \tag{9}$$

*where $f_l$'s are fixed vectors in $\mathbb{R}^{K^2}$, $c_0 \in \mathbb{R}^{K^2}$ is a constant vector with $K(K-1)$ degrees of freedom, and $C_1, C_{2l} \in \mathbb{R}$, for $l \in [K-1]$ are constants.*

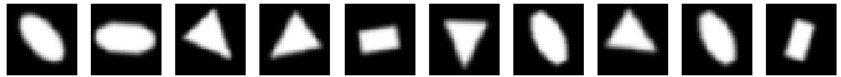

Figure 2: Samples from GEOMNIST dataset.

The specific form of $\boldsymbol{f}_l$ is not the focus here; equation (9) allows us to prove the statement of Theorem 3.1 under this choice of $\boldsymbol{H}$. The proof of Proposition 3.3 is deferred to Appendix C.2.2, where we also provide the analytical solution of $\boldsymbol{f}_l$'s when $K = 3$.

We term this model as the logits-as-features model, or L-model for short, because it is an elaborate model specifically for logits. In Section 4, we provide concrete demonstrations of our abstract feature vector $\widetilde{X}$ as logits under L-model. We numerically simulated our LE-ODE and compared its predicted dynamics with real deep learning training dynamics. The experimental results provide strong empirical support for the validity of L-model.

### 3.3 Connection with Neural Collapse

Neural collapse is a recent phenomenological finding on the geometry of the logits learned by deep neural nets at convergence with the cross-entropy loss [38] (see an explanation of neural collapse in [16]). Simply speaking, taking our L-model as an example, with balanced training samples, this model asserts that the logit vectors from different classes at convergence form an equiangular tight frame (ETF). ETFs are the best configuration to spread $K$ unit vectors in an ambient space of $p$ dimensions. Formally, we say a set of vectors $\{\boldsymbol{s}_i\}_{i=1}^K$ form an ETF in $\mathbb{R}^p$ if they are the columns of a matrix

$$\boldsymbol{S} = \sqrt{\frac{K}{K-1}}\boldsymbol{Q}\left(\boldsymbol{I}_K - \frac{1}{K}\mathbf{1}_K\mathbf{1}_K^\top\right), \tag{10}$$

where $\boldsymbol{Q} \in \mathbb{R}^{p \times K}$, and $\boldsymbol{Q}^\top\boldsymbol{Q} = \boldsymbol{I}_K$. As a direct corollary of Proposition 3.3, when $\boldsymbol{H}$ is set according to equation (8), we find that our L-model also predicts the existence of neural collapse from the local elasticity point of view.

**Proposition 3.4** (Neural Collapse of the LE-ODE). *Under L-model and the same setup as in Theorem 3.1, if $\gamma(t) > 0$ and there exists some $T > 0$ such that $B(t) < 0$ for $t \geq T$, then $\left\{\bar{\boldsymbol{X}}^k(t)/\|\bar{\boldsymbol{X}}^k(t)\|\right\}_{k=1}^K$ forms an ETF as $t \to \infty$.*

## 4 Experiments

We perform various experiments to test our theory, where we choose *logits* as our protagonist[2].

### 4.1 Setup

**Datasets and Models.** We perform experiments on a synthesized dataset called GEOMNIST containing $K = 3$ types of geometric shapes (RECTANGLE, ELLIPSOID, and TRIANGLE) and on CIFAR-10 ([28], denoted by CIFAR) with $K \in [2, 3]$ classes. A few samples from GEOMNIST are shown in Figure 2. We vary the number of training samples per class and label pollution ratio $p_{\text{err}}$ and use variants of the ALEXNET ([29]) model. More details can be found in the Appendix.

**Training Configurations.** All models are trained for $T = 10^5$ iterations (for GEOMNIST) or $T = 3 \times 10^5$ iterations (for CIFAR) with a learning rate of 0.005 and a batch size of 1 under the softmax cross-entropy loss. Models on GEOMNIST converged with training and validation losses to zero, and those on CIFAR to validation accuracies greater than $90\%$.

**Estimation Procedures.** Each experiment is repeated for $n_{\text{trial}} = 100$ independent runs to estimate $\bar{X}(t)$. We use both the isotropic feature learning model (Section 3.2.1) and the logits-as-features model (Section 3.2.2), denoted by L-model and l-model respectively, to estimate $\alpha(t)$ and $\beta(t)$. The

---

[2]Code for reproducing our experiments is publicly available at `github.com:zjiayao/le_sde.git`.

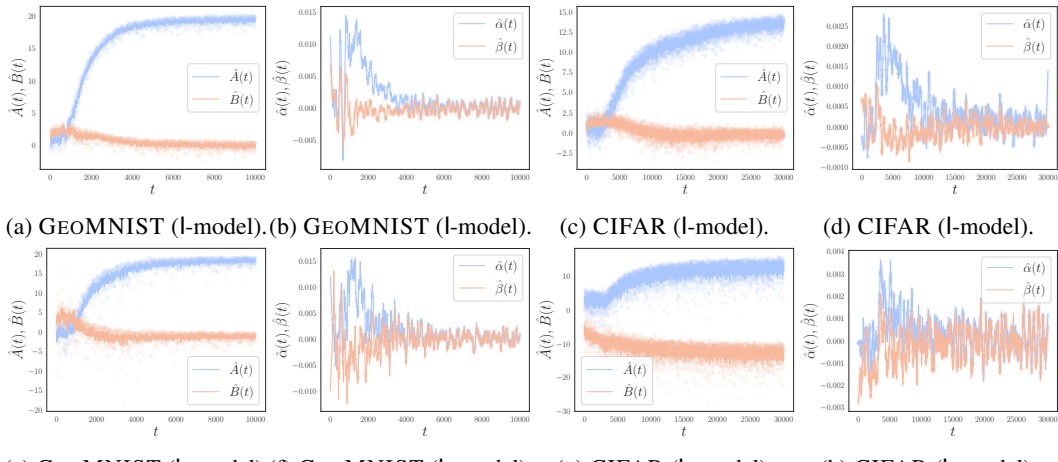

(a) GEOMNIST (l-model). (b) GEOMNIST (l-model). (c) CIFAR (l-model). (d) CIFAR (l-model).

(e) GEOMNIST (L-model). (f) GEOMNIST (L-model). (g) CIFAR (L-model). (h) CIFAR (L-model).

Figure 3: **Estimated $\widehat{A}(t)$, $\widehat{B}(t)$, $\alpha(t)$, and $\beta(t)$.** The first row was estimated using l-model and the second L-model; the first two columns are on GEOMNIST and the last two on CIFAR. The first and third rows show $\widehat{A}(t)$ and $\widehat{B}(t)$ and the other two rows $\widehat{\alpha}(t)$ and $\widehat{\beta}(t)$.

L-model is used only when $K = 3$. To estimate $\alpha(t)$ and $\beta(t)$, we first estimate $A(t)$ and $B(t)$ by

$$
\text{(l-model)} \quad \begin{cases} \widehat{A}(t) &= \text{avg avg}_k \log \left| \frac{\breve{\boldsymbol{X}}(\bar{\boldsymbol{X}}^k - \breve{\boldsymbol{X}})^{K-1}}{\boldsymbol{c}_0 \boldsymbol{c}_k^{K-1}} \right|, \quad \breve{\boldsymbol{X}}_t := \text{avg}_l \bar{\boldsymbol{X}}_t^l, \\ \widehat{B}(t) &= -\text{avg avg}_k \log \left| \frac{\boldsymbol{c}_0}{\boldsymbol{c}_k} \frac{\bar{\boldsymbol{X}}^k - \breve{\boldsymbol{X}}}{\breve{\boldsymbol{X}}} \right|, \end{cases}
$$

$$
\text{(L-model)} \quad \begin{cases} \widehat{A}(t) &= A'(t) + 2B'(t), \\ \widehat{B}(t) &= 2(B'(t) - A'(t)), \end{cases} \quad \begin{cases} A'(t) &:= \log \left| \left\langle \bar{\boldsymbol{X}}^\top \boldsymbol{v}_1 - 1 \right\rangle \right|, \\ B'(t) &:= \log \left| \left\langle \bar{\boldsymbol{X}}^\top \left( \boldsymbol{v}_2 - \frac{4}{3} \boldsymbol{v}_1 \right) \right\rangle \right|, \end{cases}
$$

(11)

where vector division is interpreted entry-wise. We write $\text{avg}_l$ for averaging over the class index, $\text{avg}$ for averaging over the coordinates, and define $\langle \boldsymbol{X} \rangle(t) := \boldsymbol{X}(t)/\boldsymbol{X}(0)$. We explain how and why to choose the vectors $\boldsymbol{v}_1$ and $\boldsymbol{v}_2$ in Appendix D.1. The main idea is to view the eigenvectors of the $Kp$-by-$Kp$ drift matrix as a concatenation of $K$ vectors of dimension $p$ and construct their linear combinations such that one or more independent components in the solution vanishes. With $A(t)$ and $B(t)$ estimated, we use the Savitzky - Golay filter to obtain $\widehat{\alpha}(t)$ and $\widehat{\beta}(t)$ through numerical differentiation. For GEOMNIST and CIFAR datasets, we choose window sizes of this filter as 191 and 551, respectively, in Figure 3, and 21 and 21, respectively, in Figure 5.

We assess the tail of $\widehat{\alpha}(t)$ and $\widehat{\beta}(t)$ by a *tail index* defined as $r_\alpha := \sup_s \{s : \lim_{t \to \infty} \alpha(t) \cdot t^s < \infty\}$ and $r_\beta$ is defined similarly. We estimate $\widehat{r}_\alpha$ by fixing an interval $[T_1, T_2]$ with $T_1 < T_2$ sufficiently large such that we may ignore terms with smaller order and have $\widehat{r}_\alpha = 1 - \text{avg}_{T_1 \le t \le T_2} \frac{\log \alpha(t)}{\log(1+t)}$, and similarly for $\widehat{r}_\beta$. We use the estimates from the last 1000 iterations for averaging in our experiments.

## 4.2 Results

**Local Elasticity in Neural Net Training.** Local elasticity manifests from our model as the heaviness of the tail of $\gamma(t) = \alpha(t) - \beta(t)$, and in Figure 3, we plot the estimations $\widehat{A}(t)$, $\widehat{B}(t)$, $\widehat{\alpha}(t)$, and $\widehat{\beta}(t)$ using both l-model and L-model. We note that (i) The estimations from the two models are visually similar, especially in the late stage of training when $t$ is large; (ii) The major difference lies in the initial stage, where the estimates from L-model behave slightly wilder. This is not surprising because of the effect of the unknown constant offset $\boldsymbol{c}_0$ in the L-model; (iii) Both $\widehat{\alpha}(t)$ and $\widehat{\beta}(t)$ behave similarly on both datasets.

**Phase Transition of Separability.** Theorem 3.1 states that separation of features under the LE-SDE takes place when $\gamma(t) = \alpha(t) - \beta(t) = \omega(1/t)$, or roughly speaking, when $r_\gamma = \min\{r_\alpha, r_\beta\} < 1$. Although we cannot directly control $\alpha(t)$ and $\beta(t)$, we can bias them by tuning the label corruption ratio $p_{\text{err}}$. When $p_{\text{err}} \approx p_{\text{err}}^* := 2/3$, we are in effect assigning labels completely at random and thus we expect a phase transition of separability should happen around $p_{\text{err}}^*$. This is indeed the story depicted in Figure 4: Figures 4a and 4b show that the validation loss and accuracy for $p \ge p_{\text{err}}$

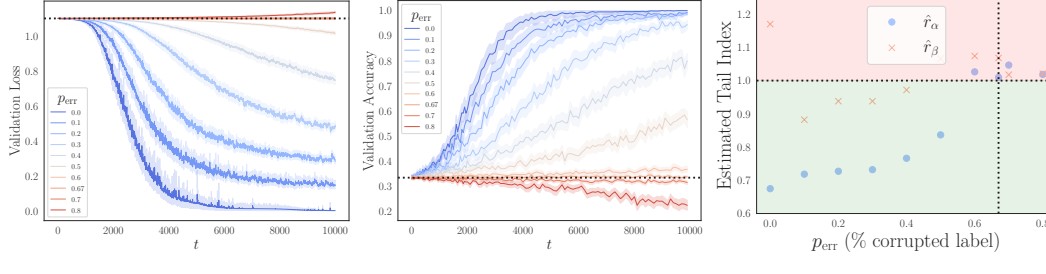

(a) Validation loss versus $p_{\text{err}}$.    (b) Validation accuracy versus $p_{\text{err}}$.    (c) Tail index versus $p_{\text{err}}$.

Figure 4: **Phase transition of separability. (a)—(b)** Validation loss and accuracy suggest separation fails for $p_{\text{err}} \geq p_{\text{err}}^* = 2/3$. The dashed line in (a) carries the value at initialization and overlaps with the case where $p_{\text{err}} = 0.6$; the dashed line in (b) is $p_{\text{err}}^* = 2/3$, when labels are assigned completely at random. **(c)** Tail indices of $\alpha(t)$ and $\beta(t)$. Note that $\gamma(t) = \alpha(t) - \beta(t)$ crosses the horizontal line $r = 1$, entering the non-separable regime (shaded in red) from the separable regime (shaded in green), around the same $p_{\text{err}}$ that cross the dashed lines in (a) and (b), as predicted by Theorem 3.1.

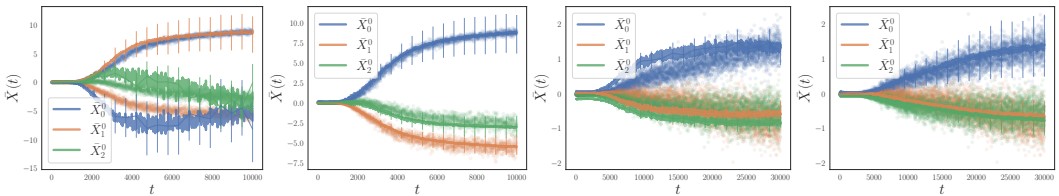

(a) GEOMNIST (l-model). (b) GEOMNIST (L-model).    (c) CIFAR (l-model).    (d) CIFAR (L-model).

Figure 5: **Simulated LE-ODE solutions versus genuine dynamics.** We use $\widehat{\alpha}(t)$ and $\widehat{\beta}(t)$ estimated from l-model ((a) and (c)) or L-model, ((b) and (d)) and numerically simulate the solution under the L-model. The results were overlaid with true dynamics from neural nets. We note L-model in general imitated true dynamics reasonably well.

are not increasing over time and in Figure 4c we observe the minimum tail index of $\alpha(t)$ and $\beta(t)$ crosses 1 from below around $p_{\text{err}} = 0.6$, entering the non-separable regime (shaded in red) from the separable regime (shaded in green), given in Theorem 3.1.

**Simulating DNN Dynamics via LE-ODE.**   Having estimated $\alpha(t)$ and $\beta(t)$, it is natural to ask, to what capacity can our LE-ODE models recover the real dynamics of deep neural nets? We use the forward Euler method to simulate the L-model using $\widehat{\alpha}(t)$ and $\widehat{\beta}(t)$ estimated from either l-model or L-model, We choose $K = 3$ and show in Figure 5 the simulated solution (solid line) with error bars depicting one standard deviation over 500 independent runs, overlaying on the real dynamics from DNNs in the background (shaded transparent markers). As the moving average may reduce the magnitudes of $\alpha(t)$ and $\beta(t)$, we rescale the simulated paths such that its first coordinate is approximately equal to the ground truth at convergence. Note that estimations from L-model can faithfully recover the genuine dynamics from neural nets, whereas those from l-model fail, notably in Figure 5a, where the simulated paths preserve the relative magnitude but fail to identify the correct order of three logits.

## 5 Discussion and Future Works

In this study, we introduce LE-SDE/ODE models that draw inspiration from the local elasticity phenomenon. Conditions for sharp phase transition of separability of features are derived. We also show that once the elasticity strengths $\alpha(t)$ and $\beta(t)$ are well estimated, our model can faithfully simulate the dynamics of neural nets.  We outline a few interesting problems for future research while leaving the details in the Appendix. **(i) General LE Matrix.** A similar result as in Theorem 3.1 may be expected for symmetric but no necessarily semi-definite LE matrices $\boldsymbol{E}(t)$. **(ii) Mini-batch Training, Imbalanced Datasets, and Label Corruptions.** Generalizing the drift matrix to $\boldsymbol{M}_t = (\boldsymbol{E}_t \otimes \boldsymbol{P}) \circ \boldsymbol{H}/K$ for a $K$-by-$K$ doubly stochastic matrix $\boldsymbol{P}$ can be used to model various sampling effects. **(iii) Beyond L-model for Imitating Genuine Dynamics of DNNs.** Although the L-model is shown to be able to mimic the real dynamics reasonably well, we postulate that a more precise model might have its $(i, j)$-th block encode the other directions other than $\boldsymbol{d}_j$.

## Acknowledgements

This work was supported in part by NSF through CCF-1934876, an Alfred Sloan Research Fellowship, the Wharton Dean's Research Fund, and ONR Contract N00014-19-1-2620. We would like to thank Dan Roth and the Cognitive Computation Group at the University of Pennsylvania for stimulating discussions and for providing computational resources.

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
