## A  Derivation of Continuous Dynamics of Binary Case

We will now derive continuous dynamics (2) in the main paper. Let $\mathbb{1}_m = 1$ if class 1 is selected at iteration $m$ and $\mathbb{1}_m = 0$ otherwise. Chaining the dynamic (2) $r$ times, we have

$$X_i^1(m-1+r) - X_i^1(m-1)$$
$$= \sum_{q=1}^{r} \left( \mathbb{1}_{m+q-1} \alpha h X_{I_{m+q-1}}^1 (m+q-2) + (1 - \mathbb{1}_{m+q-1}) \beta h X_{I_{m+q-1}}^2 (m+q-2) + \zeta_{m+q-2}^i \right).$$

When $m \gg r$, we have approximately

$$\sum_{q=1}^{r} \left( \mathbb{1}_{m+q-1} \alpha h X_{I_{m+q-1}}^1 (m+q-2) + (1 - \mathbb{1}_{m+q-1}) \beta h X_{I_{m+q-1}}^2 (m+q-2) \right)$$

$$\approx \sum_{q=1}^{r} \left( \mathbb{1}_{m+q-1} \alpha h X_{I_{m+q-1}}^1 (m-1) + (1 - \mathbb{1}_{m+q-1}) \beta h X_{I_{m+q-1}}^2 (m-1) \right)$$

$$= \sum_{q=1}^{r} \mathbb{1}_{m+q-1} \alpha h X_{I_{m+q-1}}^1 (m-1) + \sum_{q=1}^{r} (1 - \mathbb{1}_{m+q-1}) \beta h X_{I_{m+q-1}}^2 (m-1)$$

$$\approx r \cdot \frac{1}{2} \alpha h \frac{\sum_{i=1}^{n} X_i^1(m-1)}{n} + r \cdot \frac{1}{2} \beta h \frac{\sum_{j=1}^{n} X_j^2(m-1)}{n}$$

$$\approx \frac{\alpha h r}{2} \bar{X}(m-1) + \frac{\beta h r}{2} \bar{Y}(m-1).$$

Next, observe that

$$\sum_{q=1}^{r} \zeta_{m+q-2}^i \sim \mathcal{N}\left(0, \sigma^2 rh\right),$$

hence taken together, the calculations above give

$$X_i^1(m-1+r) - X_i^1(m-1) \approx \frac{\alpha h r}{2} \bar{X}(m-1) + \frac{\beta h r}{2} \bar{Y}(m-1) + \mathcal{N}\left(0, \sigma^2 rh\right).$$

Writing $\Delta t = rh$ and $t = (m-1)h$, we have

$$X_i^1(t + \Delta t) - X_i^1(t) \approx \frac{\alpha}{2} \bar{X}(t) \Delta t + \frac{\beta}{2} \bar{Y}(t) \Delta t + \sigma \mathcal{N}(0, \Delta t),$$

which is the discretization of

$$\mathrm{d}X_i^1(t) = \left( \frac{\alpha}{2} \bar{X}(t) + \frac{\beta}{2} \bar{Y}(t) \right) \mathrm{d}t + \sigma \mathrm{d}W^i(t).$$

Likewise, we can obtain the dynamics of $X_j^2$ similarly. We will next prove the separation theorem in binary classification, Theorem 2.1.

**Theorem 2.1** (Separation in Binary Classification). *Given the feature vectors $X_i^1(t)$, $X_j^2(t)$ for $i, j \in [n]$, as $t \to \infty$ and large $n$,*

*1. if $\alpha > \beta$, they are asymptotically separable with probability tending to one,*

*2. if $\alpha \leq \beta$, they are asymptotically separable with probability tending to zero.*

*Proof of Theorem 2.1.* Note that whenever $\alpha \leq \beta$, we have $\frac{c_1 - c_2}{2} e^{\frac{\alpha - \beta}{2} t} \to 0$ as $t \to \infty$, thus $X_i^1$ and $X_j^2$ are interspersed and separation happens with probability tending to zero. This also aligns with our intuition that the intra-class effect should be stronger than its inter-class counterpart.

On the other hand, when $\alpha > \beta$, ignoring a null set we may assume $c_1 > c_2$ without loss of generality. To see this, note that by definition $c_k = \mathbb{E}_{\text{data}}[X^k(0)|\theta(0) = \theta_0]$ for $k \in \{1, 2\}$ where $\theta(0)$ is all parameters of the neural net at initialization and $\theta_0$ is a particular realization given the initialization scheme. Here the expectation is taken with respect to the data distribution, and when we ignore a null

set of neural net with respect to the probability measure induced by the initialization scheme, $c_1 \neq c_2$ holds. In other words, this statement can be interpreted as "the expected feature at initialization from the first class is different from that from the second class for a neural net, except possibly on a null set in the space of neural nets with respect to the probability measure induced by the parameter initialization scheme." It suffices to show that

$$\min_{1 \leq i \leq n} \frac{c_1 - c_2}{2} e^{\frac{\alpha-\beta}{2}t} + \frac{c_1 + c_2}{2} e^{\frac{\alpha+\beta}{2}t} - c_x + X_i^1(0) + \sigma W_i^1(t)$$
$$> \max_{1 \leq j \leq n} -\frac{c_1 - c_2}{2} e^{\frac{\alpha-\beta}{2}t} + \frac{c_1 + c_2}{2} e^{\frac{\alpha+\beta}{2}t} - c_2 + X_j^2(0) + \sigma W_j^2(t),$$

which is equivalent to

$$(c_1 - c_2) e^{\frac{\alpha-\beta}{2}t} - c_1 + \min_{1 \leq i \leq n} X_i^1(0) + \sigma W_i^1(t) > -c_2 + \max_{1 \leq j \leq n} X_j^2(0) + \sigma W_j^2(t).$$

But the above display happens with probability tending to one provided $\alpha > \beta$, thus completing the proof. $\qquad \square$

## B   Further Details on Drift Modeling

### B.1   Dynamics of the Logits-as-Features Model

This section provides more details on why the construction of $\boldsymbol{H}$ in Section 3.2.2 is probably a good choice for modeling the dynamics of logits in deep neural nets. Given $K \geq 2$ classes with $n$ training examples per class, the feature vectors are the logits $\boldsymbol{X}^k(m) \in \mathbb{R}^K$ for all $k \in [K]$, where $m$ is the iteration number. When the neural net is trained under the softmax cross-entropy loss $L$, at the $m$-th iteration, if the $J_m$-th sample from the $L_m$-th class is sampled, the dynamics of the logits $\boldsymbol{X}_i^k$ should be governed by

$$\boldsymbol{X}_i^k(m) - \boldsymbol{X}_i^k(m-1) \approx h \left[ \frac{\partial \boldsymbol{X}_i^k(m-1)}{\partial \boldsymbol{w}} \frac{\partial \boldsymbol{X}_{J_m}^{L_m}}{\partial \boldsymbol{w}}^{\top} \left( \boldsymbol{e}_{L_m} - \mathrm{softmax}(\boldsymbol{X}_{J_m}^{L_m}) \right) \right]. \qquad (B.1)$$

The derivation of equation (B.1) is a straightforward computation from the Taylor approximation

$$\boldsymbol{X}_i^k(m) - \boldsymbol{X}_i^k(m-1) \approx \nabla_{\boldsymbol{w}} \boldsymbol{X}_i^k(m-1) \Delta \boldsymbol{w}(m-1), \qquad (B.2)$$

where we observe that

$$\Delta \boldsymbol{w}(m-1) = -h \frac{\partial L(\boldsymbol{w}(m-1))}{\partial \boldsymbol{w}} = h \frac{\partial \boldsymbol{X}_{J_m}^{L_m}}{\partial \boldsymbol{w}}^{\top} \left( \boldsymbol{e}_{L_m} - \mathrm{softmax}(\boldsymbol{X}_{J_m}^{L_m}) \right). \qquad (B.3)$$

However, as equation (B.1) is highly non-linear, $J_m$ and $L_m$ are random, and the Gram matrix $\frac{\partial \boldsymbol{X}_i^k(m-1)}{\partial \boldsymbol{w}} \frac{\partial \boldsymbol{X}_{J_m}^{L_m}}{\partial \boldsymbol{w}}^{\top}$ is also time-dependent, direct analyses and simulation of the exact dynamics are difficult. Note that this Gram matrix is also the key element in the NTK literature [22, 15, 2], which is treated as roughly fixed during the lazy training process.

Recall that we consider the per-class mean of logits, $\bar{\boldsymbol{X}}^k = \mathbb{E}\widetilde{\boldsymbol{X}}^k$, where $\widetilde{\boldsymbol{X}}^k$ is a random sample, for each class $k$. In the limit of $h \to 0$ with the identification of time $t = mh$, the above modeling allows us to re-write (B.1) in the per-class mean $\bar{\boldsymbol{X}}^k$ and logits from a generic sample $\widetilde{\boldsymbol{X}}^k$ in this continuous limit as

$$\mathrm{d}\widetilde{\boldsymbol{X}}_t^k \approx \mathbb{E}_{L \sim \mathrm{Unif}([K])} \left[ \mathbb{E}_{\widetilde{\boldsymbol{X}} \sim \mathcal{D}_t^L} \left[ \frac{\partial \boldsymbol{X}_i^k(m-1)}{\partial \boldsymbol{w}} \frac{\partial \widetilde{\boldsymbol{X}}}{\partial \boldsymbol{w}}^{\top} \left( \boldsymbol{e}_L - \mathrm{softmax}(\widetilde{\boldsymbol{X}}) \right) \right] \right] \mathrm{d}t + \boldsymbol{\Sigma}_t^{\frac{1}{2}} \mathrm{d}\boldsymbol{W}_t,$$

$$\approx \frac{1}{K} \sum_L \left( \left[ \mathbb{E}_{\widetilde{\boldsymbol{X}}' \sim \mathcal{D}_t^k, \widetilde{\boldsymbol{X}} \sim \mathcal{D}_t^L} \frac{\partial \widetilde{\boldsymbol{X}}'}{\partial \boldsymbol{w}} \frac{\partial \widetilde{\boldsymbol{X}}}{\partial \boldsymbol{w}}^{\top} \right] \left( \boldsymbol{e}_L - \mathrm{softmax}(\bar{\boldsymbol{X}}_t^L) \right) \right) \mathrm{d}t + \boldsymbol{\Sigma}_t^{\frac{1}{2}} \mathrm{d}W_t, \qquad (B.4)$$

$$= \frac{1}{K} \sum_L \left( \Theta_{k,L} \left( \boldsymbol{e}_L - \mathrm{softmax}(\bar{\boldsymbol{X}}_t^L) \right) \right) \mathrm{d}t + \boldsymbol{\Sigma}_t^{\frac{1}{2}} \mathrm{d}\boldsymbol{W}_t.$$

The presence of expectation over non-linearity posed considerate difficulties of using equation (B.4) to analyze neural nets; however, it motivates our LE-SDE model as a linear approximation to it while

explicitly encodes the local elasticity into the dynamics. More precisely, given a sample instance $z^k$ from the $k$-th class, in a well-trained neural network, the probability vector associated with $z^k$, i.e., the output from the softmax applied to its logits, should have its $k$-th entry as the largest and the other entries as roughly equal. Hence $e_k - \text{softmax}(\bar{X}_t^k)$ should be roughly in the direction of $e_k - \frac{1}{K}\mathbf{1}$. To approximate this limit by the limit from a linear map, we may use our choice of $\boldsymbol{H}_{kl} = \bar{\boldsymbol{H}}^l := \frac{d_l d_l^\top}{\|d_l\|^2}$ for all $k, l \in [K]$, such that in the limit of $t \to \infty$, we expect

$$e_k - \text{softmax}\left(\bar{\boldsymbol{X}}_t^k\right) \approx C\bar{\boldsymbol{H}}^k \bar{\boldsymbol{X}}_t^k, \tag{B.5}$$

for any $k \in [K]$ and some positive constant $C$. This is the intuition behind our L-model. To summarize, in our model, the local elasticity matrix $E$ describes the effect $\Theta_{k,L}$, and $\boldsymbol{H}_{k,l}$ transforms the mean logit $\bar{\boldsymbol{X}}_t^L$ to the direction governed by the supervision.

## B.2 Discussions on Linearization

We provide more details of the rationale behind our choice of $M(t)X(t)$, a seemingly linear term, as the surrogate for the non-linear drift in the dynamics given in equation B.4. Our following argument can be extended to any post-activation features. Writing equation B.4 in terms of $X(t) \in \mathbb{R}^{Kp}$ (recall in this case we have $p = K$), the concatenation of $K$ per-class feature vectors $X^k(t) \in \mathbb{R}^p$ for $k \in [K]$, and denoting by $\sigma : \mathbb{R}^K \to \mathbb{R}^K$ the softmax function for simplicity, we can express the drift term as

$$F(\widetilde{\boldsymbol{X}}(t), t) := \Theta(t)\left(\left[e_k - \sigma(\widetilde{\boldsymbol{X}}^k(t))\right]_{k=1}^K\right), \tag{B.6}$$

where we wrote $[\cdot]$ for vector concatenation and $\Theta(t) \in \mathbb{R}^{(Kp)\times(Kp)}$ for the Gram matrix. A commonly used linearization scheme in SDE for non-linear drifts by the filtering community (cf. Chapter 9.1 of [46]) is to linearize $F$ for each $t$ at the mean $\varphi(t) = (\varphi_k(t))_{k=1}^K \equiv \bar{X}(t) := \mathbb{E}_B X(t)$ where the expectation is taken with respect to the diffusion. Concretely, we have

$$F(\widetilde{\boldsymbol{X}}(t), t) \approx \widetilde{F}(\widetilde{\boldsymbol{X}}(t), t) := F(\varphi(t), t) + \nabla_X F(\varphi(t), t)\left(\widetilde{\boldsymbol{X}}(t) - \varphi(t)\right), \tag{B.7}$$

where $\nabla_X F$ denotes the Jacobian of $F$ with respect to the spacial variable $\widetilde{\boldsymbol{X}}$. For notation completeness, we introduce

$$p = (p_k)_{k=1}^K \in \mathbb{R}^{Kp}, \quad p_k := \sigma(\widetilde{\boldsymbol{X}}^k(t)) \in \mathbb{R}^p, \quad k \in [K], \tag{B.8}$$

and similarly

$$\bar{p} = (\bar{p}_k)_{k=1}^K \in \mathbb{R}^{Kp}, \quad \bar{p}_k := \sigma(\bar{\boldsymbol{X}}^k(t)) \in \mathbb{R}^p, \quad k \in [K]. \tag{B.9}$$

We write the per-class Jacobians as

$$J_{kk} = J_k := \text{diag}(\bar{p}_k) - \bar{p}_k \bar{p}_k^T. \tag{B.10}$$

Clearly, the Jacobian $\nabla F(\varphi, t) = J(t)$ can be written as a block-diagonal matrix $J(t) = (J_{kk})_{k=1}^K$ consisting of per-class Jacobians. Now continuing linearization, we can write

$$\widetilde{F}(\widetilde{\boldsymbol{X}}(t), t) = \Theta(t)\left([e_k - \bar{p}_k]_k + J(t)(\widetilde{\boldsymbol{X}}(t) - \varphi(t))\right)$$
$$= \Theta(t)\left(J(t)X(t) + [e_k - \bar{p}_k + J_k\varphi_k(t)]_k\right). \tag{B.11}$$

Define $\Psi : \mathbb{R}^{Kp} \to \mathbb{R}^{Kp} : z \mapsto [e_k - \sigma(z_k)]_k$ and write $\Psi_k : \mathbb{R}^p \to \mathbb{R}^p$ to be the $k$-th component of $\Psi$, using Taylor's theorem to expand $\Psi(z)$ around $\varphi(t)$ for each $t$, we have

$$\Psi = \Psi(\varphi) + J(t)\varphi - J(\varphi)z + o\left(\|z - \varphi\|\right), \tag{B.12}$$

or

$$\Psi(\varphi) + J(t)\varphi = \Psi(z) + J(\varphi)z + o\left(\|z - \varphi\|\right). \tag{B.13}$$

This implies that

$$\widetilde{F} = \Theta(t)J(t)\widetilde{\boldsymbol{X}}(t) + \Theta(t)R(t), \quad R(t; z) := \Psi(z) + J(t)z + o\left(\|z - \varphi(t)\|\right), \tag{B.14}$$

where $R(t; z)$ is the residue that depends on the choice of $z$ around which $\Psi(\varphi)$ is expanded. Note that the first term is a time-varying linear term in $X(t)$. As long as the residue term is negligible, we get exactly the time-varying linear map $M(t)$ in the LE-SDE model.

By choosing different $z$'s in equation B.14, we can focus on different stages of the real dynamics using the LE-SDE. Two particular choices are of great interest so as to make the residue term vanishing:

- **Around initialization.** Let $z = u := c \cdot [\mathbf{1}_K/K]_{k=1}^K$ be a scaling of vectors of ones where $c$ is some fixed constant. Then each of the $K$ components of $\sigma(u)$ assigns approximately the same probability $(1/K)$ for every label. Furthermore, $u \in \operatorname{Ker} J(t)$ for all $t$ hence the residue $R(t; u) = \Psi(u) + o\,(\|z - \varphi(t)\|)$ is a constant vector (which is colinear with $\boldsymbol{d} = (\boldsymbol{d}_j)_{j=1}^K$ defined by $\boldsymbol{d}_j = \boldsymbol{e}_j - \mathbf{1}_K/K$ when we discussed the L-model). Note that this approximation works best when the model has not learned much about the data (in the "first stage" as we call it) since $\|u - \varphi(t)\|$ is small in this regime.

- **Around convergence.** Given that the model converges, $\varphi_\infty := \varphi(\infty)$ is finite. Let $z = \varphi_\infty$, under the effective training assumption, $\|\Psi(\varphi_\infty)\| \approx 0$ by construction. Hence the residue $R(t; \varphi_\infty) = J(t)\varphi_\infty + o\,(\|\varphi(t) - \varphi_\infty\|)$. Here the $o(\cdot)$ term converges to 0 as training progresses, leaving us a term that is asymptotically equivalent to $v = (v_k)_{k=1}^K := J(\varphi_\infty)\varphi_\infty \in \mathbb{R}^{K^2}$, where $v_k = [(z_{k,i} - \sum_{j=1}^K p_{k,j}z_{k,j})p_i]_{i=1}^K \in \mathbb{R}^K$. Again, under the effective training assumption $z_k$ has its $k$-th entry $z_{k,k}$ the largest, and $p_k$ has its $k$-th entry close to 1 while the others to zero. Thus $v_{k,i} \approx 0_K$. We see that in this regime, the approximation $\Theta(t)J(t)X(t)$ is only off by a residue $o\,(\|\varphi(t) - \varphi_\infty\|)$ that eventually vanishes.

As discussed above, we choose to linearize the drift at convergence instead of around initialization in the L-model given by equation B.5 such that the residue vanishes under the effective training assumption.

## C  Miscellaneous Proofs

### C.1  Various Definitions of Separability

We first give formal definitions of separability which generalize Theorem 2.1 in Section 2.2 to our $K$-class, $p$-dimensional feature setting. In the beginning of Section 3.1, we state the definition of separability in natural language, and we formalize the definition therein as follows. It is the most natural definition in terms of linear separation by a hyperplane.

**Definition C.1** (Pairwise Separation). *We say the feature vectors $\left\{(\boldsymbol{X}_i^k)_{i\in[n]}\right\}_{k\in[K]}$ is pairwisely separable at time $t$ if for each pair of $1 \leq k < l \leq K$, there exists a direction $\boldsymbol{\nu}_{k,l}$ such that*

$$\min_i \left\langle \boldsymbol{\nu}_{k,l}, \boldsymbol{X}_i^k(t) \right\rangle > \max_j \left\langle \boldsymbol{\nu}_{k,l}, \boldsymbol{X}_j^l(t) \right\rangle. \tag{C.1}$$

In the Theorem 3.1, we claim that when $\gamma(t) = \omega(1/t)$, the pairwise separation (Definition C.1) happens with probability tending to 1 as $t \to \infty$. This is a notion of asymptotic separability stated in Theorem 3.1, yet it is a weaker notion of separation in the following sense: the hyperplane may depend on the classes $l, k$, and the hyperplane may depend on time $t$. We state the following asymptotic separable definitions of increasingly stronger guarantees, and will remark on how to obtain them in our proof of Theorem 3.1.

The direct application of Definition C.1 gives us the following (weakest) definition.

**Definition C.2** (Asymptotic Pairwise Separation). *We say the feature vectors $\left\{(\boldsymbol{X}_i^k)_{i\in[n]}\right\}_{k\in[K]}$ are asymptotically pairwisely separable if for each pair of $1 \leq k < l \leq K$, there exist directions $\boldsymbol{\nu}_{k,l}(t)$ such that*

$$\mathbb{P}\left(\min_i \left\langle \boldsymbol{\nu}_{k,l}(t), \boldsymbol{X}_i^k(t) \right\rangle > \max_j \left\langle \boldsymbol{\nu}_{k,l}(t), \boldsymbol{X}_j^l(t) \right\rangle\right) \to 1. \tag{C.2}$$

Requiring all the classes to be separable with the same hyperplane give us the following universal separation.

**Definition C.3** (Asymptotic Universal Separation). *We say the feature vectors $\left\{(\boldsymbol{X}_i^k)_{i\in[n]}\right\}_{k\in[K]}$ are asymptotically universally separable if it is asymptotically pairwisely separable, and there exists $\nu$, such that either $\boldsymbol{\nu} = \boldsymbol{\nu}_{k,l}$ or $\boldsymbol{\nu} = -\boldsymbol{\nu}_{k,l}$, for all $1 \leq k < l \leq K$ in (C.2).*

Note that we allow the universal direction differs in sign for different pair of classes. This is because we do not require separating the $k$ class in any specific order.

We note that the above definitions are in the sense of separation "in probability", which only asserts separation at an arbitrarily fixed large time $t$. Specifically, it does not guarantee the existence of a fixed direction that can always separate a pair of classes for all sufficiently large $t$. We now state the almost sure definition of asymptotic pairwise separation, which guarantees the same fixed direction separates a pair of classes for all large enough $t$. And Theorem 3.1 also holds for such definition.

**Definition C.4** (Uniform Asymptotic Pairwise Separation)**.** *We say the feature vectors* $\left\{ (\boldsymbol{X}_i^k)_{i\in[n]} \right\}_{k\in[K]}$ *are uniformly asymptotically pairwisely separable if for each pair of* $1 \le k < l \le K$*, there exists a direction* $\boldsymbol{\nu}_{k,l}$ *such that*

$$\liminf_{T\to\infty} \min_{k,l} \mathbb{P}\left( \min_i \left\langle \boldsymbol{\nu}_{k,l}, \boldsymbol{X}_i^k(t) \right\rangle > \max_j \left\langle \boldsymbol{\nu}_{k,l}, \boldsymbol{X}_j^l(t) \right\rangle, \text{ for all } t > T \right) = 1. \tag{C.3}$$

We define uniform asymptotic universal separation the same way as its non-uniform counterpart.

**Definition C.5** (Uniform Asymptotic Universal Separation)**.** *We say the feature vectors* $\left\{ (\boldsymbol{X}_i^k)_{i\in[n]} \right\}_{k\in[K]}$ *are uniformly asymptotically universally separable if it is uniformly asymptotically pairwisely separable, and there exists* $\nu$*, such that either* $\boldsymbol{\nu} = \boldsymbol{\nu}_{k,l}$ *or* $\boldsymbol{\nu} = -\boldsymbol{\nu}_{k,l}$*, for all* $1 \le k < l \le K$ *in* (C.3)*.*

The asymptotic inseparability definitions can be similarly stated.

## C.2 Proofs in Section 3

Before proving the main result, Theorem 3.1, it is convenient to first prove Propositions 3.2 and 3.3.

### C.2.1 Proof of Proposition 3.2

**Proposition 3.2** (I-model)**.** *Let* $\boldsymbol{H}_{k,l} = \boldsymbol{I}_p$*, then the solution to the LE-ODE* (6) *is given by*

$$\bar{\boldsymbol{X}}(t) = \boldsymbol{c} \mathrm{e}^{\frac{1}{K}A(t) - \frac{1}{K}B(t)} + (\boldsymbol{1}_K \otimes \boldsymbol{c}_0) \, \mathrm{e}^{\frac{1}{K}A(t) + \frac{K-1}{K}B(t)}, \tag{7}$$

*where* $\boldsymbol{c} = (\boldsymbol{c}_k)_{k=1}^K \in \mathbb{R}^{Kp}$ *and* $\boldsymbol{c}_0 \in \mathbb{R}^p$ *are constants with* $\sum_{k=1}^K \boldsymbol{c}_k = \boldsymbol{0} \in \mathbb{R}^p$ *and* $\bar{\boldsymbol{X}}(0) = \boldsymbol{c} + \boldsymbol{c}_0$*.*

*Proof of Proposition 3.2.* When $\alpha(t) = \alpha$ and $\beta(t) = \beta$ are constants, $\boldsymbol{E}$ has two distinct eigenvalues: $\lambda_0 = (\alpha - \beta)/K$ with multiplicity $K - 1$ and $\lambda_1 = (\alpha + (K-1)\beta)/K$ with multiplicity one, and the eigendecomposition $E = \boldsymbol{Q}\boldsymbol{\Lambda}\boldsymbol{Q}^{-1}$ is given by

$$\boldsymbol{\Lambda} = \mathrm{diag}(\lambda_1, \lambda_0, \dots, \lambda_0) \in \mathbb{R}^{K\times K}, \quad \boldsymbol{Q} = \begin{bmatrix} 1 & 1 & 1 & \cdots & 1 \\ 1 & -1 & 0 & \cdots & 0 \\ 1 & 0 & -1 & \cdots & 0 \\ \vdots & & & & \vdots \\ 1 & 0 & 0 & \cdots & -1 \end{bmatrix}. \tag{C.4}$$

Hence the eigenpairs of $\boldsymbol{E} \otimes \boldsymbol{I}_p$ are $(\boldsymbol{e}_1 - \boldsymbol{e}_k) \otimes \boldsymbol{e}_j$ for $1 < k \le K$ and $1 \le j \le p$ with eigenvalue $\lambda_0$ and $(\boldsymbol{1}_K \otimes \boldsymbol{e}_j)$ for $1 \le j \le p$ with eigenvalue $\lambda_1$. Reparameterizing the solution such that the initial values are $\bar{\boldsymbol{X}}^k(0) = \boldsymbol{c}_0 + \boldsymbol{c}_k$ for all $k \in [K]$, we have

$$\bar{\boldsymbol{X}}_t = \boldsymbol{c} \mathrm{e}^{\lambda_0 t} + (\boldsymbol{1}_K \otimes \boldsymbol{c}_0) \, \mathrm{e}^{\lambda_1 t}, \tag{C.5}$$

for $\boldsymbol{c}_0 \in \mathbb{R}^p$ and $\boldsymbol{c} = (\boldsymbol{c}_k)_{k=1}^K$ with $\sum_{k=1}^K \boldsymbol{c}_k = \boldsymbol{0}$. Based on this solution, it is not difficult to show that the general solution with varying $\alpha(t)$ and $\beta(t)$ is given by

$$\bar{\boldsymbol{X}}_t = \boldsymbol{c} \mathrm{e}^{\frac{1}{K}A(t) - \frac{1}{K}B(t)} + (\boldsymbol{1}_K \otimes \boldsymbol{c}_0) \, \mathrm{e}^{\frac{1}{K}A(t) + \frac{K-1}{K}B(t)}, \tag{C.6}$$

where $\sum_{k=1}^K \boldsymbol{c}_k = \boldsymbol{0}$. $\qquad\square$

### C.2.2 Proof of Proposition 3.3

**Proposition 3.3** (L-model). *Let $H$ be the same as in equation* (8), *then the solution to the LE-ODE* (6) *is given by*

$$\bar{X}(t) = c_0 + C_1 d e^{\frac{1}{K} A(t) - \frac{1}{K} B(t)} + \left( \sum_{l=1}^{K-1} C_{2l} f_l \right) e^{\frac{1}{K} A(t) + \frac{1}{K(K-1)} B(t)}, \qquad (9)$$

*where $f_l$'s are fixed vectors in $\mathbb{R}^{K^2}$, $c_0 \in \mathbb{R}^{K^2}$ is a constant vector with $K(K-1)$ degrees of freedom, and $C_1, C_{2l} \in \mathbb{R}$, for $l \in [K-1]$ are constants.*

*Proof of Proposition 3.3.* Recall that the $(k, l)$-th block of the $H$ matrix is $\bar{H}^l := d_l d_l^\top / d_l^\top d_l$. When $\alpha(t) = \alpha$ and $\beta(t) = \beta$ are constants, we immediately find the matrix $(E \otimes I_K) \circ H$ has an eigenvalue of $\alpha - \beta$ with multiplicity one whose eigenvector is $d$, and an eigenvalue of $0$ with multiplicity $K(K-1)$ – since the null spaces of $H_{kl}$ all have dimension $K-1$. We also find it has another eigenvalue $\alpha + \beta/(K-1)$ with multiplicity $K-1$. Therefore, we have the general solution (9) by noting there is an additional $1/K$ factor in the definition of $M_t$.

Specifically, when $K = 3$, we can explicitly write out the eigenvectors of $(E \otimes I_K) \circ H$ as follows. There is one eigenvector $d$ corresponding to the eigenvalue $\alpha - \beta$. The eigenvectors corresponding to the eigenvalue $0$ take the form of

$$[w_1 + w_2, 2w_1, 2w_2, 2w_3 - w_4, w_3, w_4, -w_5 - 2, w_5, w_6]^\top, \qquad (C.7)$$

where $\{w_i\}_{i=1}^6$ are free parameters; the eigenvectors corresponding to the eigenvalue $(\alpha + \beta/(K+1))/K$ take the form of

$$[-\xi_1 w_1 + \xi_2, \xi_2 w_1 - \xi_5, \xi_3 w_1 - \xi_4, -\xi_2 w_1 + \xi_4, \xi_1 w_1 - \xi_3, -\xi_3 w_1 + \xi_6, 1 - w_1, w_1, w_2]^\top, \qquad (C.8)$$

where $w_1$ and $w_2$ are free parameters and

$$\xi_1 = \frac{2\alpha + \beta}{3\beta}, \quad \xi_2 = \frac{\alpha + 2\beta}{3\beta}, \quad \xi_3 = \frac{\alpha - \beta}{3\beta},$$

$$\xi_4 = \frac{\alpha^2 + \alpha\beta + 7\beta^2}{6\alpha\beta + 3\beta^2}, \quad \xi_5 = \frac{\alpha^2 + 4\alpha\beta - 5\beta^2}{6\alpha\beta + 3\beta^2}, \quad \xi_6 = \frac{\alpha^2 - 2\alpha\beta - 8\beta^2}{6\alpha\beta + 3\beta^2}. \qquad (C.9)$$

$\square$

### C.2.3 Proof of Theorem 3.1

We are now ready to prove Theorem 3.1. We first recall the theorem statement.

**Theorem 3.1** (Separation of LE-SDE). *Under our working assumptions in Section 2.1, and in the case of local elasticity (i.e., $\gamma(t) > 0$), assume $H = (H_{ij})_{ij}$ is positive semi-definite (PSD) with positive diagonal entries. As $t \to \infty$, we have[3]:*

1. *if $\gamma(t) = \omega(1/t)$, the features are separable with probability tending to 1;*

2. *if $\gamma(t) = o(1/t)$, and the number of per-class-feature $n$ tending to $\infty$ at an arbitrarily slow rate, the features are asymptotically pairwise separable with probability $0$.*

Recall that in the main text, we only consider the most natural (pairwise) separation Definition C.1, which corresponds to Definition C.2. After the prove of this theorem, We will also remark how to extend this result to other stronger definitions, especially Definition C.5.

*Proof of Theorem 3.1.* Let $H \in \mathbb{R}^{Kp \times Kp}$ be a symmetric positive semi-definite (SPD) matrix with positive diagonal entries $(d_i)_{i=1}^{Kp}$, recall that we denote by $H_{ij} \in \mathbb{R}^{p \times p}$ the $(i, j)$-th block of $H$ for $i, j \in [K]$. We write $M_t = \frac{1}{K} (E_t \otimes I_K) \circ H$, where $(A \circ B)_{ij} = A_{ij} B_{ij}$ is the Hadamard product between two matrices of the same size.

First, assume $\alpha(t) = \alpha$ and $\beta(t) = \beta$ are constants, then from Proposition 3.2, we know $E \otimes I_p$ has an eigenvalue $\lambda_0 = (\alpha - \beta)/K$ with multiplicity $p(K-1)$ and $\lambda_1 = (\alpha + (K-1)\beta)/K$ with

---

[3]Here, $\gamma(t) = \omega(1/t)$ stands for $\gamma(t) \gg 1/t$ as $t \to \infty$. For example, $1/t^{0.5} = \omega(1/t)$ and $(t \ln t)^{-1} = o(1/t)$ as $t \to \infty$.

multiplicity $p$. Writing $\lambda_{\min}$ and $\lambda_{\max}$ as the minimum and the maximum of $\{\lambda_0, \lambda_1\}$ respectively, by Schur's theorem (Theorem 9.J.2, [37]), we have

$$\lambda_{\min} \min_{1 \le i \le Kp} d_i \le \lambda_i(\boldsymbol{M}_t) \le \lambda_{\max} \max_{1 \le i \le Kp} d_i, \tag{C.10}$$

where we write $\mu_i := \lambda_i(\boldsymbol{M}_t)$ as the $i$-th largest eigenvalue of $\boldsymbol{M}_t$, and $\boldsymbol{u}_i \in \mathbb{R}^{Kp}$ the corresponding eigenvector and recall that in this case $\boldsymbol{M}_t = \frac{1}{K}(\boldsymbol{E} \otimes \boldsymbol{I}_p) \circ \boldsymbol{H}$ does not depend on time. We will denote by $\boldsymbol{u}_i^k \in \mathbb{R}^p$ the $k$-th block of $\boldsymbol{u}_i$ for $k \in [K]$, i.e., $(\boldsymbol{u}_i^k)_j = (\boldsymbol{u}_i)_{k \times K + j}$.

Note the eigendecomposition of $\boldsymbol{H}$ is $\sum_{i=1}^{Kp} \mu_i \boldsymbol{u}_i \boldsymbol{u}_i^\top$, and thus the solution to $\bar{\boldsymbol{X}}'_t = \boldsymbol{M}_t \bar{\boldsymbol{X}}$ is

$$\bar{\boldsymbol{X}}_t = \bar{\boldsymbol{X}}_0 + \sum_{i=1}^{Kp} c_i \boldsymbol{u}_i e^{\mu_i t}, \quad \bar{\boldsymbol{X}}_0 = \sum_{i=1}^{Kp} c_i \boldsymbol{u}_i. \tag{C.11}$$

Substituting back this solution to the LE-SDE, we have

$$\begin{aligned}
\widetilde{\boldsymbol{X}}^k(t) &= \widetilde{\boldsymbol{X}}^k(0) + \boldsymbol{M}_t \bar{\boldsymbol{X}}(t) - \mathbb{E}[\widetilde{\boldsymbol{X}}^k(0)] + \boldsymbol{\Sigma}_k^{\frac{1}{2}}(t) \boldsymbol{W}^k(t) \\
&= \widetilde{\boldsymbol{X}}^k(0) + \sum_{i=1}^{Kp} c_i \mu_i \boldsymbol{u}_i^k e^{\mu_i t} - \sum_{i=1}^{Kp} c_i \boldsymbol{u}_i^k + \boldsymbol{\Sigma}_k^{\frac{1}{2}} \boldsymbol{W}^k(t),
\end{aligned} \tag{C.12}$$

where $\boldsymbol{\Sigma}_k^{\frac{1}{2}}(t)$ is the covariance for this class. By definition, to prove separation, it suffices to identify a direction $\boldsymbol{\nu}$ such that

$$\left\langle \widetilde{\boldsymbol{X}}^k(t) - \widetilde{\boldsymbol{X}}^l(t), \boldsymbol{\nu} \right\rangle > 0 \tag{C.13}$$

with probability[4] tending to 1 as $t \to \infty$ for any two classes $k \ne l$. Substituting equation (C.12), we have equivalently

$$\left\langle \sum_{i=1}^{Kp} c_i \left(\boldsymbol{u}_i^k - \boldsymbol{u}_i^l\right)\left(\mu_i e^{\mu_i t} - 1\right), \boldsymbol{\nu} \right\rangle > \left\langle \widetilde{\boldsymbol{X}}^l(0) - \widetilde{\boldsymbol{X}}^k(0), \boldsymbol{\nu} \right\rangle + \left\langle \boldsymbol{\Sigma}_l^{\frac{1}{2}} \boldsymbol{W}^l(t) - \boldsymbol{\Sigma}_k^{\frac{1}{2}} \boldsymbol{W}^k(t), \boldsymbol{\nu} \right\rangle. \tag{C.14}$$

By the Gaussian tail bound, the right-hand side of the above display is $O_{\mathbb{P}}(C_0 + C_1 \sigma_{\max} \sqrt{t}) = O_{\mathbb{P}}(\sqrt{t})$ where $C_0$ and $C_1$ are constants that do not depend on class labels and we assume $\sigma_{\max} = \max_{k \in [K]} \sup_t \|\boldsymbol{\Sigma}_k(t)\|_2$ is finite. Due to randomization in the initialization, with probability zero $\bar{\boldsymbol{X}}^k(0) = \bar{\boldsymbol{X}}^l(0)$ and thus

$$\sum_{i=1}^{Kp} c_i(\boldsymbol{u}_i^k - \boldsymbol{u}_i^l) \ne \boldsymbol{0}, \tag{C.15}$$

with probability one. Now equation (C.10) implies that $0 < \lambda_{\min} \min_i d_i \le \mu_i$, thus when $t$ is sufficiently large, there must exist at least one index $j$ with

$$c_j(\boldsymbol{u}_j^k - \boldsymbol{u}_j^l)(\mu_j e^{\mu_j t} - 1) \ne 0 \tag{C.16}$$

as otherwise equation (C.15) is contradicted. Thus separation takes place provided that

$$\exp\{\mu_j t\} > C\sqrt{t} \tag{C.17}$$

holds for some constant $C$ that only depends on $C_0$, $C_1$, and $\sigma_{\max}$. The case where $\alpha(t)$ and $\beta(t)$ are time-varying (and so is $\mu_i(t)$) is similar with $\mu_j t$ being replaced by $\int_0^t \mu_j(s)\,ds$, i.e.,

$$\exp\left\{\int_0^t \mu_j(s)\,ds\right\} > C\sqrt{t}. \tag{C.18}$$

But equation (C.10) implies that the order of $\mu_j(t)$ is the same as $\gamma(t)$ for all $t$, hence the above condition is equivalent to

$$\exp\{\Gamma(t)\} > C'\sqrt{t}, \tag{C.19}$$

---

[4]Here the randomness comes from the dynamics as well as the random sample $\widetilde{\boldsymbol{X}}^k$.

for some constant $C'$ that only depends on $C_0$, $C_1$, $\sigma_{\max}$, and the specific choice of $\boldsymbol{H}$. Note that to prove equation (C.19), it suffices to require $\gamma(t)$ has a tail that is at least $1/t$, or

$$\gamma(t) = \omega\left(\frac{1}{t}\right) \tag{C.20}$$

as $t \to \infty$. Since the order of $\mu_1(t) = \max_i \mu_i(t)$ is also the same as $\gamma(t)$ for all $t$ in light of Equation (C.10), whenever

$$\gamma(t) = o\left(\frac{1}{t}\right) \tag{C.21}$$

as $t \to \infty$, the probability of separation tends to zero as long as $n \to \infty$ at an arbitrary rate[5]. Note that when $\gamma(t) = \Theta\left(\frac{1}{t}\right)$, the separability depends non-trivially on the constant factors, and the rate of $n$ and $t$ tends to infinity.

Thus we have shown that the order of $\gamma(t)$ characterizes a sharp phase transition in terms of separability. Finally, in the above we proved for each pair of classes $k$ and $l$, a choice $\boldsymbol{\nu} = \boldsymbol{\nu}(k, l)$ exists that ensures separation. We remark that it is possible to remove class-dependence on $\boldsymbol{\nu}$ in our case, and therefore achieve a stronger sense of separability: consider the $(K(K-1)/2)$-by-$K$ matrix $\Psi$ with $\boldsymbol{u}^k_{i(k,l)} - \boldsymbol{u}^l_{i(k,l)}$ as its rows for all $k < l$ where $i(k, l) \in [K]$ is the index of the dominant eigenvalue for the separation between these two classes as discussed above. The existence of a class-independent $\boldsymbol{\nu}$ is equivalent to $\Psi\boldsymbol{\nu} \neq \boldsymbol{0}$. This is equivalent to that the nullity of $\Psi$ is less than $K$, which is obvious since $\operatorname{rank} \Psi \geq 1$ almost surely by construction. $\qquad\square$

**Remark.** In the proof of Theorem 3.1, we use a simple consequence of the Gaussian tail bound to derive the critical order for separation of $\gamma(t)$, which is $1/t$. This is correct when we deal with asymptotic pairwise separation (Definition C.2) or asymptotic universal separation (Definition C.3), essentially a law of large numbers result. To obtain the same guarantee for uniform pairwise separation (Definition C.4) or uniform universal separation (Definition C.5), we need to take extra care for the order of $\gamma(t)$. The key observation here is that if we want to guarantee a separation direction that is independent of time $t$, it is essentially an almost sure statement. To bound the influence of the Wiener process in (C.14), we now need to apply the law of the iterated logarithms. Recall that given a standard Wiener process $W_t$,

$$\limsup_{t \to \infty} \frac{W_t}{\sqrt{2t \log\log t}} = 1, \quad \liminf_{t \to \infty} \frac{W_t}{\sqrt{2t \log\log t}} = -1, \tag{C.22}$$

hence the conditions in Theorem 3.1 can be generalized as follows:

- if $\gamma(t) = \omega\left(\frac{1}{t \log\log t}\right)$, the features are separable in the sense of uniform asymptotic universal separation;

- if $\gamma(t) = o\left(\frac{1}{t \log\log t}\right)$, the features are *not* separable in the sense of uniform asymptotic universal separation.

Note that the difference between being not uniform asymptotic universally separable and uniform asymptotic universally inseparable, the latter of which happens when $\gamma(t) = o\left(1/t\right)$. In practice, the non-uniform version of the theorem is powerful enough, since we do not ask to separate in a pre-specified direction.

### C.2.4 Proof of Proposition 3.4.

**Proposition 3.4** (Neural Collapse of the LE-ODE). *Under L-model and the same setup as in Theorem 3.1, if $\gamma(t) > 0$ and there exists some $T > 0$ such that $B(t) < 0$ for $t \geq T$, then $\left\{\bar{\boldsymbol{X}}^k(t)/\|\bar{\boldsymbol{X}}^k(t)\|\right\}_{k=1}^K$ forms an ETF as $t \to \infty$.*

---

[5]Note that when $n$ is finite, even if two classes of exactly the same mean (i.e. completely intervened) have a non-zero probability of separation: consider $2n$ i.i.d. standard Wiener processes, let $n$ of them be of class 1, and the rest class 2. Then at any time, the probability the two classes are separated is when the largest $n$ belongs to one of the classes, which occurs with probability $\frac{2}{\binom{2n}{n}} > 0$.

*Proof of Proposition 3.4.* By Proposition 3.3, when $\gamma(t) > 0$ and $B(t) < 0$ eventually, the dominating component of $\bar{X}_t^k$ is $C_1 \boldsymbol{d}_k \mathrm{e}^{\frac{1}{K}A(t) - \frac{1}{K}B(t)}$. As $t \to \infty$, the unit vector in the direction of $\bar{X}^k$ tends to $\boldsymbol{d}_k$, and therefore those unit vectors form an ETF, since $\{d_j\}_{j=1}^K$ form an ETF. $\qquad\square$

### C.3  Obtaining the Hyperplane

In this subsection we briefly discuss several methods for obtaining the class-independent $\mathbb{R}^1$ projection $\boldsymbol{\nu}$ asserted by Theorem 3.1. As suggested by the proof, it is not hard to see that a random Gaussian vector $\boldsymbol{\nu}$ satisfies Theorem 3.1 (with probability one) and it can be used to construct the map $T_{\boldsymbol{\nu}}$ that asymptotically separate all classes almost surely. However, the rate of separation will depend on the choice of $\boldsymbol{\nu}$. In practice, we can find a good $\boldsymbol{\nu}$ by solving the following optimization problem

$$\max_{\|\boldsymbol{\nu}\|_2=1} \min_{k \neq l \in [K]} |\langle \boldsymbol{c}_k - \boldsymbol{c}_l, \boldsymbol{\nu} \rangle|.$$

Its solution $\boldsymbol{\nu}^*$ is plausible since informally, it is the direction that can separate all the $K$ classes in the "shortest time" with high probability. To see the intuition behind this claim, consider the worst case of the right hand side of equation (C.14), which is $O_{\mathbb{P}}(\sqrt{t})$ and the least $t$ achieving this for all $k \neq l \in [K]$ requires $\boldsymbol{\nu}$ maximizes $\min_{k \neq l \in [K]} |\langle \boldsymbol{c}_k - \boldsymbol{c}_l, \boldsymbol{\nu} \rangle|$.

**Remark.** *In practice, estimating $\boldsymbol{c}_0 + \boldsymbol{c}_l$ from several independent trials incur high variance, despite using a larger number of trials. Although the SDE is not time-reversible, since we are mainly interested in the asymptotic behaviour, we can use an interval $[T_1, T_2]$ with $T_2 > T_1 \gg 0$ when the model is almost convergent to estimate $\boldsymbol{\nu}$ as follows:*

$$\max_{\|\boldsymbol{\nu}\|_2 \leq 1} \min_{k < l} \sum_{t=T_1}^{T_2} |\langle \bar{X}^k(t) - \bar{X}^l(t), \boldsymbol{\nu} \rangle|. \tag{C.23}$$

*Note that this problem is convex and easy to solve. In practice, we observe directly setting*

$$\boldsymbol{\nu} = \frac{\bar{\boldsymbol{X}}^k(T) - \bar{\boldsymbol{X}}^l(T)}{\left\| \bar{\boldsymbol{X}}^k(T) - \bar{\boldsymbol{X}}^l(T) \right\|} \tag{C.24}$$

*for a pair $k \neq l$ and a large $T$ can obtain relatively decent separation in $\mathbb{R}^1$ compared with equation* (C.23)*, which is the case when we construct Figure 1.*

## D   More Details on Experiments

We first recall the setup of our experiments. We generate a dataset consisting of $K = 3$ simple geometric shapes (RECTANGLE, ELLIPSOID, and TRIANGLE) that are rotated to various angles and applied Gaussian blurring, which we conveniently name Geometric-MNIST or GEOMNIST for short. A few samples from GEOMNIST are shown in Figure 2. We use a varying number of training samples per class $n_{\mathrm{tr}} \in \{80, 480, 600, 4800\}$ with the validation sample per class being $n_{\mathrm{val}} = \{20, 120, 400, 1200\}$. We also pollute each label class by randomly choosing $p_{\mathrm{err}} \cdot n_{\mathrm{tr}}$ samples to flip the label to another class. (uniform across all other classes). In this setup, we fix $n_{\mathrm{tr}} = n_{\mathrm{val}} = 500$ and set $p_{\mathrm{err}} \in \{0.1, 0.2, \ldots, 0.8\}$. In addition to GEOMNIST, we use CIFAR-10 ([28], denoted by CIFAR) for a more realistic scenario with 5000 training samples and 1000 validation samples per class. We vary the total number of classes $K \in [2, 3]$. Variants of the ALEXNET model ([29]) are used, which consists of two convolutional layers and three fully-connected layers activated by the ReLU function.

### D.1   Estimation Procedures

We will discuss here how we estimate several quantities in (4), including local elasticity strengths $\alpha(t)$ and $\beta(t)$ in the l-model and the L-model, and tail indices $r_\alpha$ and $r_\beta$.

### D.1.1 Estimation of Integrated Local Elasticity Strengths $\widehat{A}(t)$ and $\widehat{B}(t)$

Recall that in Equation (11) we give the following formulae for estimating $A(t)$ and $B(t)$:

(l-model) $\quad \begin{cases} \widehat{A}(t) & = \operatorname{avg} \operatorname{avg}_k \log \left| \frac{\breve{\mathbf{X}}(\bar{\mathbf{X}}^k - \breve{\mathbf{X}})^{K-1}}{c_0 c_k^{K-1}} \right|, \\ \widehat{B}(t) & = - \operatorname{avg} \operatorname{avg}_k \log \left| \frac{c_0}{c_k} \frac{\bar{\mathbf{X}}^k - \breve{\mathbf{X}}}{\breve{\mathbf{X}}} \right|, \end{cases} \quad \breve{\mathbf{X}}_t := \operatorname{avg}_l \bar{\mathbf{X}}_t^l,$

$$\text{(D.1)}$$

(L-model) $\quad \begin{cases} \widehat{A}(t) & = A'(t) + 2B'(t), \\ \widehat{B}(t) & = 2(B'(t) - A'(t)), \end{cases} \quad \begin{cases} A'(t) & := \log \left| \left\langle \bar{\mathbf{X}}^\top \mathbf{v}_1 - 1 \right\rangle \right|, \\ B'(t) & := \log \left| \left\langle \bar{\mathbf{X}}^\top \left( \mathbf{v}_2 - \frac{4}{3} \mathbf{v}_1 \right) \right\rangle \right|, \end{cases}$

where

$$\mathbf{v}_1 = \frac{1}{4} [1, -1, -1, -1, 1, -1, -2, -2, 0]^\top, \quad \mathbf{v}_2 = \frac{1}{3} [2, -1, -1, -1, 2, -1, 0, 0, 0]^\top. \quad \text{(D.2)}$$

We will now explain how it is done.

**Estimation in l-model.** Recall from Proposition 3.2, the per-class means $\bar{\mathbf{X}}_t$ solve the LE-ODE (7) under the l-model as

$$\bar{\mathbf{X}}(t) = \mathbf{c} \mathrm{e}^{\frac{1}{K} A(t) - \frac{1}{K} B(t)} + (\mathbf{1}_K \otimes \mathbf{c}_0) \, \mathrm{e}^{\frac{1}{K} A(t) + \frac{K-1}{K} B(t)}, \quad \text{(D.3)}$$

where $\mathbf{c} = (\mathbf{c}_k)_{k=1}^K \in \mathbb{R}^{Kp}$ and $\mathbf{c}_0 \in \mathbb{R}^p$ are constants with $\sum_{k=1}^K \mathbf{c}_k = \mathbf{0}$ and $\bar{\mathbf{X}}(0) = \mathbf{c} + \mathbf{c}_0$. The specific structure of this solution implies that

$$\breve{\mathbf{X}}_t := \operatorname{avg}_l \bar{\mathbf{X}}_t^l = \frac{1}{K} \sum_{l=1}^K \bar{\mathbf{X}}_t^l = \mathbf{c}_0 \mathrm{e}^{\frac{1}{K} A(t) + \frac{K-1}{K} B(t)} \in \mathbb{R}^K, \quad \text{(D.4)}$$

and thus for all $k \in [K]$,

$$\bar{\mathbf{X}}_t^k - \breve{\mathbf{X}}_t = \mathbf{c}_k \mathrm{e}^{\frac{1}{K} A(t) - \frac{1}{K} B(t)}. \quad \text{(D.5)}$$

It follows that

$$\left| \frac{\mathbf{c}_0}{\mathbf{c}_k} \frac{\bar{\mathbf{X}}^k - \breve{\mathbf{X}}}{\breve{\mathbf{X}}} \right| = \left| \mathbf{1}_K \mathrm{e}^{B(t)} \right|, \quad \left| \frac{\mathbf{c}_0}{\mathbf{c}_k} \frac{\bar{\mathbf{X}}^k - \breve{\mathbf{X}}}{\breve{\mathbf{X}}} \right| = \left| \mathbf{1}_K \mathrm{e}^{A(t)} \right|, \quad \text{(D.6)}$$

for all $k \in [K]$. Taking logarithm and averaging over $K$ classes and $K$ coordinate, we have the estimation equation of l-model in equation (11).

**Estimation in L-model.** Recall from Proposition 3.3, the per-class means $\bar{\mathbf{X}}_t$ solve the LE-ODE (9) under L-model as

$$\bar{\mathbf{X}}(t) = \mathbf{c}_0 + C_1 \mathbf{d} \mathrm{e}^{\frac{1}{K} A(t) - \frac{1}{K} B(t)} + \left( \sum_{l=1}^{K-1} C_{2l} \mathbf{f}_l \right) \mathrm{e}^{\frac{1}{K} A(t) + \frac{1}{K(K-1)} B(t)}, \quad \text{(D.7)}$$

where $\mathbf{c}_0$ is a constant vector with $K(K-1)$ free parameters; $\mathbf{f}_l$'s are eigenvectors corresponding to the eigenvalue $(A(t) + B(t)/(K-1))/K$. Although exact solutions can be obtained for general $K$, they are overly complicated thus we will restrict our attention to the case where $K = 3$. Define

$$\mathbf{v}_1 = \frac{1}{4} [1, -1, -1, -1, 1, -1, -2, -2, 0]^\top, \quad \mathbf{v}_2 = \frac{1}{3} [2, -1, -1, -1, 2, -1, 0, 0, 0]^\top, \quad \text{(D.8)}$$

from the proof for Proposition 3.3 (Appendix C.2.2), we immediately have

$$\mathbf{v}_1^\top \mathbf{f}_j = 0, \quad j = 1, 2, \quad \mathbf{v}_2^\top \mathbf{c}_0 = 0, \quad \mathbf{1}_K^\top \mathbf{d} = 0, \quad \text{(D.9)}$$

and

$$\mathbf{v}_1^\top \mathbf{c}_0 = 1 - \frac{w_1 + w_2 + w_3}{4}, \quad \mathbf{v}_1^\top \mathbf{d} = 1, \quad \mathbf{v}_2^\top \mathbf{f}_j = 1, \quad \mathbf{v}_2^\top \mathbf{d} = \frac{4}{3}. \quad \text{(D.10)}$$

Recall that we define $\langle \mathbf{X}(t) \rangle := \mathbf{X}(t)/\mathbf{X}(0)$, with vector-division interpreted as elementwise division, writing $C_3 = w_1 + w_2 + w_3$ with $w_i$'s being defined in Appendix C.2.2, we have

$$\begin{cases} \bar{\mathbf{X}}_t^\top \mathbf{v}_1 & = C_1 \mathrm{e}^{\frac{1}{K} A(t) - \frac{1}{K} B(t)} + 1 - \frac{C_3}{4}, \\ \bar{\mathbf{X}}_t^\top \mathbf{v}_2 & = \frac{3C_1}{4} \mathrm{e}^{\frac{1}{K} A(t) - \frac{1}{K} B(t)} + (C_{21} + C_{22}) \mathrm{e}^{\frac{1}{K} A(t) + \frac{1}{K(K-1)} B(t)}, \end{cases} \quad \text{(D.11)}$$

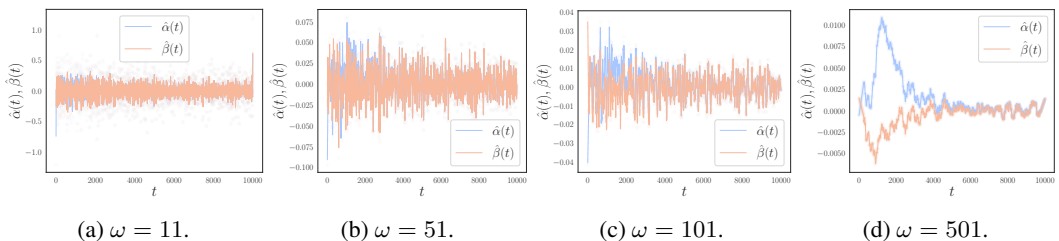

| (a) $\omega = 11$. | (b) $\omega = 51$. | (c) $\omega = 101$. | (d) $\omega = 501$. |

Figure D.1: **Effect of the window size $\omega$ in the Savitzky–Golay filter.** These estimations are performed on the GEOMNIST dataset with $p_{\text{err}} = 0$ and $n_{\text{tr}} = 3000$.

and thus

$$
\begin{cases}
\left\langle \bar{\boldsymbol{X}}_t^\top \boldsymbol{v}_1 - 1 \right\rangle & = \dfrac{C_1 \mathrm{e}^{\frac{1}{K} A(t) - \frac{1}{K} B(t)} - \frac{C_3}{4}}{C_1 - \frac{C_3}{4}} \\[3mm]
\left\langle \bar{\boldsymbol{X}}_t^\top \left(\boldsymbol{v}_2 - \frac{4}{3}\boldsymbol{v}_1\right) \right\rangle & = \dfrac{(C_{21}+C_{22})\mathrm{e}^{\frac{1}{K} A(t) + \frac{1}{K(K-1)} B(t)} + \frac{3C_3}{16}}{C_{21}+C_{22}+\frac{3C_3}{16}}.
\end{cases}
\tag{D.12}
$$

We define

$$
\begin{cases}
A'(t) & = \log \left| \left\langle \bar{\boldsymbol{X}}_t^\top \boldsymbol{v}_1 - 1 \right\rangle \right|, \\[2mm]
B'(t) & = \log \left| \left\langle \bar{\boldsymbol{X}}_t^\top \left(\boldsymbol{v}_2 - \frac{4}{3}\boldsymbol{v}_1\right) \right\rangle \right|,
\end{cases}
\tag{D.13}
$$

when $t$ is sufficiently large such that

$$
\mathrm{e}^{\frac{1}{K} A(t) - \frac{1}{K} B(t)} \gg \frac{C_3}{4C_1}, \quad \mathrm{e}^{\frac{1}{K} A(t) + \frac{1}{K(K-1)} B(t)} \gg \frac{3C_3}{16(C_{21}+C_{22})},
\tag{D.14}
$$

we have approximately

$$
A'(t) \approx \frac{1}{K} A(t) - \frac{1}{K} B(t), \quad B'(t) \approx \frac{1}{K} A(t) + \frac{1}{K(K-1)} B(t),
\tag{D.15}
$$

where $K = 3$. Hence $A(t)$ and $B(t)$ can be recovered by

$$
\widehat{A}(t) = A'(t) + 2B'(t), \quad \widehat{B}(t) = 2(B'(t) - A'(t)).
\tag{D.16}
$$

Although equation (D.16) is only an approximation that is precise only when $t$ is large, we will nonetheless use equation (D.16) for estimation in the L-model for all $t$.

### D.1.2 Estimation of $\alpha(t)$ and $\beta(t)$

Once we have estimates for $A(t)$ and $B(t)$, namely $\widehat{A}(t)$ and $\widehat{B}(t)$, we may numerically differentiate these estimates to obtain $\widehat{\alpha}(t)$ and $\widehat{\beta}(t)$. Although the composition of finite difference quotient and moving average yields visibly well results, we shall use the well-established Savitzky–Golay filter for this purpose. There is a window size parameter $\omega$ in this filter which roughly corresponds to the window size in moving averages: a smaller $\omega$ preserves more fluctuations in the original data and a larger $\omega$ smooths the data more. As a rule of thumb, we test on a set of different values for $\omega$ and choose one that is both informing and not losing too much detail in our presentations. Specifically, in the main paper, we use $\omega = 191$ for experiments on GEOMNIST and $\omega = 551$ for those on CIFAR. In the case of simulating LE-ODE solutions, we chose $\omega = 21$ to preserve finer details. We show in Figure D.1 estimated $\widehat{\alpha}(t)$ and $\widehat{\beta}(t)$ trained on GEOMNIST with $p_{\text{err}} = 0$ and $n_{\text{tr}} = 3000$ samples per class under various window sizes $\{11, 51, 101, 301\}$ for the Savitzky–Golay filter. Note that the general trend is not discovered until the window size $\omega$ is reasonably large.

### D.1.3 Estimation of Tail Index

We are interested in the tail behavior of $\alpha(t)$ and $\beta(t)$. Taking $\alpha(t)$ as an example, suppose

$$
\alpha(t) \sim \frac{\alpha_0}{(1+t)^r},
\tag{D.17}
$$

for some $r$, which is the tail index of $\alpha(t)$, we have

$$
A(t) = \begin{cases}
\frac{\alpha_0 (1+t)^{1-r}}{1-r}, & 0 < r < 1, \\[2mm]
\alpha_0 \log(1+t), & r = 1,
\end{cases}
\tag{D.18}
$$

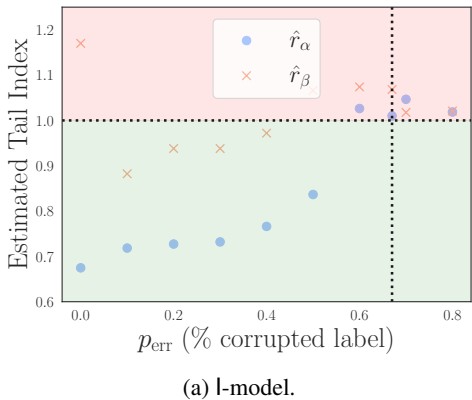
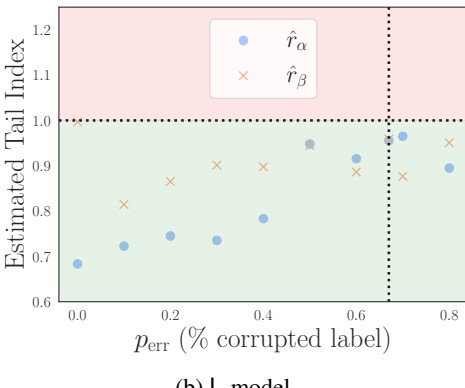

(a) l-model.

(b) L-model.

Figure D.2: **Estimated tail indices versus label corruption ratio $p_{\text{err}}$.** The tail indices are estimated using equation (D.19) with $\widehat{\alpha}(t)$ and $\widehat{\beta}(t)$ estimated via (a) l-model, and (b) L-model. Although the case for the L-model does not exhibit a clear phase transition, we note around $p_{\text{err}} \approx 2/3$, the tail index of $\widehat{\beta}(t)$ begins to dominate that of $\widehat{\alpha}(t)$.

Hence with $t$ being sufficiently large, we can estimate $r$ using

$$\frac{\log A(t)}{\log(1+r)} = (1-r) + \frac{\log \alpha_0}{\log(1+t)} - \frac{\log(1-r)}{\log(1+t)} \approx 1-r, \tag{D.19}$$

and $\widehat{r}_\alpha = 1 - \text{avg}_{t \geq T_0} \log A(t)/\log(1+t)$ for some sufficiently large $T_0$. We estimate $\widehat{r}_\beta$ similarly. Although this estimator suffers from large bias when the true model has a constant offset, i.e., when $\alpha(t) \sim \alpha_1 + \alpha_0/(1+t)^r$, we choose it over other estimators based on $\alpha(t) \sim \alpha_1 + \alpha_0/(1+t)^r$ as it is simpler and it is directly based on the integrated local elasticity strength $A(t)$ without the need to perform numerical differentiation beforehand.

### D.2 More Results from Experiments

**Effects of Training Sample Size $n$.** We show in Figure D.3 estimated $\widehat{A}(t)$, $\widehat{B}(t)$, $\widehat{\alpha}(t)$ and $\widehat{\beta}(t)$ versus training time $t$ on both GEOMNIST and CIFAR under both l-model and L-model with various per-class sample size $n = n_{\text{tr}} \in \{80, 480, 600, 4800\}$. We choose the window size $\omega = 151$ when applying the Savitzky–Golay filter. Note that the estimates under both l-model and L-model do not vary significantly under different choices of $N$ and share similar trends: (i) $\beta(t)$ is dominated by $\alpha(t)$; (ii) $\alpha(t)$ has an initial increasing stage and a second stage converging to the vicinity around zero. This is expected since our theory is independent of $n$ once $n$ is reasonably large.

**Effects of Number of Classes $K$.** In Figure D.4 we show the estimated $\widehat{A}(t)$, $\widehat{B}(t)$, $\widehat{\alpha}(t)$ and $\widehat{\beta}(t)$ versus training time $t$ on CIFAR under the l-model with number of classes $K \in \{2, 3\}$. We note that in both cases, generally speaking, the shapes of $A(t)$ and $B(t)$ are similar: $A(t)$ increases while $B(t)$ decreases, suggesting this behavior is more general and is likely independent of the number of classes. On the other hand, note that the number of classes affects the scale of $A(t)$ and $B(t)$.

**Effects of Label Corruption Ratio $p_{\text{err}}$.** In Section 3 we have shown that separability depends on the tail behavior of $\alpha(t) - \beta(t)$. In Figure D.5 we demonstrate this more directly. Here we intentionally choose a large window size of $\omega = 351$ to highlight the trends of $\alpha(t)$ and $\beta(t)$. Note that $A(t)$ and $B(t)$ (and consequently $\alpha(t)$ and $\beta(t)$) become more mixed and indistinguishable as we increase the label corruption ratio $p_{\text{err}}$ from 0 (no corrupted label) to 0.8. This also reaffirms the important rule played by the local elasticity strengths $\alpha(t)$ and $\beta(t)$ in terms of separability.

We also plot the estimated tail indices versus label corruption ratio $p_{\text{err}}$ using $\alpha(t)$ and $\beta(t)$ estimated under both l-model and L-model in Figure D.2. Although in Figure D.2b there is no sharp phase transition boundary when we increase $p_{\text{err}}$ as Figure D.2b does, we observe that at around $p_{\text{err}} = 2/3$, the estimated tail index of $\widehat{\beta}(t)$ begins to dominate that of $\widehat{\alpha}(t)$, which also supports our Theorem 3.1.

**Local Elasticity Strengths Adjusted for Logits Norms** In our LE-SDE/ODE model l-model and L-model, each block of the $\boldsymbol{H}$ matrix has operator norm 1. Hence it is of interests to inspect the

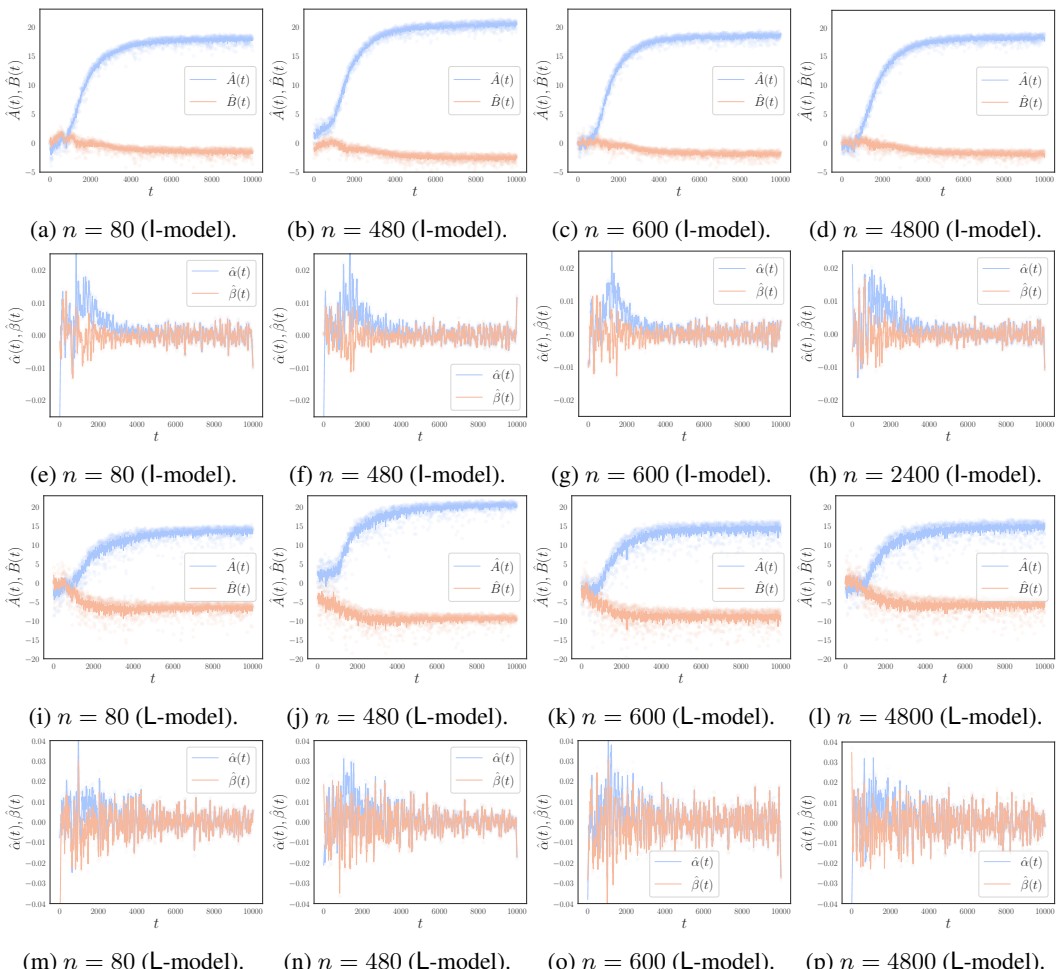

Figure D.3: **Effect of per-class sample size** $n$. Estimated $\widehat{A}(t)$, $\widehat{B}(t)$ (the first and the third rows) and $\widehat{\alpha}(t)$, $\widehat{\beta}(t)$ (the second and the fourth rows), under l-model (the first two rows) and L-model (the last two rows). Note that $n$ appears to be insignificant in determining the shapes and magnitudes of the curves.

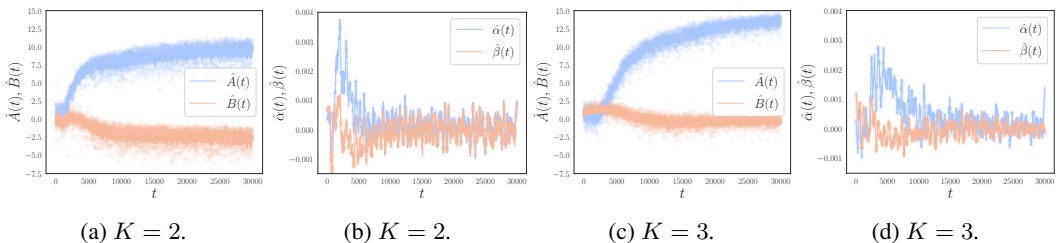

Figure D.4: **Effect of number of classes** $K$. Estimated $\widehat{A}(t)$, $\widehat{B}(t)$ ((a) and (c)) and $\widehat{\alpha}(t)$, $\widehat{\beta}(t)$ ((b) and (d)) on CIFAR with $K = 2$ ((a)-(b)) and $K = 3$ ((c)-(d)), all under l-model. Note that the general trends with different $K$'s are similar.

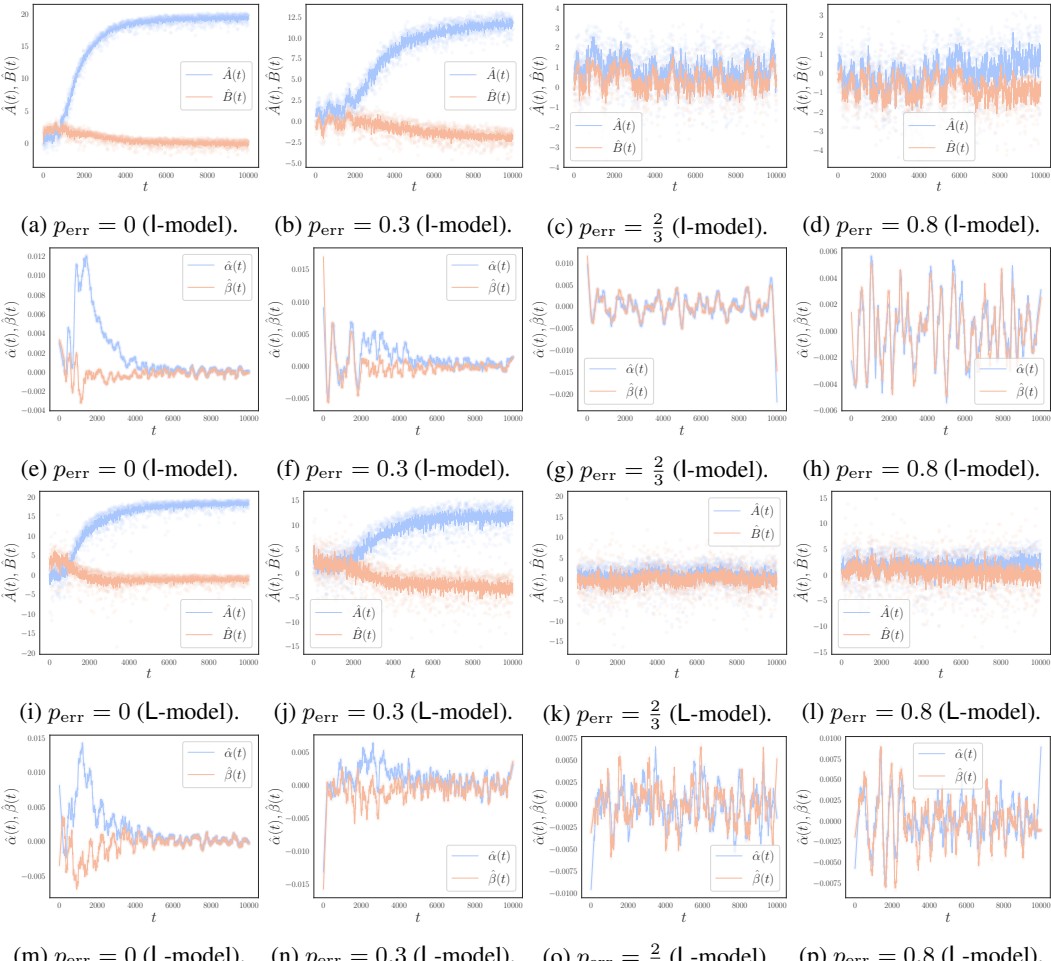

(a) $p_{\text{err}} = 0$ (l-model).    (b) $p_{\text{err}} = 0.3$ (l-model).    (c) $p_{\text{err}} = \frac{2}{3}$ (l-model).    (d) $p_{\text{err}} = 0.8$ (l-model).

(e) $p_{\text{err}} = 0$ (l-model).    (f) $p_{\text{err}} = 0.3$ (l-model).    (g) $p_{\text{err}} = \frac{2}{3}$ (l-model).    (h) $p_{\text{err}} = 0.8$ (l-model).

(i) $p_{\text{err}} = 0$ (L-model).    (j) $p_{\text{err}} = 0.3$ (L-model).    (k) $p_{\text{err}} = \frac{2}{3}$ (L-model).    (l) $p_{\text{err}} = 0.8$ (L-model).

(m) $p_{\text{err}} = 0$ (L-model).    (n) $p_{\text{err}} = 0.3$ (L-model).    (o) $p_{\text{err}} = \frac{2}{3}$ (L-model).    (p) $p_{\text{err}} = 0.8$ (L-model).

Figure D.5: **Effect of label corruption ratio $p_{\text{err}}$.** Estimated $\widehat{A}(t)$, $\widehat{B}(t)$ (the first and the third rows) and $\widehat{\alpha}(t)$, $\widehat{\beta}(t)$ (the second and the fourth rows), under l-model (the first two rows) and L-model (the last two rows) on GEOMNIST under various label corruptions ratios $p_{\text{err}} \in \{0, 0.3, 2/3, 0.8\}$ are shown. Note $\widehat{A}(t)$ and $\widehat{B}(t)$, $\widehat{\alpha}(t)$ and $\widehat{\beta}(t)$ become indistinguishable when $p_{\text{err}}$ is large, supporting Theorem 3.1.

local elasticity strengths $\alpha(t)$ and $\beta(t)$, when adjusted for the evolution of the norm of $\bar{\boldsymbol{X}}^k(t)$. As a surrogate, we simply multiply the strengths by the average norm, $\text{avg}_k \left\| \bar{\boldsymbol{X}}^k(t) \right\|_2$ and show the results in Figure D.6. Again we use a large window size $\omega = 351$ in the Savitzky–Golay filter to highlight the general trends. We observe that when $p_{\text{err}} = 0$, the general trend is similar to the unadjusted versions: $\alpha(t)$ has an initial increasing stage, and then converges to the vicinity of zero. We also note that $\alpha(t)$ and $\beta(t)$ become more indistinguishable as we increase $p_{\text{err}}$, similarly to the unadjusted cases Figure D.5.

**Simulations of the LE-ODE.** With $\alpha(t)$ and $\beta(t)$ estimated (using either the l-model or the L-model), we can simulate the LE-ODE under either l-model or L-model. In our setup, the initial value for the $k$th class is set to be $\boldsymbol{\zeta}^K \sim \mathcal{N}_K(\boldsymbol{0}, \sigma_k \boldsymbol{I}_{K^2})$ for all $k \in [K]$ with $\sigma_k = \left\| \bar{\boldsymbol{X}}^k(0) \right\|_2 / \sqrt{K}$ with $\bar{\boldsymbol{X}}^k(0)$ sampled from simulations in DNNs. Empirically, we find that simulations under the l-model does not generate faithful trajectories compared with the ground truth (genuine dynamics from simulations on deep neural nets), as shown in Figure D.7. Hence in Figure D.8, we only show the case when the simulation is done under the L-modelwhere the captions indicate which model (l-model or L-model) the estimation of $\alpha(t)$ and $\beta(t)$ is performed. Here $\bar{X}_i^k(t) \in \mathbb{R}$ denotes the $i$-th logit from

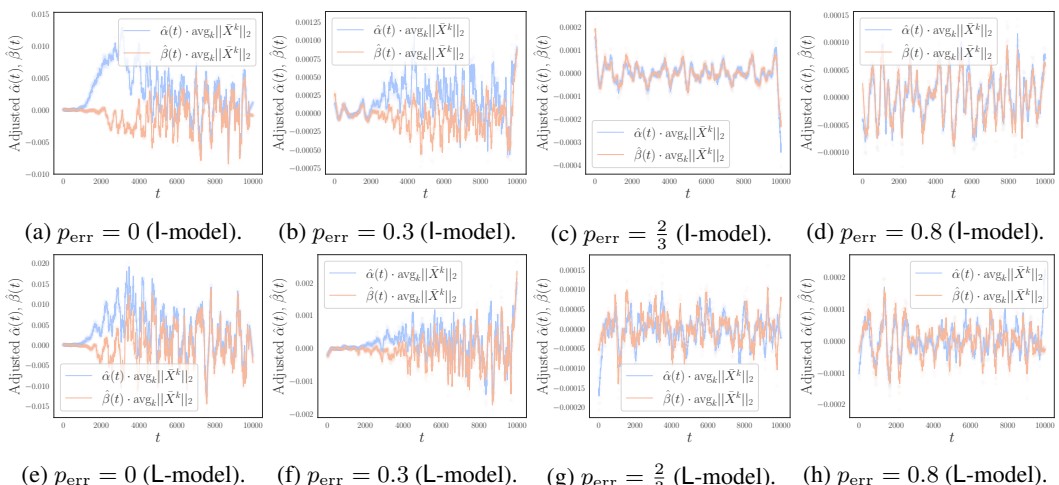

(a) $p_{\mathrm{err}} = 0$ (l-model).    (b) $p_{\mathrm{err}} = 0.3$ (l-model).    (c) $p_{\mathrm{err}} = \frac{2}{3}$ (l-model).    (d) $p_{\mathrm{err}} = 0.8$ (l-model).

(e) $p_{\mathrm{err}} = 0$ (L-model).    (f) $p_{\mathrm{err}} = 0.3$ (L-model).    (g) $p_{\mathrm{err}} = \frac{2}{3}$ (L-model).    (h) $p_{\mathrm{err}} = 0.8$ (L-model).

Figure D.6: **Effect of label corruption ratio $p_{\mathrm{err}}$ (adjusted for logits norm).** Estimated $\widehat{\alpha}(t)$, $\widehat{\beta}(t)$ under l-model (the first two rows) and L-model (the last two rows) on GEOMNIST under various label corruptions ratios $p_{\mathrm{err}} \in \{0, 0.3, 2/3, 0.8\}$ are shown. All quantities were multiplied by the average norm of per-class means in each iteration, i.e., Note $\widehat{A}(t)$ and $\widehat{B}(t)$, $\widehat{\alpha}(t)$ and $\widehat{\beta}(t)$ become indistinguishable when $p_{\mathrm{err}}$ is large, supporting Theorem 3.1.

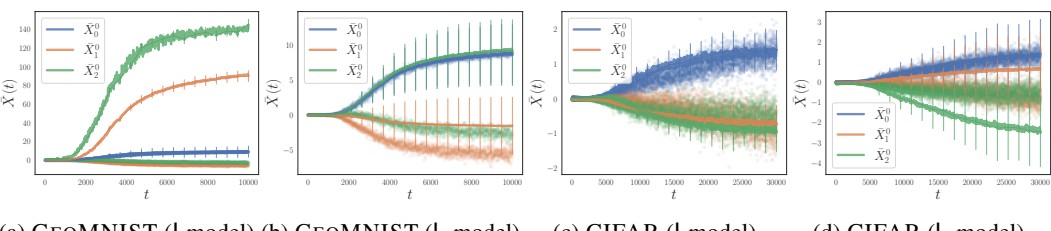

(a) GEOMNIST (l-model). (b) GEOMNIST (L-model).    (c) CIFAR (l-model).    (d) CIFAR (L-model).

Figure D.7: **Simulated LE-ODE solutions under the l-model versus genuine dynamics.** We use $\widehat{\alpha}(t)$ and $\widehat{\beta}(t)$ estimated from l-model ((a) and (c)) or L-model ((b) and (d)) and numerically simulate the solution under the l-model. The results were overlaid with true dynamics from neural nets. Note that the results are uniformly bad, indicating l-model is not expressive enough to capture the genuine dynamics in deep neural nets.

the per-class mean logits vector of the $k$-th class; as explained in Appendix B, a well-trained model should have the $k$-th logit being the largest among all $i \in [K]$ in $\bar{X}_i^k$ when $t$ is sufficiently large. Since the estimation of $\alpha(t)$ and $\beta(t)$ relies on numerical differentiation, which smooths the data and reduces their magnitudes, we manually align the $k$-th logit from the $k$-th per-class mean vector, $\bar{X}_i^k$, with that from the ground truth. All simulations are performed with $K = 3$ and for $N = 500$ trials with the initial data being Gaussian random vectors with zero mean and identity covariance such that the norm at initialization is approximately equal to that from the ground truth.

We observe the following: **(i) Both l-model and L-model approximately preserve the relative magnitude between different logits.** We find that the ratios between converging values of sample paths (dashed lines), $\lim_{t \to \infty} \bar{X}_i^k(t)/\bar{X}_j^k(t)$ for $i \neq j$, are roughly equal to the ground truth. This indicates that both models can capture the relative magnitudes of logits in real dynamics. **(ii) The l-model fails to identify the correct class.** After manual alignment, we note the l-model does not always yield faithful results, meaning the largest logit from the $k$-th class, $\max_{l \in [K]} \bar{X}_l^k(t)$ for large $t$, is not necessarily $k$. This is not surprising though, since the l-model itself does not differentiate features from different classes, and $\alpha(t)$ and $\beta(t)$ thus estimated fails to honor the interactions between different classes. (iii) **The L-model is able to identify the correct class while oblivious to incorrect classes.** On the other hand, simulated trajectories from the L-model faithfully recover the trajectories of correct classes $\mathrm{argmax}_{l \in [K]} \bar{X}_l^k$ for each $1 \leq k \leq K = 3$. However, for any $k \in [K]$, we note that the trajectories of incorrect classes (i.e., $\{j : j \neq k\}$) are sometimes mismatched. This is because that the L-model, by construction, only uses the information from the correct class, i.e., $\boldsymbol{H}_{i,j}$

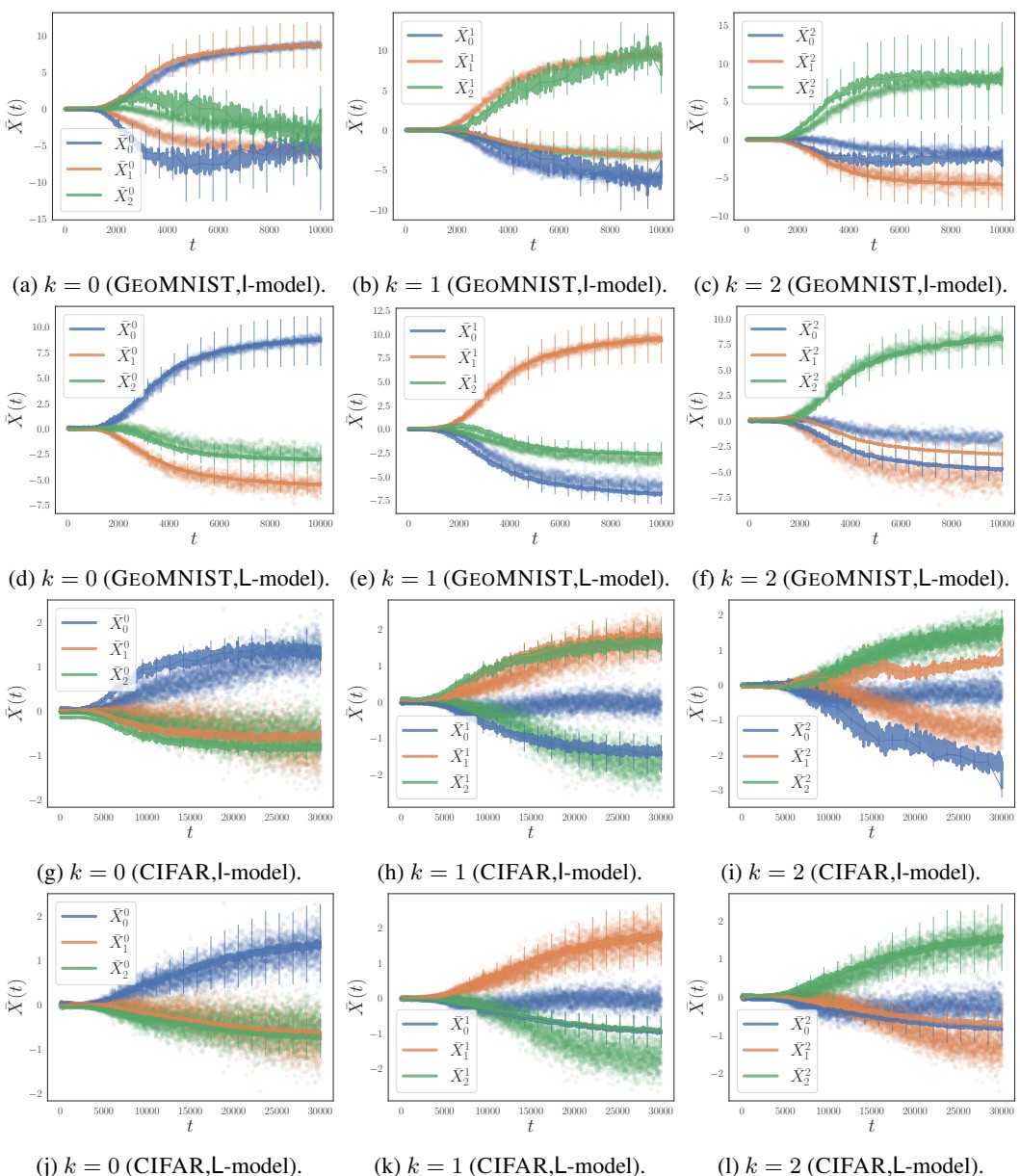

(a) $k = 0$ (GEOMNIST,l-model). (b) $k = 1$ (GEOMNIST,l-model). (c) $k = 2$ (GEOMNIST,l-model).

(d) $k = 0$ (GEOMNIST,L-model). (e) $k = 1$ (GEOMNIST,L-model). (f) $k = 2$ (GEOMNIST,L-model).

(g) $k = 0$ (CIFAR,l-model). (h) $k = 1$ (CIFAR,l-model). (i) $k = 2$ (CIFAR,l-model).

(j) $k = 0$ (CIFAR,L-model). (k) $k = 1$ (CIFAR,L-model). (l) $k = 2$ (CIFAR,L-model).

Figure D.8: **Simulated LE-ODE paths under the L-model versus genuine dynamics.** The simulation is done under the L-model with estimated $\widehat{\alpha}(t)$ and $\widehat{\beta}(t)$ from the l-model (the first and the third rows) and the L-model (the second and the fourth rows), on the GEOMNIST (the first two rows) and CIFAR (the last two rows). We show the trajectories for each class $k \in [3]$, where $\bar{X}_l^k(t)$ denotes the path of the $l$-th logit of the $k$-th class for $k, l \in [3]$.

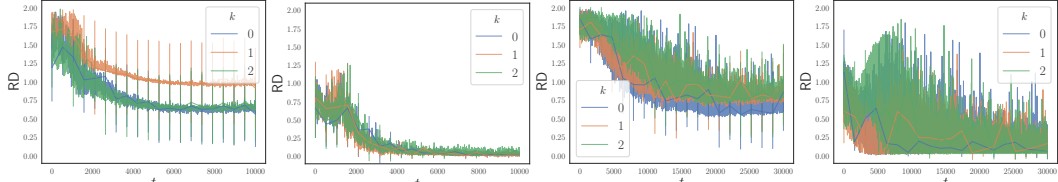

(a) GEOMNIST (l-model).(b) GEOMNIST (L-model).    (c) CIFAR (l-model).    (d) CIFAR (L-model).

Figure D.9: **Relative difference RD$_k$ between genuine and simulated dynamics.** The RD is computed according to equation D.21 (the lower the better). Note that the L-model performs better than l-model throughout training and better captures the later stages of the training (indicated by decreasing RD), supporing our discussions in Appendix B.2.

is set to be $d_j d_j^\top / d_j^\top d_j$ while not specifying other directions. A model that is capable of identifying incorrect classes needs necessarily more information on those classes. We postulate that a better model might be

$$H_{i,j} = p_j \frac{d_j d_j^\top}{d_j^\top d_j} + \sum_{l \neq j}^{K} p_l \frac{d_l d_l^\top}{d_l^\top d_l}, \tag{D.20}$$

with $p_j > p_l$ for all $l \neq j$. We leave explorations along this direction in future works.

**Residue of LE-ODE Simulations.**    Under the same experiment setup, we also visualize the residue of using LE-ODE to imitate the genuine dynamics of neural nets, as shown in Figure D.9. We measure the goodness-of-fit via relative difference (RD) defined for each class $k \in [K]$ as

$$\mathrm{RD}_\mathrm{k}(t) := \frac{\left\| \bar{X}^k(t) - \bar{Y}^k(t) \right\|_{H^k}}{\left( \left\| \bar{X}^k(t) \right\|_2 + \left\| \bar{Y}^k(t) \right\|_2 \right) / 2}, \tag{D.21}$$

where $\bar{X}(t)$ and $\bar{Y}(t)$ are genuine and simulated trajectories, respectively, and $\|\cdot\|_H$ denotes the norm induced by the matrix $H$ that is used to define models (i.e., the identity matrix for the l-model and $\bar{H}$ in equation 8 for the L-model). This choice normalizes the difference under the similarity defined by $H$ (which we care the most) and ranges from 0 to 2 (the lower the better). From the results we note that: **(i) The L-model performs consistently better than the l-model throughout training.** We note that the RD under the L-model are overall smaller and there is no significant differences across classes. **(ii) The L-model is better suited for capturing later stages of training.** This can be seen from a decreasing trend of the RD under the L-model, which corroborates our discussions in Appendix B.2. In particular, the approximation becomes better as training progresses (indicated by a decreasing RD). However, the performance of the L-model around initialization is still commendable. **(iii) The L-model is not perfect.** Although RD under L-model is small in the terminal stage (of the order $10^{-2}$ to $10^{-1}$), it is non-zero, and it has a higher RD in the early stage of training. This indicates that non-dominant directions are also important for the LE-ODE to capture the remainder of the feature similarity. A possible avenue for future research in this regard is discussed in equation D.20.

### D.3    The Two-Stage Behavior of Logits Evolution

An interesting observation that can be made when going through the experiments is the emergence of a two-stage behavior in many quantities. Specifically: (i) the training loss and validation loss do not decrease at a perceivable rate in the first few hundreds (or thousands) iterations; then they begin to drop at a relatively fast rate until convergence; (ii) the local elasticity strength $\alpha(t)$ increases at the initial stage, then drops, which is also manifested by the behavior of $A(t)$, which resembles roughly a sigmoidal curve; (iii) the magnitudes of $\mathrm{avg}_{k \in [K]} \left\| \bar{X}^k \right\|_2$ also resembles that of $A(t)$ (not shown), which has a fast growing stage and a converging stage.

We coin this seemingly generic phenomenon as the *two-stage behavior* that consists of a *de-randomization* stage and an *amplification* stage, and demonstrate the supporting experiments in Figure D.10. Here we trained on GEOMNIST with $K = 3$, where each triangle is a hyperplane

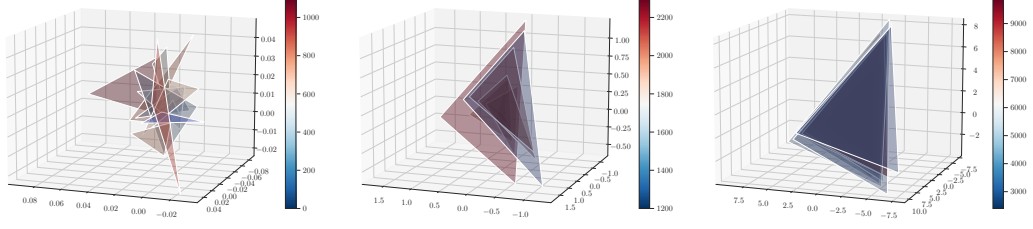

| (a) De-randomization stage. | (b) Amplification stage. | (c) Amplification stage (cont'd). |

Figure D.10: **The hyperplanes formed by $\{\bar{X}_t^k\}_{k=1}^3$ with colors corresponding to iterations.** (a) In the first 1200 iterations, the validation loss does not drop perceivably, and the hyperplanes are visibly chaotic. (b)-(c) Starting from around the 1200-th iteration, the validation loss drops, and the hyperplanes exhibit converging behavior.

spanned by the per-class mean logits vectors, $\bar{\boldsymbol{X}}^k(t) \in \mathbb{R}^3$ for $k \in [3]$. In Figure D.10a we plot those hyperplanes for the first 1200 iterations; Figure D.10b iterations from 1200 to 2400; and Figure D.10c the remaining iterations. We observe that in the first 1200 iterations, the hyperplanes are "chaotic" in that their behavior is highly dependent on specific initialization values. In this de-randomization stage, the supervision guides the dynamics to identify the correct and *deterministic* (c.f. Proposition 3.4) direction for the separation of features from random initialization, thus the name *de-randomization*. With the correct direction being identified (around iteration 1200), the losses begin to drop at a relatively fast speed and the hyperplane remains approximately the same throughout the training while the magnitude of logits increases, which pushes the classes to be more discriminative and further drives down the losses, hence the *amplification* stage.

We believe a more precise characterization of $\alpha(t)$ and $\beta(t)$ and potentially generalization to the $\boldsymbol{E}$ matrix would likely help us to study this two-stage behavior more rigorously and potentially answer the interesting questions such as *how long does the first stage take on average? How does local elasticity ($\alpha(t)$ and $\beta(t)$) affect this behavior?* We leave these questions for future works.

## E  Future Work and Extensions

**General LE Matrix.**    Throughout the paper, we have modeled the LE matrix $\boldsymbol{E}$ as $(\alpha(t)-\beta(t))\boldsymbol{I}_K + \beta(t)\mathbf{1}_K\mathbf{1}_K^\top$. Although it depends on $t$, this model falls short when we move into the more realistic realm where the inter-class and intra-class effects are dependent on the class labels. When $\boldsymbol{E}$ is SPD, under the same assumptions as in Theorem 3.1, due to a theorem by Schur (Theorem 9.B.1, [37]), we know the eigenvalues of $(\boldsymbol{E} \otimes \boldsymbol{I}_K) \circ \boldsymbol{H}$ are bounded within $\lambda_{\min}(\boldsymbol{E}) \min_i H_{ii}$ and $\lambda_{\max}(\boldsymbol{E}) \max_i H_{ii}$, where $\{H_{ii}, i \in [Kp]\}$ is the diagonal entries of the matrix $\boldsymbol{H}$, hence we expect a very similar result in this case as in Theorem 3.1. In the more general case where $\boldsymbol{E}$ is symmetric but not necessarily semi-definite, a more precise analysis on the spectrum of $(\boldsymbol{E} \otimes \boldsymbol{I}_K) \circ \boldsymbol{H}$ is needed, though the proof framework would not be too different.

**Mini-batch Training, Imbalanced Datasets, and Label Corruptions.**    As discussed in Section 3, we can incorporate mini-batches and imbalanced datasets in our model easily. Taking imbalanced datasets as an example, recall that each block of $\boldsymbol{M}_t$ in equation (5) takes the form of $E_{k,l}\boldsymbol{H}_{k,l}/K$, where $1/K$ signifies that each class has the same $1/K$ probability of being sampled during any iteration in training. This can be generalized by changing the $(k,l)$-th block to $E_{k,l}\boldsymbol{H}_{k,l} \cdot p_l$ for $\sum_l p_l = 1$, where $p_l$ is the probability of class $l$ being sampled. More succinctly, instead of defining $\boldsymbol{M}_t = (\boldsymbol{E}_t \otimes \boldsymbol{I}_K) \circ \boldsymbol{H}/K$, we let $\boldsymbol{M}_t = (\boldsymbol{E}_t \otimes \boldsymbol{P}) \circ \boldsymbol{H}/K$ for a $K$-by-$K$ doubly stochastic matrix $\boldsymbol{P}$ that models this sampling effect. In the same vein, we can also model the case when the data are polluted by corrupted labels. Let $\boldsymbol{P} = (p_{k,l})_{k,l} \in \mathbb{R}^{K \times K}$ with $p_{k,l}$ representing the probability of a sample from class $k$ mis-labelled as class $l$. A well-defined model needs further assumption on the structure of $\boldsymbol{P}$ and we leave this theoretical modeling to future work.

**Covariance Structures and Fine-Grained Analyses.**    Although our model encompasses a covariance term in the LE-SDE model, we do not explicitly use its structure. Nonetheless, as indicated

from the proof of Theorem 3.1 (cf. Appendix C.2.3), the relative magnitude of the covariance to the drift term (i.e., the local elasticity effect) affects the separation when $\gamma(t) = \Theta(1/t)$. However, when the order of $\gamma(t)$ is guaranteed to be strictly above or below $1/t$ as $t \to \infty$, the covariance affects the separation only through the constant factor for the separation rate. That said, a more precise analysis of covariance would by all means facilitate fine-grained analyses at the edge of separation.

**Beyond L-model for Imitating Genuine Dynamics of DNNs.** We show in Section 4 that using estimates of $\alpha(t)$ and $\beta(t)$, the L-model can be used to imitate the genuine dynamics of DNNs. As is shown in more detail in Appendix D.2, we note that although simulations under the L-model are already superior to those under the l-model in that the correct classes are identified, the L-model sometimes still fails to identify the correct trajectories for the incorrect classes. This is not very surprising though, as the supervision from the labels only affects L-model though the $\boldsymbol{H}$ matrix, whose $(i,j)$-th block is defined as $\bar{\boldsymbol{H}}^j = \boldsymbol{d}_j \boldsymbol{d}_j^\top / \boldsymbol{d}_j^\top \boldsymbol{d}_j$ where $\boldsymbol{d}_j = \boldsymbol{e}_j - \boldsymbol{1}_K/K$ — which only encodes information about the correct class. We postulate that a more precise model might be to assign the $(i,j)$-th block of $\boldsymbol{H}$ as

$$\boldsymbol{H}_{i,j} = p_j^j \bar{\boldsymbol{H}}^j + \sum_{l \neq j}^K p_l^j \bar{\boldsymbol{H}}^l, \quad p_j^j > p_l^j, \quad \sum_{l=1}^K p_l^j = 1, \quad \forall l \neq j, \quad j \in [K]. \tag{E.1}$$

**The Two-Stage Behavior.** In Section 4, we observed a clear two-stage behavior of our numerical simulation of our LE-SDE, as well as in real deep learning dynamics. The first stage is a *de-randomization* stage, which gradually eliminates the effect of random initialization and searches for the correct directions to be separated. The second stage is an *amplification* stage, where the model amplifies the magnitudes of the features in those directions. As can be seen from Appendix D.2, these empirical observations naturally lead to many interesting questions: Is this a universal phenomenon in deep learning, and what are the conditions to guarantee entering the second stage? Can our LE-SDE predict such a two-stage phenomenon theoretically? What is the role of local elasticity in this transition? We leave the investigation of these questions to future works.