# OpenReview forum: "Imitating Deep Learning Dynamics via  Locally Elastic Stochastic Differential Equations"
_NeurIPS.cc/2021/Conference — NeurIPS 2021 Poster_

### Official Review · Reviewer_imQZ · 2021-07-06

**Rating:** 5
**Confidence:** 4

**Summary:**

The paper proposes a proxy tractable dynamics to study how the latent features of neural networks evolve inter- and intra-class during training. The core of this dynamics is a linear ODE, which is shown to exhibit inter-class separability once a “locally elastic” condition is met. The paper then examines this dynamics for two choices of the H matrix (which is a parameter of the dynamics), and concludes with experiments to demonstrate the relevance of the proposed theory with the actual simulation.


=====================
After rebuttal:
Thanks for the detailed reply!

Overall while I am pleased that the authors made a lot of efforts to clarify, I'm not quite convinced. Some notes:
- I agree that time-dependence of E(t) is a good point; my concern about linearization (which I believe is also the concern of some other reviewers) is actually about why at each time t, F(X(t),t), in the authors' response, is modeled by a linear relationship with X(t). Why ignoring higher-order terms? Are they much smaller than the linear term? I do not see why the higher-order terms can be absorbed into the noise term.
- I do not agree with the argument that $c_1\neq c_2$ assumption can be alleviated by taking the results "in probability". Clearly the derivation in the paper requires taking a law-of-large-numbers average.

I agree (and well understood) that the paper does not aim to mimic exactly the dynamics; the aim is to provide a reasonable proxy dynamics that can explain interesting properties. However I feel the paper needs a careful justification to convince readers that this proxy is not oversimplifying.

Given that the paper may attract interests, I'm willing to raise the score, but not fully convinced of the quality to recommend acceptance. If a reviewer is willing to champion the paper, accepting the paper would not be a bad outcome to me.

**Limitations And Societal Impact:**

Yes

**Main Review:**

The subject of the paper is timely and the result, if convincingly shown to be a good proxy to study neural net dynamics, would be very interesting. However I do not find the proposal to have good potential (or have been sufficiently demonstrated to have good potential), and in fact, I suspect the proxy dynamics could be over-simplifying. More comments follow.

The paper models the evolution of the features by a linear class-based approximation, plus some noise. Firstly the paper does not give an intuition or an argument why the relation should be linear. I think this is an important modeling assumption: linear approximation is typically good only if the effect size is small. While it is intuitive near the end of training when changes are small, it is unclear why it should hold throughout all training time, especially at time closer to initialization when a lot of the training takes place. This shortcoming could be critical, since the paper models the whole evolution, and so any mis-modeling around initialization could lead to bad results eventually by the end of training time.

Secondly the paper does not justify why the influence onto one feature could be simplified per class (by restricting the H matrix to be dependent only on the classes). Similar to above, this may not hold closer to initialization when different classes could occupy well-overlapped regions in the feature space. The only piece of argument I can find from the paper is the expectation of locally elastic behavior, but it is important to note that the notion of (class-based) local elasticity in this paper is different from papers by W. Su an co-authors.

Thirdly — a point which is mainly to point out the modeling downside, but perhaps not as critical as previous points — one can see that the stochastic part in SDE modeling does not matter at all. Rather it is the linear ODE, resulted from the averaging over samples within the same class, that dictates the picture. The fact that it is a linear ODE makes it straightforward to obtain separability. However it may seem to be oversimplifying neural nets, which are known to be highly nonlinear.

One specific comment I have on the theorems (in particular, Theorem 2.1, which is easier to discuss) is that separability cannot hold if $c_1 = c_2$. Recall that $c_1$ is the average of the features in class 1 at initialization, and the same for $c_2$. With the typical initialization in neural nets, I doubt this holds. For example, if the feature $X = Wx$ (linear model) where W is the weight matrix, with $W$ being standard zero-mean Gaussian, wouldn’t the average be zero and hence $c_1 = c_2$? Also note that the assumption $c_1\neq c_2$ does not make sense at all: an untrained unbiased network should not be able to do any classification yet.

Another concern is on the guesses on H (the I-model and L-model). It is clear that I-model is insufficient. While it seems L-model is good as per Fig. 5, this guess is based on a well-trained net, and so it is unclear how well it models the behavior near initialization. This is not clear from Fig. 5, since already around t=3000 (where it is not easy to spot any deviation from Fig. 5), the validation accuracy is 90%, i.e. training is close to being finished; in other words, the good match after t>3000 is an easy test for the theory. In fact, one can observe quite big deviation in Fig D.8. This raises a doubt on the proposed guesses of H.

Fig 4(c) is also unconvincing. Firstly we know that I-model is not a good choice, so this should have been done for L-model instead. But when looking at one for L-model (Fig. D.2(b)), one sees a mixed picture. This figure seems to suggest the non-separable regime should start around p=0.5, but this is at odds with firstly the correct threshold p=0.66 and secondly Fig. 4(b) where the validation accuracy for p=0.5 is significantly better than random guessing.

Some other comments:
- Corollary 3.4 does not seem to be correct: for example, if B(t) = -0.1 and A(t) = t, then the term ${\bf d}e^{A(t)/K - B(t)/K}$ is not dominating the other term.
- The proofs are not fully rigorous. For instance, to deduce the equations after line 133 on page 4, one has to do careful probabilistic argument to prove it holds at all t, as n tends to infinity.
- In Fig. (4) caption: “crosses, 1 entering…” is not readable.


**Time Spent Reviewing:**

8

---

> ### Author Response · Authors · 2021-08-10
> **Response to Reviewer imQZ**
>
>
> Thanks for reviewing our paper and for your comments! We really appreciate your careful reading and thoughtful questions. Please find below our responses to your comments.
>
> > *Firstly the paper does not give an intuition ... by the end of training time.*
>
> **Oversimplification.** We would also like to invite you to our response in the main thread. We want to walk through our thinking process while constructing the model here.
>
> Our aim is not to faithfully recover the real dynamic but to show that LE is integral to a (good) deep learning theory through a tractable model, thereby pinpointing further directions in this area. In this spirit, a linear approximation is the most straightforward starting point we can use to illustrate this point. On a technical note, a linear approximation may still be a reasonable assumption because the noise term can absorb higher-order terms. This extension requires us to assume heavy-tailed distributions instead of the Gaussian distribution. We have some preliminary results toward this extension and wish to present them in the updated version, provided that our papers were finally accepted.
>
> We also wish to clarify that our model is **not** a simple linearization.
>
> Note that the temporal dependence of the LE matrix $E(t)$ makes our modeling more “nonlinear”: it provides sufficient flexibility to capture time-varying LE effects. Taking the L-model as an example, since we are interested in the dynamics of the logits, we model the $H$ matrix to be such that it *eventually* maps the features to the same point in expectation as what the genuine dynamics would do, provided training is effective (that is, features become more discriminative as training progresses). As such our model is not a vanilla “linearization” around the model at convergence. We should have made this point clearer in the writing. We have modified the paper accordingly.
>
> **Effective size.** We found that the model is more similar to exploring different directions in the feature space before a lot of training took place (Cf. discussions on the two-stage behavior at the end of the Appendix). Around initialization, the LE coefficients $\alpha(t)$ and $\beta(t)$ are observed to be very small. They become larger when the correct directions are identified (around 1200-1300 steps) and gradually die down as the model converges. We think the LE matrix $E(t)$ controls the “effective sizes” in your question, but we do apologize if we misunderstand the meaning of “effective sizes” in your question.
>
> **Modeling at initialization.** We think the sensitivity of LE-SDE/LE-ODE to initialization depends mainly on the choice of the surrogate $M$, whose temporal part governs the scale of feature interaction along time. Around initialization, $\alpha(t)$ and $\beta(t)$ are usually small, which can alleviate "mismodelling around initialization", as experiments on L-model showed.
>
> > *Secondly the paper does not justify ... by W. Su an co-authors.*
>
> **Simplifying influences onto features.** Sorry that we were not clear enough - here the same assumption of effective training is used. As training proceeds, the features of the same class become more similar in the underlying feature space. We agree with you that LE defined by W. Su and co-authors also considered intra-class variations, i.e., different samples in the sample class might exhibit LE to different degrees. We think at this stage, this technical detail makes modeling unnecessarily more complicated while the benefit is only marginal. We chose to keep the “first-order” information by considering class memberships, which captures the main (and we think one of the most important) aspect of the LE. To generalize our LE-SDE to also model intra-class variations of LE effects, one possible avenue we have considered is to pull this effect out from the drift to the diffusion term by constructing the diffusion matrix more carefully. We only mentioned this in brevity in the original submission to avoid confusion but added more details in the Appendix of the revised version.
>
> **Initialization.** Around initialization, although features from different classes may overlap with each other (in the fixed feature space dedicated by our choice of $H$, for example), the very low LE strength made them hard to affect each other before a favorable direction for classification was reached (which is around 1200-1300 iteration in our experiments on GeoMNIST). This actually fit in the LE framework by noting:
> - (a) The LE developed by W. Su and co-authors refined that the LE phenomenon should become stronger as training progresses, for the very reason of random initialization.
> - (b) In the LE-SDE/LE-ODE model, recall that the LE matrix $E(t)$ depends on time; thus one *expects* $\alpha(t)$ and $\beta(t)$ to be very small around initialization, and increases as training progresses when LE phenomenon holds. This is observed when training is effective and *not* observed when training is not effective (Cf. Figure D.8 as we increased the label corruption ratio).
>
>
> > *Thirdly — a point which ...  highly nonlinear.*
>
> Indeed, we restricted our attention to the LE-ODE to circumvent the estimation of the covariances in the LE-SDE. We wish to clarify that the nonlinearity of neural nets is mainly for the parameters and activations therein. The underlying feature space *should* be able to discriminate features in a less nonlinear fashion (as otherwise, take the logit features as an example, one cannot distinguish the features correctly by a generalized linear model such as the last softmax layer of the NN). Although the LE-SDE/LE-ODE has a drift term that is linear in the feature $X(t)$, it is nonetheless not time-homogeneous, thus offering more flexibility and making the theory more interesting and less obvious.
>
> > *One specific comment ... any classification yet.*
>
> Good question! Yes, even if we are using a neural net, the same can be said that $E X^1(0) = E X^2(0)$ since weights are zero-mean Gaussian initialized. However, we are *not* concerned about equality in expectation: the theorems require that $c_1 \ne c_2$ with probability one, which holds in theory since these are continuous random variables. In practice, this probability can be safely omitted.
>
> > *Another concern ... proposed guesses of H.*
>
> **Figures 4 and 5.** Thanks for reading our experiments thoroughly.
> The choice of $H$ in the L-model: Yes, you are right that choice $H$ can be viewed to be derived from the well-trained net. But the flexibility in the time-dependent LE strength $E(t)$ made it possible for the model to correct its behavior around initialization. We will also include zoomed-in versions of Figure 5 that show the behavior near initialization, which we think to be quite aligned for the case of L-mode.
>
> **Deviations in Figure D.8.** Yes, we also think the $H$ in the L-model provides a decent way for imitating feature dynamics but is definitely not perfect. The merit of the choice of $H$ in the L-model is that it preserves the dominant coordinate for logit features (that is, the $k$-th coordinate of the features from the $k$-class) while being somewhat oblivious about the rest (the $j$-th coordinate of the features from the $k$-class for $j \ne k$). This manifests in Figure D.8 from the observation that the dominant coordinate can be imitated more faithfully (Cf. Figures D.8(k) and D.8(l), which we guessed to be the “deviation” you mentioned). We discussed in Appendix D.2 (equation D.20) possible avenues for addressing this issue to more faithfully imitating the dynamics of other coordinates by incorporating the non-dominant directions into the definition of $H$, but we did not go into details in the experiments. Still, we think the L-model did a relatively good job in imitating the genuine dynamics.
>
> > *Fig 4(c) is also unconvincing ... random guessing.*
>
> The issue here is that estimation of the polynomial tail indices under the presence of noise is notoriously tricky, which warrants independent research interest. We think the essence of Figure 5 and D.8 is not about how close they fit into the exact regions as the theorem dictates, but whether there is an apparent phase transition correlated with the label corruption ratio. Note that in both Figure 4(c)  and Figure D.2(b), we observe that at around $p = 0.6$,
> - the tail index of $\alpha$ surpasses that of $\beta$, violating the condition in the theorem;
> - the tail indices are overall larger than those with $p < 0.6$.
>
> You are right that the validation accuracy is significantly better than random guessing for $p = 0.5$, which seems to contradict the behavior of $p = 0.5$ in Figure D.2(b), but it is not straightforward to pin down exactly around what point does the phase transition occur since the estimated indices for $\alpha$ and $\beta$ are pretty similar when $p = 0.5$, and one may argue that it is hard to say whether the conditions in the theorem fails in this case.
>
> **Some other comments**
>
> > *Corollary 3.4  ... other term.*
>
> We should state more clearly that Corollary 3.4 requires the conditions in Theorem 3.1 since Corollary 3.4 assumes features are separable. When $A(t) = t$, the tail of $\alpha(t)$ is not decaying at all for the Corollary to hold.
>
> > *The proofs are ... to infinity.*
>
> This part of the proof is when we substitute back the solution to LE-ODE back to the LE-SDE, which is equation C.9 for the general case. The fastest way of seeing this is to fix a large but finite $T$ (when the model is guaranteed to be convergent) and perform the analyses within $0 < t < T$. We added more details in the Appendix explaining these. If you have any other concerns about the proof, please definitely let us know, and we’d be happy to elaborate them in greater detail.
>
> > *In Fig. (4) ... readable.*
>
> Sorry for the confusion; it meant "... crosses the horizontal line of r = 1, entering ...".

---

> ### Author Response · Authors · 2021-08-31
> **Further Response to Reviewer imQZ**
>
> Thanks for your reply and for considering increasing your rating!
> We wish to take this opportunity to clarify your further concerns.
>
> > I do not agree with the argument that c1 ≠ c2 assumption can be alleviated by taking the results "in probability".
>
> Thanks for following up with this question! We realized that this question might have arisen because we didn’t clarify what probability space we are working on and with respect to what sigma field we are taking the expectations. Sorry about that.
>
> Using the binary case an example, in the proof, the condition $c_1 \ne c_2$ is needed for arriving at a conclusion from the last equation in Appendix A, which is essentially of the form
>
> $$
> (c_1 - c_2) e^{C t} + D(n) > E N(t),
> $$
>
> where $C > 0$, $D(n)$ models the initial condition, $E > 0$ is a positive constant, and $N(t)$ is a standard Gaussian process. By construction, $c_k = \mathbb{E}[X_1^k(0)]$ for $k = 1,2$.
>
> Here we are working with a **fixed** neural network that is randomly initialized. Let $\theta$ be its parameters; in the definition of $c_k$ we meant
>
> $$
> 	c_k = \mathbb{E}_{\mathrm{data}}[ X^k(0) | \theta = \theta_0 ],
> $$
>
> where $\theta_0$ is a realization of the weights under the random initialization; $X^k(0)$ is the feature from the $k$-th class under the data distribution. Here the expectation averages out the randomness from the dataset only.
>
> Hence when we wrote $c_1 \ne c_2$, we meant “*for a given NN, the expected feature from the first class is different from that from the second class over the data distribution*.” This statement is true when we ignore a null set w.r.t. the probability measure induced by the randomness of the network initialization. Overall, this statement can be interpreted as “*the expected feature at initialization from the first class is different from that from the second class for an NN, except possibly on a null set in the space of NNs (wrt the probability measure induced by the parameter initialization scheme)*.”
>
> As we see, here the statement $c_1 \ne c_2$ is actually in the almost sure sense (w.r.t. the parameter initialization), but the sense of probability would suffice for us to conclude that as $t \to \infty$, separation happens with probability tending to $1$.
>
> We didn’t use probability formalisms before as we thought they might unnecessarily complicate the formula as we write out sigma fields for the data, parameter initialization, mini-batch sampling, etc. We have revised the main text to emphasize the randomness we are working and hopefully, they could clear out the ambiguity.
>
>
> > Why at each time t, F(X(t),t), in the authors' response, is modeled by a linear relationship with X(t).
>
> Thanks for clarifying the question. We agree that modeling the drift term as linear in $X(t)$ contains limitations that our previous response didn't fully address.
> We have added a detailed discussion in the Appendix on how we obtain the linearization from the SDE for the features (equation (B.4) in Appendix B). We hope that you could take a look at them in a further response we added separately in the main thread.
>
> In that derivation, we have used the following conditions (with the effective training assumption assumed):
> 1. Decoupling of the expectation (Equation (B.4));
> 2. Linearize the drift by first-order Taylor expansion of the drift $F(X(t), t)$;
> 3. Taylor expansion again around convergence.
>
> The first point is argued when intra-class features are similar (analogous to the decoupling used in deriving the iterative maps in the NTK paper); the second point is what we realize to be a commonly used method in the filtering community, though we did not find many rigorous references.
> Finally, analyses show that the residue of the approximation vanishes as training progresses, when we expand around convergence.
>
> The most important limitation is that the linearization could only cater to either the initialization stage or the terminal stage (Cf. our discussions in the main thread). This is indeed a limitation of linearity, in the same spirit as we cannot fit three non-colinear points using a straight line. Our intuition is to fit the terminal stage (which has a vanishing residue), and use the fact that the $F(X(t), t)$ is small around initialization to argue the residue is bounded in this regime.
>
> Despite this limitation, we still think the LE-SDE/LE-ODE may of value to the community for the following reasons:
>
> **Tractability:** closed-form solutions are easy to obtain; the quadratic form, for example, would make it much complicated and less intuitive.
>
> **Approximability:** with the above discussion, we think the approximation error should not be too severe. Indeed, as we will show by a few new figures, we observe the approximation error is indeed reasonable.
>
> **Novelty:** we hope this work could open new doors in using phenomenological models as surrogates to study NN dynamics. The road towards unboxing NNs is long, our climb will be steep, but we are still hopeful that we will get there.
>
>
>
> > Why ignoring higher-order terms? Are they much smaller than the linear term?
>
> Thank you for pointing this out! This question is very important, but we did not elaborate enough in the paper or our previous responses. We will squeeze a few new figures showing the approximation errors measured by the relative difference defined for each class $k \in [K]$ as
>
> $$
> 	\operatorname{\mathsf{RD} _ k}(t) :=
> 		\frac{|| X^k(t) - Y^k(t) | |_ {H^k} }
> 		     {\left(|| X^k(t) || _ 2 + ||Y^k(t) || _ 2 \right) / 2},
> $$
>
> where $X^k(t)$ and $Y^k(t)$ are genuine and simulated features of $k$-th class at time $t$, respectively; the norm in the numerator is with respect to the $H^k$ matrix (the identity matrix for the I-model, and the $H^k$ defined in equation (9) in Section 3.2.2) and the denominator is the usual $2$-norm. Please find below the link to these figures (*anonymously hosted*):
>
> - Figure 1 (GeoMNIST + I-Model): https://imgur.com/a/c1sGS7F
> - Figure 2 (GeoMNIST + L-Model): https://imgur.com/a/JnX8QCk
>
> We observe that the L-model has a vanishing RD, which partly supports our previous discussion. The error around initialization is not small, though, a limitation of linearization. However, we think that error is reasonable (since the values $X^k(t)$ around initialization is small, the absolute error is bounded). The imperfection of this approximation indicates that to exactly recover the dynamics, one must go beyond linearization, which is also our ongoing quest.

---

> > ### Comment · Reviewer_imQZ · 2021-09-02
> > **thanks**
> >
> > I thank the authors for the responses!
> >
> > Thanks for clarifying the $c_1\neq c_2$ issue; it's indeed an error on my end.
> >
> > Thanks for giving a more thorough discussion on the linear modeling assumption. From my understanding, the linearization is essentially limited by where to linearize (initialization or convergence), which is the concern in my initial review.
> >
> > I think the discussion will take longer than the scope of NeurIPS reviewing format. I would not change my recommendation at this point, though I do not have a strong opposition against acceptance. I foresee good discussions around the limitations and potential future works if this paper is ultimately accepted.

---

### Official Review · Reviewer_qrk4 · 2021-07-10

**Rating:** 7
**Confidence:** 4

**Summary:**

The paper provides a simple phenomenological model of neural networks for K-class classification. Each sample is assigned a feature modeling NN features, and these features are evolved via SDEs in which the drift term of a sample is a linear transformation of the average features of each class. Separation, which means the case in which samples from different classes can be separated by hyperplanes, holds w.h.p. when the intra-class drift coefficients larger than the inter-class coefficients in a certain sense, and it does not hold otherwise. Two specific choices of features are considered: isotropic features and logit features. The latter are shown to be better than the former at modeling the feature evolution for NN on CIFAR and a dataset of their own creation.


**Limitations And Societal Impact:**

The paper has no potential negative societal impact. The limitations of the paper are clear from the beginning: the paper follows a phenomenological approach, which precludes a deep understanding of the factors at play but might be beneficial in terms of the simplicity and predictive power of the model.

**Main Review:**

The paper is well written and clear. Its greatest contribution is the finding that such a simple model is able to describe the feature evolution of neural networks, as shown in experiments. This work shows, with its limitations, that the phenomenological approach can in some settings produce results that closely resemble the real NN. It opens up several questions that I hope can be answered in future works: When is this model valid? Can we somehow infer the drift terms from the neural network weights, and then design architectures that favor separation?

When presenting the experiments, the plots for $\hat{\alpha}$ and $\hat{\beta}$ are very noisy and it is unclear what the decay of this quantities is.  If possible, it would be interesting to denoise the plots and show them in a log-log scale, to see whether indeed the separation results claimed in Theorem 3.1 translate to real NN.

Typos: The paper needs to be proofread for typos. I point out some. In line 133 $c_x, c_y$ should be $c_1, c_2$. In line 184, “semi-positive definite” should be “positive semi-definite”.  In line 188, I believe that the sentence should be “the features are asymptotically pairwise separable with probability 0,” as the current sentence means that the features are asymptotically pairwise separable with probability 1.

**Time Spent Reviewing:**

2.5

---

> ### Author Response · Authors · 2021-08-10
> **Response to Reviewer qrk4**
>
> Response to Reviewer qrk4
>
> Thanks for reviewing our paper and your comments! We are really excited to have your positive opinions on exploring our formulation further. We also think studying model validity and using LE-SDE/ODE to inform architectural design is very promising. We believe concurrent works with such efforts, including ours, can hopefully unveil more insights on the relationship between LE and DNNs. Please find below our responses to your comments, which we believe will significantly improve the presentation of the final version, provided that our papers were accepted.
>
>
> > *When presenting the experiments, the plots for α^ and β^ are very noisy ...*
>
> Thanks for the suggestion! We did try to obtain better visualization. One of our attempts was to plot them in a semilog scale in the $x$ axis, yet the large noise drives many estimates below zero in an interval where the values should be positive. The negative estimates prevent us from the use of loglog plot. To overcome that, we tried various ways to estimate and denoise $\hat{\alpha}(t)$ and $\hat{\beta}(t)$ including
> - (i) Using moving-averages.
>   - (a) Moving average on $\hat{A}(t)$ and $\hat{B}(t)$ (triangular/Gaussian windows).
>   - (b) Numerical differentiation of the results.
>   - (c) Moving average again (triangular and Gaussian windows).
> - (ii) Directly applying the Savitzky-Golay filter on $\hat{A}(t)$ and $\hat{B}(t)$ with order 1 to obtain differentiated and smoothened $\hat{\alpha}(t)$ and $\hat{\beta}(t)$.
>
> We found the results were visually similar when the window sizes are similar. Although using a larger window size could lead to much smoother plots, some details around the boundaries might be missing. We wondered if you have any suggestions or pointers for better smoothing techniques? We are more than happy to try that.
>
> > *In line 188, I believe that the sentence should be ...*
>
> Yes, we've fixed the typos. Sorry for the confusion; your understanding of line 188 is absolutely correct. Our verbosity in phrasing those sentences was an attempt to distinguish “not separable with probability 1” and “separable with probability 0.”

---

> ### Author Response · Authors · 2021-09-05
> **Message to Reviewer qrk4**
>
> Dear Reviewer qrk4,
>
> Many thanks for your helpful and insightful comments that have helped us significantly improved the presentation of our paper. Does any of our responses address our questions in a satisfactory manner? Please let us know if you have other questions and we are always happy to further address questions. Thank you!

---

### Official Review · Reviewer_zWV1 · 2021-07-13

**Rating:** 5
**Confidence:** 3

**Summary:**

This paper builds off a recently proposed phenomenon of deep learning models, local elasticity, which, says the that impact in feature space of a gradient update from an input is in general larger for feature of data in the same class as the input rather than a different class. In this work the authors propose a set of SDE's modeling SGD that captures the inter and intra class effect of back propagation. They find that there is a sharp phase transition in the dynamics governed by these SDEs when they do or don't display local elasticity. In particular, when the SDEs are locally elastic then the features of the training data are guaranteed to become linearity separable, while this is not the case otherwise. They demonstrate this theoretical analysis empirically with CIFAR-10 and connect there results to another recently proposed phenomenon, neural collapse.

**Limitations And Societal Impact:**

Yes.

**Main Review:**

**Originality/Significance:**
The authors seek to understand how neural networks are able to achieve linear separability explaining the recent empirical phenomenon of neural collapse.  They conjecture that this is caused by the network possessing local elasticity.  They construct a SDE in feature space that under certain assumptions will provably lead to feature separability when local elasticity is met.  My greatest concern with this work was that it was not clear why the SDE they use should reflect the actual feature dynamics of a real neural network.  This rational was not made clear theoretically and as far as I could understand the empirical explorations required training a DNN to estimate parameters of their model, which was then simulated, and then compared back to the empirical data used in the estimation.  Even accepting the validity of this model to track the feature dynamics it was not clear how this understanding should inform future work or answer questions like why is depth needed or non-linearities to achieve feature separability.  Despite these comments I thought the direct modeling of feature dynamics rather than just parameter dynamics was interesting and could be a promising direction for future work.

**Clarity/Quality:**
This work has grammatical and spelling mistakes throughout (some listed below) and should be more carefully edited. The overall structure was fine and I appreciated the authors use of a simple setting for introducing the theory before diving into the more general expressions. I found the empirical setup (in particular the estimation procedure) very difficult to follow. The authors did not spend enough time focusing on assumptions applied in their theory or limitations of their empirical approach to "validating" the theory. Finally, future work and related work should be expanded, with particular emphasis on how this model could inform algorithmic improvement or answer open questions.


**Major Areas for Improvement:**

 - Equation (1) is the basis of your analysis, but you made very little effort to build it up or discuss the assumptions applied.  For example, why should we expect that the dynamics of the features are driven by a linear drift term? How should I think of equation (1) being derived or implied from the SGD parameter updates? All the related work you discuss in "SGD as SDEs" is about SDEs in parameter space, your SDE derived from eq. 1 is operating in feature space. This is a significant difference and should be discussed!
 - Discussion on "Local Elasticity and Phenomenological Models" could be much more detailed.  This is clearly the closest related work section and it would be good to go into detail about their claims and the implications of follow up works.
- Is $\alpha(m)$ and $\beta(m)$ implicitly assumed to be non-negative? For example, is $\alpha,\beta \ge 0$ in theorem 2.1?
- What is the relationship between equation (2) and an Ornstein-Uhlenbeck process?  Imagine the setting of $n=1$ and let $Z(t) = [X^1(t), X^2(t)]$, then doesn't equation (2) describe the OU process $dZ(t) = AZ(t) + \sigma dW(t)$ where $A = \frac{1}{2}\begin{bmatrix}\alpha &\beta\newline \beta & \alpha \end{bmatrix}$?  If this is the case then how does theorem 2.1 relate to the eigenvalues of $A$?  I think this perspective/connection to a classical SDE would be helpful for understanding.
- In equation (5) the potentially anisotropic diffusion $\Sigma_t$ kind of came out of nowhere.  So in your SDE model you do assume the noise is anisotropic and it is potentially temporally dependent?  This is not the case in equation (2)?  Discussing what is being generalized here would be helpful.
- The analysis and experiments assume a batch size of 1?  How would the results change for a realistic scenario with mini batches (batch size = m > 1)?
- Could you provide more intuition for the expressions in equation (11)?
- To estimate $\alpha(t)$ and $\beta(t)$, "each experiment is repeated for ntrial = 100 independent runs".  Does this mean you train a neural network 100 independent times keeping track of the logits through time? The estimated $\hat{\alpha}(t)$ and $\hat{\beta}(t)$ are then used to simulate your SDE and these dynamics are displayed over one of the trials of training in figure 5? Can you confirm that your simulated DNN dynamics require training a DNN?  I apologize if I didn't follow the experimental setup.
- You write "This link between a tractable mathematical model (LE-SDE/ODE) and an expressive yet complicated black box would potentially yield greater benefits towards demystifying the magic of deep models" but the future directions seem to be strategies for expanding the model rather than providing explanation for how to use the model.  For example, does the model suggest hyperparameter settings or architecture design that would facilitate separability?


**Specific Comments:**

- [equation 4] - equation (1) is between m+1 and m, while equation (4) is between m and m-1.  Might consider making these consistent?
- [equation 4] - In the current notation it wasn't clear to me why equation (1) can't capture equation (4) (i.e. why can't I pull the $H$ into the definition of $E$)?
- [line 47] - "essentialingredient" => essential ingredient
- [line 58] - "emerge" => emergence
- [line 74] - ". also appears" => missing a word?
- [line 116] - "that common" => that is common or commonly discussed
- [line 190-191] "discussions on the empirically best ways of choosing the universal direction $\nu$ are detailed in the Appendix"... what is the universal direction?  Where is $\nu$ discussed earlier?
- [line 196] - "H is SDP" => do you mean SPD (semi-positive definite)?
- [line 325] - "Aa" => "A"

**Time Spent Reviewing:**

5

---

> ### Author Response · Authors · 2021-08-10
> **Response to Reviewer zWV1**
>
>
> Thanks for reviewing our paper and for your comments! We really appreciate that you found the binary setting useful for introducing the theory and your interest in our work’s many details. Please find below our responses to your comments.
>
> **Modeling choice.**
> Thanks for raising this question! We would also like to invite you to our response in the main thread. Our main goal in this paper is to model the relationship between local elasticity (LE) and DNNs, and (hopefully) to convince the reader that the LE property should be reflected in a (good) deep learning theory. We wish to render our model as straightforward as possible while capturing the LE phenomenon, and these simple sets of SDEs/ODEs serve our goal, though it is clear that the *real* dynamics of DNN (in feature space) is far *more* complex. Our modeling is not a “vanilla” linearization of the drift coefficient, but rather, we allow the LE strength matrix $E(t)$ to be time-dependent while assuming the similarity measure in the feature space is fixed throughout (which is specified by the $H$ matrix). The rationale is that the strength of LE naturally changes as training progresses, yet how the features interact with each other in the feature space should be measured using a fixed ruler. In fact, we continued to improve the presentation and polish the paper after submission, and the next version shall address all these points much more clearly, provided that the paper was accepted.
>
> **Applications and future work.**
> Thanks for pointing out applications of the LE-SDE model, such as reasoning about the necessity of using nonlinearity or informing architectural design. We think these are crucial questions and would be use cases of our LE-SDE/LE-ODE model. Although we did not discuss these aspects in great detail, we observed that the subtlety of the behavior of $\alpha(t)$ often correlates with model performance (cf. Figures D.5 and D.6). This observation hints that the evolution of $\alpha(t)$ is likely highly dependent on those architectural choices, and a possible avenue for addressing these questions might be a finer-grained analysis of how these architectural choices affect $\alpha(t)$.
>
> **Empirical setup is hard to follow.**
> Thanks for raising this issue; we are aware that the empirical section was perhaps treated too briefly, albeit we tried to include more details in Appendix D. The main idea, as you have pointed out, of using genuine dynamics of NNs to estimate the hyper-parameters in the LE-ODE model, is to validate whether the LE-ODE model is expressive enough to imitate the dynamics of NNs, similar in spirit to calculating the residue of a fitted regression model. The results (Figure 5 and D.8) show that the simple I-model did a relatively poor job, but the simulated trajectory using the L-model seemed more convincing. This suggests that the L-model can capture most of the trajectorial information in NNs, despite its simplicity.
>
> We have included more discussions of the empirical procedures, focusing on the intuitions, as you also requested.
>
> **Major areas for improvement**
>
> Please find below our per-point responses.
>
> > *Equation (1) is the basis of your analysis, but you made very little effort to build it up or discuss the assumptions applied... This is a significant difference and should be discussed!*
>
> Our LE-SDE/LE-ODE is motivated from both an LE perspective and an SDE perspective. Taking the L-model as an example, let us denote by $F(X(t), t)$ the drift term in the feature dynamics of DNNs (equation B.3, which can be derived from the SGD updating equation for parameters), our goal is to find a tractable $M$ surrogate for the drift term such that $M X(t)$ and $F(X(t), t)$ are equal as $t \to \infty$, provided that training is effective. The structure of the map $F$ informs us the space it operates in as $t \to \infty$, and a natural choice for $M$ would be to “decompose” it into a projection to this space (the $H$ space), and a magnitude component $E(t)$. Note that the magnitude $E(t)$ is time-dependent; thus we do not assume the features interact at the same scale in the feature space $H$ throughout training, and this is not a “vanilla” linearization. We think this choice offers a balance of expressivity and tractability.
>
> > *Discussion on "Local Elasticity and Phenomenological Models" could be much more detailed.*
>
> Thanks, we added more discussions on LE in the main texts, similar to our above explanation on how we modeled the relationship between LE and feature dynamics.
>
> > *Is $\alpha(m)$ and $\beta(m)$ implicitly assumed to be non-negative?*
>
> We interpret $\alpha(m)$ and $\beta(m)$ (and their continuous version) as quantities determined by the dataset, training algorithm, model architecture, etc, and do NOT assume they are non-negative. The main theorem only requires conditions on their relative order of tails.
>
> > *What is the relationship between equation (2) and an Ornstein-Uhlenbeck process...*
>
> Yes, you are right, equation (2) is an OU process, and the eigenpairs of matrix $A$ in your notation determines the invariant measure: a positive eigenvalue indicates there is a direction that can separate $X^1(t)$ and $X^2(t)$ as $t \to \infty$. We too think it is interesting to apply tools from classical SDE to analyze LE-SDE, such as expected hitting time to a region where the model is sufficiently accurate, which is one direction we’ve been following up. We did not link much to classical SDE in hope to simplify the writing but will add several remarks in the right place. Please let us know if you have any pointers or thoughts on this.
>
> > *In equation (5) the potentially anisotropic diffusion...*
>
> Yes, in the 1D model (equation (2)) we used the simplest setup with the same constant diffusion across classes. Our intended way for generalization is to model the diffusion
> $\Sigma(t)  = \frac{\Sigma^b(t) \otimes I_K}{S} + \operatorname{diag}\left(n_k^{-1} \Sigma_k(t)\right)_k$
> by intra- and inter-class components modeled separately by $\Sigma_k(t)$ and $\Sigma^b(t)$ with different scalings. Although these would become more important when we perform analyses on the LE-SDE, since the experiments of our paper focused on LE-ODE, we keep the discussion on $\Sigma(t)$ minimal to avoid confusion. We added a paragraph in the future work section and more details in the Appendix on this.
>
> > *The analysis and experiments assume a batch size of 1 ...*
>
> Yes, the batch size is fixed to $1$ in the experiments, but the theory applies to the general setting where $m > 1$ (and also to when there is label corruption, or non-uniform sampling is used) by adjusting the sampling matrix $P$ (Cf. Section 5 and ), we also added more details on it in the Appendix. The case when $m > 1$ corresponds to changing the drift $M(t) = E(t) \otimes H$ to $M(t) = (E(t) P) \otimes H$ in the LE-SDE and a change of the scaling of the covariance matrix accordingly. The effect of $m$ is smoothed out when we move into LE-ODE so long as mini-batch sampling is uniform; and will be captured by the structure of the sampling matrix $P$ otherwise.
>
> > *Could you provide more intuition for the expressions in equation (11)?*
>
> We are sorry that there is a small typo in equation (11); please see Appendix D or the code in the supplementary for the correct expression. The main idea is to view the eigenvectors of the $Kp$-by-$Kp$ drift matrix $M$ as a concatenation of $K$ vectors of dimension $p$ and constructs linear combinations of them such that one or more independent components in the solution (e.g., the first term in equation 8) can be canceled out in order to estimate an expression involving $A(t)$ and $B(t)$ in the other term (e.g., the exponent in the second term in equation 8). This can be done for the I-model, and for the L-model under certain conditions (details in Appendix D).
>
> > *To estimate α(t) and β(t) ...*
>
> Yes, we trained NNs (variants of the AlexNet model) for $N=100$ independent times, keeping track of the logits through time. We used these results to obtain estimates $\hat{\alpha}(t)$ and $\hat{\beta}(t)$ and fed them into the LE-ODE model. Simulating dynamics using LE-ODE requires knowledge of $\alpha(t)$ and $\beta(t)$. We used the estimates $\hat{\alpha}(t)$ and $\hat{\beta}(t)$ for this purpose. The goal of these experiments is to check the validity of the L-model: if the model is a poor surrogate of the feature dynamics, no matter how hard we try to estimate the parameters in the model, the simulated dynamics cannot be close to the genuine ones.
>
> > *You write "This link ...*
>
> Good question! We think expanding/generalizing the model and identifying practical use cases of the model should be studied concurrently. Since we know that the behavior of $\alpha(t)$ and $\beta(t)$ affect separation, analyzing how architecture design and hyperparameters in the training procedure interact $\alpha(t)$ and $\beta(t)$ would be an avenue in this regard. Similarly, when we have a more refined/generalized model, the dependence of the model parameters on the architecture design would become clearer, too.
>
>
> **Specific comments.** Thanks again for your careful reading; we’ve fixed the typos and made notations more consistent per your suggestion.
>
> > *[equation 4] - In the current notation ...*
>
> Yes, you are correct, we can pull $H$ into the definition of $E$, but we think writing the drift as $M = E \otimes H$ makes the structural properties and our intention clearer since we model the time-dependent LE strength matrix $E(t)$ and a fixed matrix $H$ that defines similarity in the feature space separately. We think this may also make it easier to connect with the binary case in Section 2 (when $H$ is just a scalar $1$).

---

> > ### Comment · Reviewer_zWV1 · 2021-08-14
> > **Thank you for your clarifications**
> >
> > Thank you for the detailed response and for addressing individually the areas for improvement I suggested.  Your answers were helpful.  I also appreciate the authors discussion of their modeling choice.  However, I still don't feel convinced for why I should expect the modeling choice used to reflect real dynamics or given this model that there would be valuable implications for future work (beyond simply expanding the model).  As such, I am not inclined to change my rating.

---

> > > ### Author Response · Authors · 2021-08-31
> > > **Further Response to Reviewer zWV1**
> > >
> > >
> > > Thank you for your reply! We’re happy that you found our response helpful, and we respect your decision. But allow us to take this chance to try to clarify your further concerns.
> > >
> > > ### Linearization
> > >
> > > The construction of the LE-SDE/LE-ODE model captures the LE phenomenon, which was shown to be prevalent in well-trained models. It follows from W Su’s series of works that when a model is not LE, very likely it does not learn well. Hence under the effective training assumption, with modeling parameters estimated from real dynamics, the LE-SDE/LE-ODE model should capture at least the “LE aspect” of the real dynamics. Indeed, experiments demonstrated that they were even more capable in imitating real dynamics to a certain extent provided the model specification makes sense (Cf. comparison between the I-model and the L-model).
> > >
> > > We have also added more details to the Appendix on the reasons/conditions of the linearization $M(t)X(t)$ from our ongoing following-up work. Due to its length, we added them as a separate response in the main thread. We hope you could kindly take a look at it. In summary, the approximation relies on the following three conditions (with the effective training assumption assumed):
> > > 1. Decoupling of the expectation (Equation (B.4)). This is similar to the argument used to derive iterative covariances in the NTK literature;
> > > 2. Linearize the drift by first-order Taylor expansion of $F(X(t), t)$. We read that this is actually a common method in the filtering community, though we did not find many references.;
> > > 3. Taylor expansion again around convergence. The residue vanishes as the model converges.
> > >
> > > We think the major limitation is that linearization could only accommodate for either the initialization stage or the terminal stage (like we cannot fit a line exactly to three non-colinear points). We choose to fit the terminal stage and argue that due to the drift being small around initialization, the absolute approximation error is also small in this regime. In fact, this limitation is intrinsic to linearization.
> > >
> > > Despite this limitation, we think the merit of the model lies in its tractability and simplicity, its curious connection between separation results and LE, and its demonstrated effectiveness of imitating genuine dynamics. Concretely, we added a few new figures showing the approximation error (defined as a relative difference that ranges from $0$ to $2$, the lower, the better), please see below for *anonymous* links:
> > >
> > > - Figure 1 (GeoMNIST + I-Model): https://imgur.com/a/c1sGS7F
> > > - Figure 2 (GeoMNIST + L-Model): https://imgur.com/a/JnX8QCk
> > >
> > > These figures support our argument: the vanishing of error as training progresses; relatively large RD around initialization (note that the absolute value of the error is still small in this regime).
> > >
> > > We hope this could illustrate the merits of our model in the current form, and the potential of building more exact and complex surrogates on top of it.
> > >
> > > ### Future Work and Broader Impact
> > >
> > > For future work, while expanding the model has immediate practical use cases, we did start off on the journey to the approximation and convergence results of the LE-SDE/LE-ODE. We have preliminary results on correlating this with the LE behavior, which depends on the model architecture, training dataset, and optimization algorithms. We think this line of future work would shed more light on understanding feature dynamics and is potentially valuable.

---

> > > ### Author Response · Authors · 2021-09-02
> > > **Message to Reviewer zWV1**
> > >
> > > Dear Reviewer zWV1,
> > >
> > > Many thanks for your response that enabled us to further improve our paper. As the deadline is approaching, we are wondering if you can kindly take a look at our additional response and we hope that this has carefully addressed your concern. Thanks!

---

### Official Review · Reviewer_J1us · 2021-07-13

**Rating:** 7
**Confidence:** 5

**Summary:**

This paper proposed a stochastic differential equation model based on the local elasticity assumption of features of neural networks.  The local elasticity phenomenon says samples have greater influence to other samples in the same class than those in different classes. Starting from this assumption, a system of linear SDEs is proposed for features corresponding to each data point, with the coefficients encoding the difference of inter and intra classes impact. Two types of features are studied---the isotropic feature learning model, and the logics-as-features model. For the two models, dynamics of the class means are derived as a system of ODEs and the solutions are found, depending on the coefficients about local elasticity. From the solutions, the separability of classes is theoretically connected with local elasticity. Specifically, the classes are separable if the local elasticity effect is nonzero, while not separable otherwise. The transition between the two cases is sharp.

Numerical experiments are conducted to justify the effectiveness of the proposed model to characterize the actual dynamics of features. Experiments are done on deep convolutional networks on the GeoMNIST and CIFAR10 dataset. Firstly, the coefficients in the SDEs (and the ODEs for the class means) are estimated using real feature dynamics given by the training process (I understand this step as a kind of "fitting"). Then, the ODEs with estimated coefficients are simulated, and the results are compared with the real feature dynamics. It shows that in many cases (but not all) the trajectories produced by the SDEs recover the real trajectories given by the training.

**Limitations And Societal Impact:**

The authors have address the limitations of the work. Since this is a fundamental study not directly related to any applications, I do not think societal impact is necessary.

**Main Review:**

This paper provides a new way to understand the training process of deep neural networks. Specifically, the dynamics of features, rather than the parameters, is considered. Due to the stochastic nature of the training, the dynamics of features is formulated as SDEs. The work is not purely theoretical, since its modeling and analysis is built on the local elasticity assumption, which is observed in practice but not proven theoretically. However, considering the local elasticity is a special property for nonlinear machine learning models, such as deep neural network, and does not appear for linear models, the LE-SDEs still provide some insights to the interaction between different data during the training process of neural networks. Numerical experiments also show that the class means given by LE-SDEs with properly calculated coefficients match the real class means well.

My major concern about the work lies on the linear formulation of the LE-SDE. By the formulation, the influence of one data to other data is linear. (this influence actually happens on features, of course). However, no explanation or justification is provided for this linear formulation. Does it come from the stochastic gradient descent update, or come from the choices of features? Will the linear formulation still work if other optimization algorithms are used, or other quantities in the neural network are picked as features? The authors should discuss the rationality and potential limitation of the linear formulation.

The paper is generally well written and easy to read. Though, there are still some typos and seemingly wrong expressions (perhaps I'm wrong). To mention one, in the equations between line 121 and 122, \alpha and \beta are fixed in the expressions, regardless of the classes L_m. I believe this is not the right formulation.

**Time Spent Reviewing:**

4 hours

---

> ### Author Response · Authors · 2021-08-10
> **Response to Reviewer J1us**
>
>
> Thanks for reviewing our paper and for your comments! We are excited that you find the insights into the interaction between different data during the training process of neural networks interesting. Please find below our per-point responses to your comments.
>
> > *My major concern about the work lies on the linear formulation of the LE-SDE... The authors should discuss the rationality and potential limitation of the linear formulation.*
>
>
>
> - **Formulation of the LE-SDE, rationality, and limitation.** This is a common concern raised by other reviewers; we would also like to invite you to the response in the main thread. We wish to emphasize that the primary goal of our modeling is not to *exactly* recover the training dynamics of DNNs, but rather to explore how local elasticity (LE) interacts with DNN training. Moreover, through modeling their relationships, we wish to uncover how LE affects separability in DNN models. Although the $H$ matrix in the I-model and L-model is fixed, we think the freedom offered by the time-dependent LE strength matrix $E(t)$ provides ample expressive power for us to gain insights and imitate DNN feature dynamics.
>
>   In our modeling, the main idea is to approximate the drift term by a surrogate that is linear in $X(t)$ that eventually maps features to the same points (note though, the surrogate is still time-dependent, and the equation is non-time-homogeneous). In fact, we think it can be motivated by both the SGD update equation and the LE phenomenon (by viewing the $M(t)$ matrix as defining per-class similarities). (i) From the LE perspective, the LE coefficient matrix $E(t)$ tracks the strength of the LE phenomenon, whereas the $H$ matrix defines a way to measure similarity between features in the feature space. The intuition is that as training progresses, the LE strengths can change over time, whereas we assume the underlying feature space is fixed. (ii) From the SGD perspective, taking the L-model as an example, we can derive the feature dynamics in DNNs (equation B.4 in Appendix B). Let us denote by $F(X(t), t)$ the drift term in equation B.4, our approximation amounts to identifying a surrogate $M$ such that $F(X(t), t) = M X(t)$ as $t \to \infty$ provided training is effective.
>
>
>  - **Future work.** We really appreciate your constructive comments, especially in investigating other algorithms. This is a very important direction that our current paper does not address due to space limitations. Our submission only considers SGD (although allowing mini-batch, label-corruptions, and non-uniform sampling) but leaves the case under other optimization algorithms untouched. For example, it would be potentially exciting to derive a similar LE-SDE/LE-ODE model catering to accelerated gradient methods. Since there are great works on ODE/SDE views for understanding SGD and accelerated gradient algorithms such as by W. E and collaborators; B. Shi, W. Su, M. Jordan and collaborators, and more, the starting point is quite well-studied. We have some preliminary ideas on this, and it’s great to know that it may appeal to a larger audience in the community. Please let us know if you have any pointers or thoughts in this regard.
>
> >  *In the equations between line 121 and 122, $\alpha$ and $\beta$ are fixed in the expressions, regardless of the classes $L_m$.*
>
> - **Equation between lines 121 and 122.** In general, our model can generalize to allow $\alpha(t)$ and $\beta(t)$ to depend on the class indices. Indeed, we think this should be the direction for generalizing our model to more complex datasets where the LE effect among classes is non-homogeneous. On the other hand, this generalization would make deriving closed-form solutions harder while not improving the results in the current theorems. To avoid the overly complicated dependence on the class indices, we adopt the non-trivial yet straightforward case throughout the paper: there are only two free parameters in $E(t)$, $\alpha(t)$ and $\beta(t)$, at any fixed time $t$.

---

### Author Response · Authors · 2021-08-10
**General Response on Modeling Choice**

First of all, we want to thank all four reviewers for carefully reviewing our paper and their constructive comments. We have read them thoroughly, fixed the typos, and added more details as suggested. We are pleased to see several reviewers expressed their interests in our work, suggested future directions, and pinpointed the novelty and strength of our paper, including modeling the link between local elasticity (LE) phenomenon and DNN training, as well as its potential to imitate DNN dynamics.

We want to respond here to the common question raised by Reviewers J1us, zWV1, and imQZ on the modeling choice we made in the LE-SDE/LE-ODE, in particular, whether the drift term $(E(t) \otimes H) X(t)$ is expressive enough to capture the dynamics in DNN. We feel that we should really make this clearer in our original submission, but hopefully, it is still not too late to explicitly state our goals and rationales here.

### Modeling Goals

We want to clarify that our LE-SDE/LE-ODE is designed to capture how the LE phenomenon manifests in DNNs, which is the first of its kind to the best of our knowledge. To *exactly* recover deep learning dynamics is goodwill, yet it is too ambitious to reach for the current LE-SDE/ODE model. This difficulty also applies to the popular NTK and mean-field formalisms, which have been good modeling tools despite limitations. Our goal is to provide a *meaningful* model that captures LE for understanding the training dynamics of DNNs while remaining tractable for theoretical analyses. This work’s *ultimate* aim is to demonstrate that local elasticity is indispensable for *faithfully* modeling the dynamics of deep learning training, and our LE-SDE/LE-ODE is merely a simple (perhaps the simplest) but arguable effective approach. We understand that our submission didn’t sufficiently reflect this important point due to space and time constraints. In the new version, we’ll definitely make this point much clearer.

Our model, while not sophisticated enough to recover the *exact* dynamics of DNNs, is shown capable of *imitating* the dynamics to a great extent while enabling us for analyzing its separability thanks to the time-dependent formulation of the LE coefficient matrix $E(t)$ (or the coefficients $\alpha(t)$ and $\beta(t)$). We believe our model has the potential to facilitate us understand DNNs better and inform architectural design.

### Intuitions and Rationales

We would like to emphasize that our LE-SDE/LE-ODE is not a simple linear approximation; the assumptions we made were not for simplifying dynamics in DNNs, but for capturing how LE manifests its effect in the training process. Concretely, the drift coefficient $M(t)$ in our model encompasses an LE matrix $E(t)$ and a transformation matrix $H$. The matrix $E$ captures the magnitude of the LE effects, and matrix $H$ characterizes the space in which the LE operates (analogous to the kernel similarity used in He & Su’s original paper). Our model assumes LE strengths change over time, but it traces the similarity between features using a fixed “ruler” in the feature space by virtue of the fixed $H$. The intuition behind this is that the underlying feature space should be fixed (though unknown), and the training pushes the features to be more discriminative in this feature space (while the magnitude of LE can change over time).

Given the above discussions, we would like to state more intuitions and rationales of our model through the following two lenses. We want to reiterate that the core assumption of our model lies in “effective training,” that is, as training progresses, the features become more discriminative: features from the same class are more similar whereas those from different classes are more distinct, measured by some similarity measure in the feature space.

- **The LE perspective.** Our intuition is to approximate the drift term in the real dynamics by a surrogate $M$ that works the best when the model is locally elastic. Recall that the phenomenon of local elasticity states that the intra-class effects outweigh the inter-class effects, and the distinction becomes more significant as the training progresses. In our LE-SDE/LE-ODE model, the LE matrix $E(t)$ measures the *strength* of LE effects, which changes over time due to training, whereas the $H$ matrix defines the feature space and similarity within. For example, under the L-model, this similarity is given by $<X^k_i | H_{kl} |X^l_j>$ when $X^l_j$ is the trained sample (note that it is not symmetric). We chose the $H$ matrix to define the feature space when the model is well-trained for this reason. When the model is not locally elastic (e.g., near initialization), the approximation of our model is mainly controlled by the temporally-dependent $E(t)$ matrix, and the LE coefficients are generally small (Cf. Figure D.3) in this regime.

- **The SGD perspective.** Taking logit features as an example, in the updating equation of logit features (Equation B.1, where the approximation is with respect to step size $h \to 0$), the true dynamics of the features can be derived (Cf. Equation B.3). Let us denote by $F(X(t), t)$ the drift term, our goal is to find a surrogate $M$ such that $F(X(t), t) = M X(t)$ as $t \to \infty$ under the effective training assumption. Now the genuine dynamics $X(t)$ from NN training and the simulated dynamics using the surrogate $Y(t)$ (this notation was not used in the original submission) is such that $X(0) = Y(0)$ (by initial conditions of the LE-SDE/LE-ODE) and $X(\infty) = Y(\infty)$ (thanks to effective training and the choice of $H$ in the L-model) in expectation over the randomness of initialization and dataset sampling. The time-varying LE matrix $E(t)$ allows the features to interact can be viewed as performing scaling in the temporal domain to align the two curves at both $t = 0$ and $t = T$ for a large $T$ when the model is convergent. This allows features to interact at different time scale in the feature space $H$ throughout training. Around initialization, the feature space is yet to be learned, the LE strengths $\alpha(t)$ and $\beta(t)$ are very small, causing the model to be rather oblivious to the gradient updates; whereas as training progresses local elasticity emerges with increasing LE strengths. Note that this behavior is observed in our experiments under different datasets and architectures.


Although we could have modeled more complicated schemes for the drift term, such as a quadratic form in $X(t)$ (which we did not include in the paper), we won’t be having much gain in terms of expressive power, yet the theoretical details would be much more cumbersome for both the readers and us. The current modeling is a simplification that already leads to crisp insights as far as LE is concerned. We strived to strike a balance between expressivity and tractability, and we feel the decomposition into the temporal-dependent LE strength $E(t)$ and a fixed similarity “measure” $H$ in the feature space is the right abstraction level at this stage and is a good first step to take. We hope our work will be pioneering in the study of relationships between LE and DNNs that will open new doors for future research.

---

### Author Response · Authors · 2021-08-31
**Further Discussions on Linearization**


Thank you all four reviewers, for reading our responses! We're happy to see that many reviewers found our responses clarifying and helpful. We wish to respond here to the common question by reviewers zWV1 and imQZ by providing more details on the modeling choice, the limitations, and why we think it is effective. These are based on our ongoing following-up work on the approximation results of the LE-SDE and we have added them in the Appendix.

We hope this could demonstrate the usefulness and potential of the LE-SDE/LE-ODE, and hopefully could convince you to kindly re-considering the significance/impact of this work.

### Approximation of the Non-linear Drift

We will use the logit features and the L-model as an example, though the argument can be extended to any post-activation features. Writing Equation (B.4) in terms of $X(t) \in \mathbb{R}^{Kp}$ (recall in this case $p=K$), the concatenation of $K$ per-class feature vectors $X^k(t) \in \mathbb{R}^p$ for $k \in [K]$, as we did in the main text, and denoting by $\sigma:\mathbb{R}^K \to \mathbb{R}^K$ the softmax function for simplicity, we can express the drift term as

$$
F(X(t), t) = \Theta(t) \left( \left[ e_k - \sigma(X^k(t)) \right]_{k=1}^K \right),
$$

where we wrote $[\cdot]$ for vector concatenation and $\Theta(t) \in \mathbb{R}^{(Kp)\times (Kp)}$ is the Gram matrix. A commonly used linearization method in SDE for non-linear drifts (mostly by the filtering community, for example see Chapter 9.1 of [1]) is to linearize for each $t$ at the mean $\phi(t) =(\phi_k(t))_{k=1}^K \equiv \bar{X}(t) :=\mathbb{E}_B X(t)$ where the expectation is taken w.r.t. the Brownian motion. Concretely, we have
$$
    F(X(t), t) \approx \tilde{F}(X(t), t) := F(\phi(t), t) + \nabla_X F(\phi(t), t) (X(t) - \phi (t)),
$$

where $\nabla_X F$ denotes the Jacobian of $F$ wrt the spacial variable. For notation simplicity, introduce

$$
    p=(p_k)_{k=1}^K \in \mathbb{R}^{Kp}, \quad p_k := \sigma (X^k(t)) \in \mathbb{R}^p, \quad k \in [K],
$$

and similarly $ \bar{p} = (\bar{p} _ k) _ {k=1}^K, \quad \bar{p} _ k := \sigma (\bar{X}^k(t))$. Writing $J _ {kk} = J _ k := \operatorname{diag}(\bar{p} _ k)-\bar{p} _ k\bar{p} _ k^T$,
clearly, the Jacobian $\nabla F(\phi, t) = J(t)$ with $J(t) = (J _ {kk}) _ {k=1}^K$ being the block-diagonal matrix of per-class Jacobians. Now continuing linearization, we can write

$$
\begin{aligned}
    \tilde{F}(X(t), t) &= \Theta(t) \left( [e_k - \bar{p}_k]_k + J(t)(X(t) - \phi(t))\right) \\\\
    &= \Theta(t) \left( J(t) X(t) + \left[ e_k - \bar{p}_k + J_k \phi_k(t) \right]_k\right).
\end{aligned}
$$

Define $\Psi: \mathbb{R}^{Kp} \to \mathbb{R}^{Kp} : z \mapsto [e_k - \sigma(z_k)]_k$ and write $\Psi_k: \mathbb{R}^{p} \to \mathbb{R}^p$ to be the $k$-th component of $\Psi$, using Taylor's theorem to expand $\Psi(z)$ around $\phi(t)$ for each $t$, we have

$$
    \Psi = \Psi(\phi) + J(t)\phi - J(\phi) z + o\left(||z-\phi||\right),
$$

or

$$
    \Psi(\phi) + J(t) \phi = \Psi(z) + J(\phi) z + o\left(||z-\phi||\right).
$$

This implies that

$$
\tilde{F} = \Theta(t) J(t) X(t) + \Theta(t) R(t), \quad R(t;z) := \Psi(z) + J(t) z + o\left(||z-\phi(t)||\right),
$$

where $R(t;z)$ is the residue that depends on the choice of $z$ around which $\Psi(\phi)$ is expanded. Note that no matter what $z$ we choose (which can depend on $t$), the dependence on the feature $X(t)$ is **always through a time-varying linear map**, which motivates our choice of the time-varying linear map $M(t)$ in the LE-SDE model.


By choosing different $z$, we can approximate different stages of the real dynamics using the LE-SDE. Two particular choices may be of interest:

- Around initialization. Let $z=u := c \cdot [\boldsymbol{1}_K/K] _{k=1}^K$  be a scaling of vectors of ones where $c$ is some fixed constant. Then each of the $K$ components of $\sigma(u)$ assigns approximately the same probability ($1/K$) for every label. Furthermore, $u \in \operatorname{Ker} J(t)$ for all $t$ hence the residue $R(t; u) = \Psi(u) + o\left(||z-\phi(t)||\right)$ is a constant vector (which is colinear with $d$ defined in Equation (10)). Note that this approximation works best when the model has not learned (the "first stage" as we call it) since $||u-\phi(t)||$ is small in this regime.

- Around convergence. Given that the model converges, $\phi_{\infty} := \phi(\infty)$ is finite. Let $z=\phi_{\infty}$, under the effective training assumption, $|| \Psi(\phi_{\infty}) || \approx 0$ by construction. Hence the residue $R(t; \phi_ {\infty}) =  J(t)\phi_ {\infty} + o\left(||\phi(t)-\phi_{\infty})||\right)$. Here the $o(\cdot)$ term converges to $0$ as training progresses, leaving us a term that is asymptotically equivalent to $v = (v_k) _ {k=1}^K := J(\phi _ {\infty}) \phi _ {\infty} \in \mathbb{R}^{K^2}$, where $v_k = [(z_{k,i}- \sum_{j=1}^K p _ {k,j} z_{k, j}) p _ i] _ {i=1}^K \in \mathbb{R}^K$. Again, under the effective training assumption $z_k$ has its $k$-th entry $z_{k,k}$ the largest, and $p_k$ has its $k$-th entry close to $1$ while the others to zero. Thus $v_{k,i} \approx 0_K$. We see that in this regime, the approximation $\Theta(t) J(t) X(t)$ is only off by a residue $o\left(||\phi(t) - \phi_{\infty}||\right)$ that eventually vanishes.

For this reason, we choose to linearize the drift at convergence instead of around initialization in the L-model (Cf. Appendix B2) such that the residue vanishes under the effective training assumption.

To wrap up, we have used the following conditions:
1. Decoupling of an expectation in equation (B.4);
2. Linearize the drift in $X(t)$; and
3. Taylor expansion around convergence.

We think the first condition can be assumed when intra-class data samples do not vary much, such that an argument similar to that used in the derivation of CNTK [3] can be used; the second condition is mainly used in the filtering community (Chapter 9 of [1]), which we did not find many rigorous references. We think extensions to second-order approximations are possible and have been investigating this (Cf. Chapter 9 of [2]). Lastly, although our choice of expansion only caters for models at convergence, we think the fact that $\Theta(t)$ being small around initialization (argued heuristically and also shown in the experiments) could alleviate the error.  We do not have a quantitative result on this error yet, but we hope to get it done in our following-up work.

### Visualizing Approximation Error

We plan to squeeze a few new figures showing the approximation errors (the same setup as we did for Figures 4 and 5) to demonstrate that the LE-SDE can approximate decently well at convergence, and the result is reasonable around initialization. We measure the error through relative difference (RD) defined for each class $k \in [K]$ as

$$
\operatorname{\mathsf{RD} _ k}(t) := \frac{|| X^k(t) - Y^k(t) || _ {H^k} }{\left(|| X^k(t) || _ 2 + ||Y^k(t) || _ 2 \right) / 2},
$$

where $X^k(t)$ and $Y^k(t)$ are the real and simulated dynamics of the $k$-th class, respectively. Below, we show Figure 1 RD of the GeoMNIST + I-model and Figure 2 RD of the GeoMNIT + L-model. All set up is identical to our experiment section, whose detail is given in the “Simulations of the LE-ODE” paragraph of Appendix D.2.

Here the norm in the numerator is with respect to the $H^k$ matrix (the identity matrix for the I-model, and the $H^k$ defined in equation (9) in Section 3.2.2) and the denominator is the usual $2$-norm. This choice measures normalized difference under the similarity defined by $H^k$.
RD ranges from $0$ to $2$ (lower the better), and it can alleviate extremely small values of the estimates.

**Figure 1 (GeoMNIST + I-Model):** Anonymous imgur link: https://imgur.com/a/c1sGS7F

**Figure 2 (GeoMNIST + L-Model):** Anonymous imgur link: https://imgur.com/a/JnX8QCk

From the results, we observe that:
1.  L-model performs better than I-model in that (i) the RD is generally smaller for all $t$; and (ii) no significant difference between classes;
2.  L-model performs better at the later stages of training, as supported by the above discussion. In particular, the approximation became better as training progresses (indicated by decreasing RD). However, the performance of the L-model around initialization is still commendable;
3.  L-model is not perfect - the RD is not driven to zero (but at the order of $10^{-2}$ to $10^{-1}$), indicating the non-dominant direction is also important for capture the remainder of the feature similarity (Cf. discussions on “Simulations of the LE-ODE in Appendix D.2).

We agree that it is not straightforward to argue whether the absolute value of the RD indicates the drift approximation is good or not since there is no baseline. Nevertheless, we hope these figures could help the readers to have a better idea of the approximation error of LE-SDE/ODE.

### Epilogue

To summarize, in the revised paper, we added these details on why we think linearization $M(t)X(t)$ of the drift is reasonable. It is certainly not perfect -- as we discussed, it can approximate either the initialization state or the terminal stage well (limitation of *all* linearizations). Trading-off for tractability, we hope this simple yet effective model could open new doors for using surrogate phenomenological models to study properties of real dynamics, and its interesting link between separation and LE could inspire researchers to look at NNs through a new lens.

### References

[1] Simo Särkkä, Arno Solin. _Applied Stochastic Differential Equations_ (Institute of Mathematical Statistics Textbooks). Cambridge: Cambridge University Press.

[2] Andrew Jazwinski. _Stochastic Processes and Filtering Theory_. Academic Press.

[3] Sanjeev Arora, Simon S. Du, Wei Hu, Zhiyuan Li, Ruosong Wang. _Fine-Grained Analysis of Optimization and Generalization for Overparameterized Two-Layer Neural Networks_. https://arxiv.org/abs/1901.08584

---

### Decision · Program_Chairs · 2021-09-27

**Decision:**

Accept (Poster)

**Comment:**

This paper studies the training dynamics of neural network by considering a stochastic differential equation model based on the local elasticity assumption of features of neural networks. The proposed model is justified by numerical experiments and sheds new light on the training dynamics. Overall the referees all find this paper interesting and novel, and hence the meta-reviewer would recommend acceptance of the paper as a poster.